# Posterior Sampling by Combining Diffusion Models with Annealed Langevin Dynamics

**Zhiyang Xun**
UT Austin
zxun@cs.utexas.edu

**Shivam Gupta**
UT Austin*
shivamgupta@utexas.edu

**Eric Price**
UT Austin & Microsoft Research
ecprice@cs.utexas.edu

## Abstract

Given a noisy linear measurement $y = Ax + \xi$ of a distribution $p(x)$, and a good approximation to the prior $p(x)$, when can we sample from the posterior $p(x \mid y)$? Posterior sampling provides an accurate and fair framework for tasks such as inpainting, deblurring, and MRI reconstruction, and several heuristics attempt to approximate it. Unfortunately, approximate posterior sampling is computationally intractable in general.

To sidestep this hardness, we focus on (local or global) log-concave distributions $p(x)$. In this regime, Langevin dynamics yields posterior samples when the exact scores of $p(x)$ are available, but it is brittle to score–estimation error, requiring an MGF bound (sub-exponential error). By contrast, in the unconditional setting, diffusion models succeed with only an $L^2$ bound on the score error. We prove that combining diffusion models with an *annealed* variant of Langevin dynamics achieves conditional sampling in polynomial time using merely an $L^4$ bound on the score error.

## 1 Introduction

Diffusion models are currently the leading approach to generative modeling of images. Diffusion models are based on learning the "smoothed scores" $s_{\sigma^2}(x)$ of the modeled distribution $p(x)$. Such scores can be approximated from samples of $p(x)$ by optimizing the score matching objective [HD05]; and given good $L^2$-approximations to the scores, $p(x)$ can be efficiently sampled using an SDE [SE19, HJA20, SSDK+21] or an ODE [SME20].

Much of the promise of generative modeling lies in the prospect of applying the modeled $p(x)$ as a *prior*: combining it with some other information $y$ to perform a search over the manifold of plausible images. Many applications, including MRI reconstruction, deblurring, and inpainting, can be formulated as *linear measurements*

$$y = Ax + \xi \qquad \text{for} \qquad \xi \sim \mathcal{N}(0, \eta^2 I_m) \tag{1}$$

for some (known) matrix $A \in \mathbb{R}^{m \times d}$. *Posterior sampling*, or sampling from $p(x \mid y)$, is a natural and useful goal. When aiming to reconstruct $x$ accurately, it is 2-competitive with the optimal in any metric [JAD+21] and satisfies fairness guarantees with respect to protected classes [JKH+21].

Researchers have developed a number of heuristics to approximate posterior sampling using the smoothed scores, including DPS [CKM+23], particle filtering methods [WTN+23, DS24], Diff-PIR [ZZL+23], and second-order approximations [RCK+24]. Unfortunately, unlike for unconditional sampling, these methods do not converge efficiently and robustly to the posterior distribution. In fact, a lower bound shows that *no* algorithm exists for efficient and robust posterior sampling in

---

*Now at Google DeepMind.

39th Conference on Neural Information Processing Systems (NeurIPS 2025).

general [GJP+24]. But the lower bound uses an adversarial, bizarre distribution $p(x)$ based on one-way functions; actual image manifolds are likely much better behaved. Can we find an algorithm for provably efficient, robust posterior sampling for relatively nice distributions $p$? That is the goal of this paper: we describe conditions on $p$ under which efficient, robust posterior sampling is possible.

A close relative to diffusion model sampling is *Langevin dynamics*, which is a different method for sampling that uses an SDE involving the *unsmoothed* score $s_0$. Unlike diffusion, Langevin dynamics is in general *slow* and *not robust* to errors in approximating the score. To be efficient, Langevin dynamics needs stronger conditions, like that $p(x)$ is log-concave and that the score estimation error satisfies an MGF bound (meaning that large errors are exponentially unlikely).

However, Langevin dynamics adapts very well to posterior sampling: it works for posterior sampling under exactly the same conditions as it does for unconditional sampling. The difference from diffusion models is that the *unsmoothed* conditional score $s_0(x \mid y)$ can be computed from the unconditional score $s_0(x)$ and the explicit measurement model $p(y \mid x)$, while the *smoothed* conditional score (which diffusion needs) cannot be easily computed.

So the current state is: diffusion models are efficient and robust for unconditional sampling, but essentially always inaccurate or inefficient for posterior sampling. No algorithm for posterior sampling is efficient and robust in general. Langevin dynamics is efficient for log-concave distributions, but still not robust. Can we make a robust algorithm for this case?

*Can we do posterior sampling with* log-concave $p(x)$ *and* $L^p$-*accurate scores?*

## 1.1 Our Results

Our first result answers this in the affirmative. Algorithm 1 uses a diffusion model for initialization, followed by an *annealed* version of Langevin dynamics, to do posterior sampling for log-concave $p(x)$ with just $L^4$-accurate scores. Annealing is necessary here; see Section F for why standard Langevin dynamics would not suffice in this setting.

**Assumption 1** ($L^4$ score accuracy). *The score estimates $\widehat{s}_{\sigma^2}(x)$ of the smoothed distributions $p_{\sigma^2}(x) = p(x) * \mathcal{N}(0, \sigma^2 I_d)$ have finite $L^4$ error, i.e.,*

$$\mathbb{E}_{p_{\sigma^2}(x)}[\|\widehat{s}_{\sigma^2}(x) - s_{\sigma^2}(x)\|^4] \leq \varepsilon_{score}^4 < \infty.$$

**Theorem 1.1** (Posterior sampling with global log-concavity). *Let $p(x)$ be an $\alpha$-strongly log-concave distribution over $\mathbb{R}^d$ with $L$-Lipschitz score. For any $0 < \varepsilon < 1$, there exist $K_1 = \mathrm{poly}(d, m, \frac{\|A\|}{\eta\sqrt{\alpha}}, \frac{1}{\varepsilon})$ and $K_2 = \mathrm{poly}(d, m, \frac{\|A\|}{\eta\sqrt{\alpha}}, \frac{1}{\varepsilon}, \frac{L}{\alpha})$ such that: if $\varepsilon_{score} \leq \frac{\sqrt{\alpha}}{K_1}$, then there exists an algorithm that takes $K_2$ iterations to sample from a distribution $\widehat{p}(x \mid y)$ with*

$$\mathbb{E}\left[\mathrm{TV}(\widehat{p}(x \mid y), p(x \mid y))\right] \leq \varepsilon.$$

For precise bounds on the polynomials, see Theorem E.6. To understand the parameters, $\frac{\|A\|}{\eta\sqrt{\alpha}}$ should be viewed as the signal-to-noise ratio of the measurement.

**Local log-concavity.** Global log-concavity, as required by Theorem 1.1, is simple to state but a fairly strong condition. In fact, Algorithm 1 only needs a *local* log-concavity condition.

As motivation, consider MRI reconstruction. Given the MRI measurement $y$ of $x$, we would like to get as accurate an estimate $\widehat{x}$ of $x$ as possible. We expect the image distribution $p(x)$ to concentrate around a low-dimensional manifold. We also know that existing compressed sensing methods (e.g., the LASSO [Tib96, CRT06]) can give a fairly accurate reconstruction $x_0$; not *as* accurate as we are hoping to achieve with the full power of our diffusion model for $p(x)$, but still pretty good. Then *conditioned on $x_0$*, we know basically where $x$ lies on the manifold; if the manifold is well behaved, we only really need to do posterior sampling on a single branch of the manifold. The posterior distribution on this branch can be log-concave even when the overall $p(x)$ is not.

In the theorem below, we suppose we are given a Gaussian measurement $x_0 = x + \mathcal{N}(0, \sigma^2 I_d)$ for some $\sigma$, and that the distribution $p$ is nearly log-concave in a ball polynomially larger than $\sigma$. We can then converge to $p(x \mid x_0, y)$.

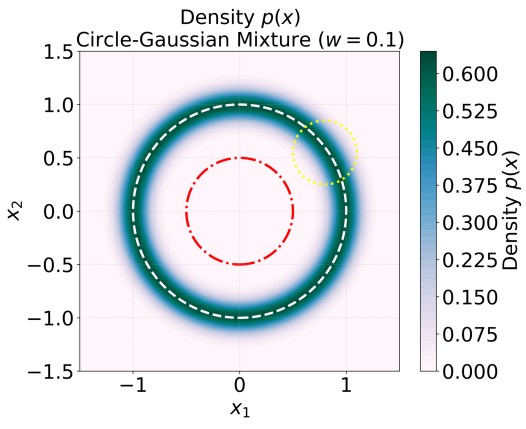

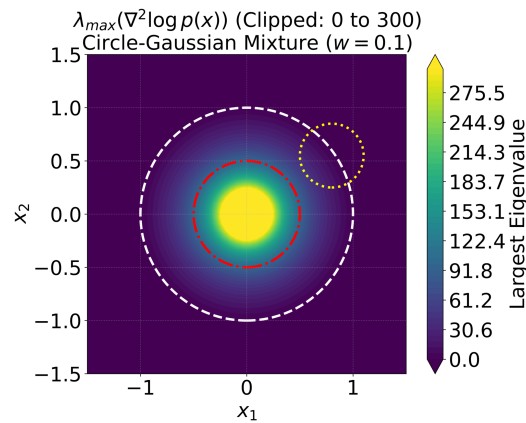

(a) Density of $p$, the uniform distribution over the unit circle (white), convolved with $\mathcal{N}(0, w^2 I_2)$.

(b) $\lambda_{\max}(\nabla^2 \log p(x))$ reaches $\Omega(1/w^4)$ near the center, demonstrating strong non-log-concavity.

Figure 1: A "locally nearly log-concave" distribution suitable for Theorem 1.2: uniform on the unit circle plus $\mathcal{N}(0, w^2 I_2)$. The Hessian's largest eigenvalue is much smaller near the bulk of the density than it is globally. Specifically, for $\|A\|w/\eta = O(1)$, a Gaussian measurement $\tilde{x}$ with $\sigma \leq cw$ and $\varepsilon_{\text{score}} \leq cw^{-1}$ for small enough $c > 0$ enables sampling from $p(x \mid y, \tilde{x})$.

**Theorem 1.2** (Posterior sampling with local log-concavity). *For any $\varepsilon, \tau, R, L > 0$, suppose $p(x)$ is a distribution over $\mathbb{R}^d$ such that*

$$\Pr_{x' \sim p}\left[\forall x \in B(x', R) : -LI_d \preceq \nabla^2 \log p(x) \preceq (\tau^2/R^2)I_d\right] \geq 1 - \varepsilon.$$

*Then, there exist $K_1, K_2 = \text{poly}(d, m, \frac{\|A\|\sigma}{\eta}, \frac{1}{\varepsilon})$ and $K_3 = \text{poly}(d, m, \frac{\|A\|\sigma}{\eta}, \frac{1}{\varepsilon}, L\sigma^2)$ such that: Given a Gaussian measurement $x_0 = x + \mathcal{N}(0, \sigma^2 I_d)$ of $x \sim p$ with $\sigma \leq R/(K_1 + 2\tau)$. If $\varepsilon_{score} \leq \frac{1}{K_2\sigma}$, then there exists an algorithm that takes $K_3$ iterations to sample from a distribution $\widehat{p}(x \mid x_0, y)$ such that*

$$\mathbb{E}_{y, x_0}\left[\text{TV}(\widehat{p}(x \mid x_0, y), p(x \mid x_0, y))\right] \lesssim \varepsilon.$$

If $p$ is globally log-concave, we can set $\sigma = \infty$ so $x_0$ is independent of $x$ and recover Theorem 1.1; but if we have local information then this just needs local log-concavity. For precise bounds and a detailed discussion of the algorithm, see Section E.2.

The largest eigenvalue of $\nabla^2 \log p(x)$ quantifies the extent to which the distribution departs from log-concavity at a given point. In Figure 1, we show an instance of a locally nearly log-concave distribution: $x$ is uniformly on the unit circle plus $\mathcal{N}(0, w^2 I_2)$. This distribution is very far from globally log-concave, but it is nearly log-concave within a $w$-width band of the unit circle. See Section E.4 for details.

**Compressed Sensing.** In compressed sensing, one would like to estimate $x$ as accurately as possible from $y$. There are many algorithms under many different structural assumptions on $x$, most

Table 1: Summary of theorems and corresponding algorithms.

| Theorem | Setting | Method | Target |
|---------|---------|--------|--------|
| Theorem 1.1 | Global log-concavity | Algorithm 1 | $p(x \mid y)$ |
| Theorem 1.2 | Local log-concavity with a Gaussian measurement $x_0$ | Run Algorithm 1 using $p(x \mid x_0)$ as the prior (Algorithm 2) | $p(x \mid x_0, y)$ |
| Corollary 1.3 | Local log-concavity with an arbitrary noisy measurement $x_0$ | Run Algorithm 2 but replace $x_0$ with $x_0' = x_0 + \mathcal{N}(0, \sigma^2 I_d)$ (Algorithm 3) | small $\|x - x_0\|$ |

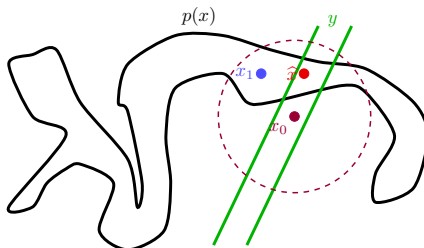

Figure 2: Corollary 1.3 sampling process. Given the distribution $p(x)$ and measurement $y$, we (1) start with a warm start estimate $x_0$, which may not lie on the effective manifold containing $p(x)$; (2) use the diffusion process to sample from $p(x)$ in a ball around $x_0$, getting $x_1$ on the manifold but not matching $y$; and finally (3) use annealed Langevin dynamics to converge to $p(x \mid y)$. This works if $p(x)$ is *locally* close to log-concave, even if it is *globally* complicated. See Section E.3 for a more detailed discussion.

notably the LASSO if $x$ is known to be approximately sparse [Tib96, CRT06]. The LASSO does not use much information about the structure of $p(x)$, and one can hope for significant improvements when $p(x)$ is known. Posterior sampling is known to be near-optimal for compressed sensing: if any algorithm achieves $r$ error with probability $1 - \delta$, then posterior sampling achieves at most $2r$ error with probability $1 - 2\delta$. But, as we discuss above, posterior sampling cannot be efficiently computed in general.

We can use Theorem 1.2 to construct a competitive compressed sensing algorithm under a "local" log-concavity condition on $p$. Suppose we have a naive compressed sensing algorithm (e.g., the LASSO) that recovers the true $x$ to within $R$ error; and $p$ is usually log-concave within an $R \cdot \mathrm{poly}$ ball; then if *any exponential time* algorithm can get $r$ error from $y$, our algorithm gets $2r$ error in polynomial time.

**Corollary 1.3** (Competitive compressed sensing). *Consider attempting to accurately reconstruct $x$ from $y = Ax + \xi$. Suppose that:*

- *Information theoretically (but possibly requiring exponential time or using exact knowledge of $p(x)$), it is possible to recover $\widehat{x}$ from $y$ satisfying $\|\widehat{x} - x\| \leq r$ with probability $1 - \delta$ over $x \sim p$ and $y$.*

- *We have access to a "naive" algorithm that recovers $x_0$ from $y$ satisfying $\|x_0 - x\| \leq R$ with probability $1 - \delta$ over $x \sim p$ and $y$.*

- *For $R' = R \cdot \mathrm{poly}(d, m, \frac{\|A\|R}{\eta}, \frac{1}{\delta})$,*

$$\Pr_{x' \sim p} \left[ \forall x \in B(x', R') : -LI_d \preceq \nabla^2 \log p(x) \preceq 0 \right] \geq 1 - \delta.$$

*Then we give an algorithm that recovers $\widehat{x}$ satisfying $\|\widehat{x} - x\| \leq 2r$ with probability $1 - O(\delta)$, in $\mathrm{poly}(d, m, \frac{\|A\|R}{\eta}, \frac{1}{\delta})$ time, under Assumption 1 with $\varepsilon_{score} < \frac{1}{\mathrm{poly}(d, m, \frac{\|A\|R}{\eta}, \frac{1}{\delta}, LR^2)R}$.*

That is, we can go from a decent warm start to a near-optimal reconstruction, so long as the distribution is locally log-concave, with radius of locality depending on how accurate our warm start is. To our knowledge this is the first known guarantee of this kind. Per the lower bound [GJP+24], such a guarantee would be impossible without any warm start or other assumption.

Figure 2 illustrates the sampling process of Corollary 1.3. The initial estimate $x_0$ may lie well outside the bulk of $p(x)$; with just an $L^4$ error bound, the unsmoothed score at $x_0$ could be extremely bad. We add a bit of spherical Gaussian noise to $x_0$, then treat this as a *spherical* Gaussian measurement of $x$, i.e., $x + \mathcal{N}(0, RI)$; for spherical Gaussian measurements, the posterior $p(x \mid x_0)$ *can* be sampled robustly and efficiently using the diffusion SDE. We take such a sample $x_1$, which now won't be too far outside the distribution of $p(x)$, then use $x_1$ as initialization for annealed Langevin dynamics to sample from $p(x \mid y)$. The key part of our paper is that this process will never evaluate a score with respect to a distribution far from the distribution it was trained on, so the process is robust to error in the score estimates.

---

**Algorithm 1** Sampling from $p(x \mid Ax + \mathcal{N}(0, \eta^2 I_m) = y)$

---

1: **function** POSTERIORSAMPLER($p : \mathbb{R}^d \to \mathbb{R}$ , $y \in \mathbb{R}^m$, $A \in \mathbb{R}^{m \times d}$, $\eta \in \mathbb{R}$)
2:     Let $\eta_1 > \eta_2 > \cdots > \eta_N = \eta$ and $T_1, \ldots, T_{N-1}$ be an admissible schedule.
3:     Initialize $y_N = y$
4:     **for** $i = N - 1$ down to 1 **do**
5:         $y_i = y_{i+1} + \mathcal{N}(0, (\eta_i^2 - \eta_{i+1}^2)I_m)$
6:     **end for**
7:     Sample $X_1 \sim p(x)$                    ▷ Approximately, using the diffusion SDE (5)
8:     **for** $i = 1$ to $N - 1$ **do**
9:         Let $\widehat{s}_{i+1}$ be the estimated score function for $s_{i+1}(x) = \nabla \log p(x \mid y_{i+1})$.
10:         Initialize $x_0 = X_i$.
11:         Simulate the SDE for time $T_i$:

$$\mathrm{d}x_t = \widehat{s}_{i+1}(x_t^{(h)}) \, \mathrm{d}t + \sqrt{2} \, \mathrm{d}B_t \qquad (2)$$

12:         Here, $x_t^{(h)} = x_{h \cdot \lfloor t/h \rfloor}$ is the discretized $x_t$, where $h$ is a small enough step size.
13:         Set $X_{i+1} \leftarrow x_{T_i}$
14:     **end for**
15:     **Return:** $X_N$ as an approximation of $p(x \mid Ax + \mathcal{N}(0, \eta^2 I_m) = y)$.
16: **end function**

---

We summarize our results in Table 1.

## 2  Notation and Background

We consider $x \sim p(x)$ over $\mathbb{R}^d$. The "score function" $s(x)$ of $p$ is $\nabla \log p(x)$. The "smoothed score function" $s_{\sigma^2}(x)$ is the score of $p_{\sigma^2}(x) = p(x) * \mathcal{N}(0, \sigma^2 I_d)$.

**Unconditional sampling.**    There are several ways to sample from $p$ using the scores. Langevin dynamics is a classical MCMC method that considers the following overdamped Langevin Stochastic Differential Equation (SDE):

$$dX_t = s(X_t)dt + \sqrt{2}dB_t, \qquad (3)$$

where $B_t$ is standard Brownian motion. The stationary distribution of this SDE is $p$, and discretized versions of it, such as the Unadjusted Langevin Algorithm (ULA), are known to converge rapidly to $p(x)$ when $p(x)$ is strongly log-concave [Dal17]. One can replace the true score $s(x)$ with an approximation $\widehat{s}$, as long as it satisfies a (fairly strong) MGF condition

$$\mathbb{E}_{x \sim p(x)} \left[ \exp \left( \|s(x) - \widehat{s}(x)\|^2 / \varepsilon_{mgf}^2 \right) \right] < \infty, \quad \text{for some } \varepsilon_{mgf} > 0. \qquad (4)$$

In particular, [YW22] showed that Langevin dynamics needs an MGF bound for convergence, and an $L^p$-accurate score estimator for any $1 \le p < \infty$ is insufficient.

An alternative approach, used by diffusion models, is to involve the smoothed scores. Starting from $x_0 \sim \mathcal{N}(0, I_d)$, one can follow a different SDE [And82]:

$$dX_t = (X_t + 2s_{\sigma_t^2}(X_t))dt + \sqrt{2}dB_t \qquad (5)$$

for a particular smoothing schedule $\sigma_t$; the result $x_T$ is exponentially close (in $T$) to being drawn from $p(x)$. This also has efficient discretizations [CCL$^+$22, CCSW22, BBDD24], does not require log-concavity, and only requires an $L^2$ guarantee such as [CCL$^+$22]

$$\mathbb{E}_{x \sim p_{\sigma^2}(x)} \left[ \|s_{\sigma^2}(x) - \widehat{s}_{\sigma^2}(x)\|^2 \right] < \varepsilon^2$$

to accurately sample from $p(x)$. One can also run a similar ODE with similar guarantees but faster [CCL$^+$23].

**Posterior sampling.** Now, in this paper we are concerned with *posterior sampling*: we observe a noisy linear measurement $y \in \mathbb{R}^m$ of $x$, given by

$$y = Ax + \xi \qquad \text{for} \qquad \xi \sim \mathcal{N}(0, \eta^2 I_m),$$

and want to sample from $p(x \mid y)$. The unsmoothed score $s_y(x) := \nabla_x \log p(x \mid y)$ is easily computed by Bayes' rule:

$$\nabla_x \log p(x \mid y) = \nabla_x \log p(x) + \nabla_x \log p(y \mid x) = s(x) + \frac{A^\top (y - Ax)}{\eta^2}.$$

Thus we can run the Langevin SDE (3) with the same properties: if $p(x \mid y)$ is strongly log-concave and the score estimate satisfies the MGF error bound (4), it will converge quickly and accurately.

Naturally, researchers have looked to diffusion processes for more general and robust posterior sampling methods. The main difficulty is that the smoothed score of the posterior involves $\nabla_x \log p(y \mid x_{\sigma_t^2})$ rather than the tractable unsmoothed term $\nabla_x \log p(y \mid x)$. Because the smoothed score is hard to evaluate exactly, a range of approximation techniques has been proposed [BGP$^+$24, CKM$^+$23, MK25, RCK$^+$24, SVMK23, WYZ23]. One prominent example is the DPS algorithm [CKM$^+$23]. Other methods include Monte Carlo/MCMC-inspired approximations [CJeILCM24, DS24, WSC$^+$24, EKZL25], singular value decomposition and transport tilting [KVE21, KSEE22, WYZ23, BH24], and schemes that combine corrector steps with standard diffusion updates [CL23, CY22, CSRY22, KBBW23, LKA$^+$24, SKZ$^+$24, SSXE22, ZZL$^+$23, AVTT21, XC24, RLdB$^+$24, RRD$^+$23]. These approaches have shown strong empirical performance, and several provide guarantees under additional structure of the linear measurement; however, general guarantees for fast and robust posterior sampling remain limited beyond these restricted regimes.

Several recent studies [JAD$^+$21, ZCB$^+$25, KVE21] use various annealed versions of the Langevin SDE as a key component in their diffusion-based posterior sampling method and achieve strong empirical results. Still, these methods provide no theoretical guidance on two key aspects: how to design the annealing schedule and why annealing improves robustness. None of these approaches come with correctness guarantees for the overall sampling procedure.

**Comparison with Computational Lower Bounds.** Recent work of [GJP$^+$24] shows that it is actually *impossible* to achieve a general algorithm that is guaranteed fast and robust: there is an exponential computational gap between unconditional diffusion and posterior sampling. Under standard cryptographic assumptions, they construct a distribution $p$ over $\mathbb{R}^d$ such that

1. One can efficiently obtain an $L^p$-accurate estimate of the smoothed score of $p$, so diffusion models can sample from $p$.

2. Any sub-exponential time algorithm that takes $y = Ax + \mathcal{N}(0, \eta^2 I_m)$ as input and outputs a sample from the posterior $p(x \mid y)$ fails on most $y$ with high probability.

Our algorithm shows that, once an additional noisy observation $\tilde{x}$ that is close to $x$ is provided, then we can efficiently sample from $p(x \mid y, \tilde{x})$, circumventing the impossibility result.

To illustrate why the extra observation helps, consider the following simplified version of the hardness instance:

$$p := q * \mathcal{N}(0, \sigma^2 I_d), \quad q(x) := \frac{1}{2^{d/2}} \sum_{s \in \{0,1\}^{d/2}} \delta((s, f(s)) - x).$$

Here, $f : \{0,1\}^{d/2} \to \{0,1\}^{d/2}$ is a one-way permutation — it takes exponential time to compute $f^{-1}(x)$ for most $x \in \{0,1\}^{d/2}$. $\delta(\cdot)$ is the Dirac delta function, and we choose $\sigma \ll d^{-1/2}$. Thus, $p(x)$ is a mixture of $2^{d/2}$ well-separated Gaussians centered at the points $(s, f(s))$.

Assume we observe

$$y = Ax + \mathcal{N}(0, \eta^2 I_d), \quad A = \begin{pmatrix} 0 & I_{d/2} \end{pmatrix}, \quad \sigma \ll \eta \ll d^{-1/2},$$

and let $\mathrm{rnd}(y)$ denote the vertex of $\{0,1\}^d$ closest to $y$. Then the posterior $p(x \mid y)$ is approximately a Gaussian centered at $(f^{-1}(\mathrm{rnd}(y)), \mathrm{rnd}(y))$ with covariance $\sigma^2 I_d$. Generating a single sample would therefore reveal $f^{-1}(\mathrm{rnd}(y))$, which requires $\exp(\Omega(d))$ time.

However, suppose we have a coarse estimate $x_0$ satisfying $\|x_0 - x\| < 1/3$ (e.g., obtained by compressed sensing). Then, $x_0$ uniquely identifies the correct $(s, f(s))$ with $f(s) = \mathrm{rnd}(y)$, and the remaining task is just sampling from a Gaussian. Therefore, this hard instance becomes easy once we have localized the task and does not contradict our Theorem 1.2.

We are able to handle the hard instance above well because it is exactly the type of distribution our approach is designed for: despite its complex global structure, it exhibits well-behaved local properties. This gives an important conceptual takeaway from our work: the hardness of posterior sampling may only lie in localizing $x$ within the exponentially large high-dimensional space.

Therefore, although posterior sampling is an intractable task in general, it is still possible to design a robust, provably correct posterior sampling algorithm — once we have localized the distribution. We view our work as a first step towards this goal.

# 3 Techniques

The algorithm we propose is clean and simple, but the proof is quite involved. Before we dive into the details, we provide a high-level overview of the intuitions behind the algorithm, concentrating on the illustrative case where the *prior* density $p(x)$ is $\alpha$-strongly log-concave. Under this assumption, every posterior density $p(x \mid y)$ is *also* $\alpha$-strongly log-concave. Therefore, posterior sampling could, in principle, be performed using classical Langevin dynamics.

The challenge arises because we lack access to the exact posterior score $s_y(x)$. We only possess an estimator derived from an estimate $\widehat{s}(x)$ of the *prior* score $s(x)$:

$$\widehat{s}_y(x) := \widehat{s}(x) + \frac{A^\top (y - Ax)}{\eta^2}.$$

Assumption 1 implies an $L^4$ accuracy of $\widehat{s}_y$ on average, but how do we use this to support Langevin dynamics, which demands exponentially decaying error tails?

## 3.1 Score Accuracy: Langevin Dynamics vs. Diffusion Models

> *Why can diffusion models succeed with merely $L^2$-accurate scores, whereas Langevin dynamics require MGF accuracy?*

Both diffusion models and Langevin dynamics utilize SDEs. The $L^2$ error in the score-dependent drift term relates directly to the KL divergence between the true process (using $s(x)$) and the estimated process (using $\widehat{s}(x)$). Consequently, bounding the $L^2$ score error with respect to the current distribution $\widehat{p}_t$ controls the KL divergence.

Diffusion models leverage this property effectively. The forward process transforms data into a Gaussian, and the reverse generative process starts exactly from this Gaussian. At any time $t$, suppose $\widehat{p}_t$ is close to $p_{\sigma_t^2}$, then

$$\underset{x_t \sim \widehat{p}_t}{\mathbb{E}}[\|s_{\sigma_t^2}(x_t) - \widehat{s}_{\sigma_t^2}(x_t)\|^2] \approx \underset{x_t \sim p_{\sigma_t^2}}{\mathbb{E}}[\|s_{\sigma_t^2}(x_t) - \widehat{s}_{\sigma_t^2}(x_t)\|^2] \leq \varepsilon_{\mathrm{score}}^2$$

by the $L^2$ accuracy assumption. This keeps the process *close* to the ideal process, ensuring overall small error.

Langevin dynamics, by contrast, often starts from an arbitrary, not predefined initial distribution $p_{\mathrm{initial}}$. An $L^p$ score accuracy guarantee with respect to $p_{\mathrm{target}}$ alone does not ensure accuracy for points $x_t$ that are not on the distributional manifold of $p_{\mathrm{target}}$ (consider running Langevin starting from $x_0$ in Figure 2). Therefore, a stronger MGF error bound is needed to prevent this from happening.

## 3.2 Adapting Langevin Dynamics for Posterior Sampling

While we can only use Langevin-type dynamics for posterior sampling, we possess a source of effective starting points: we can sample $x_0 \sim p(x)$ efficiently using the unconditional diffusion model. Intuitively, $x_0$ already lies on the data manifold. The score estimator $\widehat{s}_y(x)$ initially satisfies:

$$\underset{x_0 \sim p(x)}{\mathbb{E}}[\|s_y(x_0) - \widehat{s}_y(x_0)\|^2] = \underset{x_0 \sim p(x)}{\mathbb{E}}[\|s(x_0) - \widehat{s}(x_0)\|^2] \leq \varepsilon_{\mathrm{score}}^2.$$

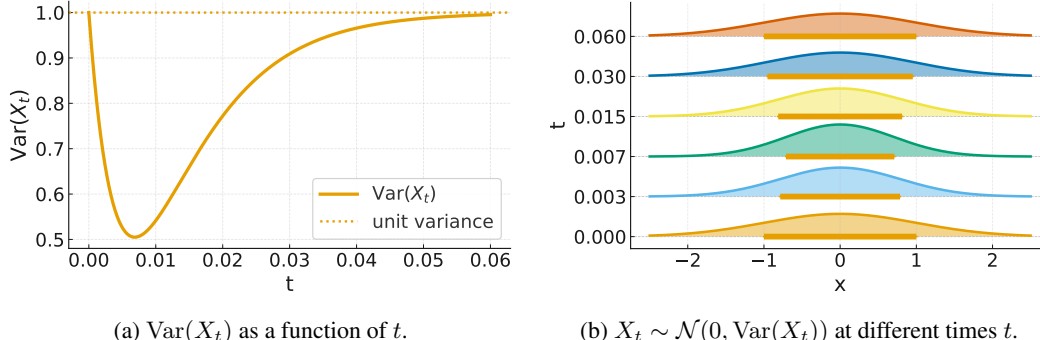

(a) $\mathrm{Var}(X_t)$ as a function of $t$.  (b) $X_t \sim \mathcal{N}(0, \mathrm{Var}(X_t))$ at different times $t$.

Figure 3: Let $p = \mathcal{N}(0, 1)$ and $y = x + \mathcal{N}(0, 0.01)$. Starting from $X_0 \sim p$, run the Langevin SDE $\mathrm{d}X_t = s_y(X_t)\,\mathrm{d}t + \sqrt{2}\,\mathrm{d}B_t$. Averaging over $y$, the marginal of $X_t$ remains Gaussian; its variance first contracts and then returns toward the prior. There is an intermediate time $t^*$ where $X_{t^*}$ has a constant factor lower variance; in high dimensions, this means $X_{t^*}$ is concentrated on an exponentially small region of $p$, so an $L^p$ bound on score error under $p$ does not effectively control the error under $X_{t^*}$. See Section F for details.

As the dynamics evolves, the distribution $p(x_t)$ transitions from $p(x)$ towards $p(x \mid y)$. If $x_t$ converges to $p(x \mid y)$, we again expect reasonable accuracy on average:

$$\mathbb{E}_y[\mathbb{E}_{x_t \sim p(x|y)}[\|s_y(x_t) - \widehat{s}_y(x_t)\|^2]] = \mathbb{E}_y[\mathbb{E}_{x_t \sim p(x|y)}[\|s(x_t) - \widehat{s}(x_t)\|^2]] \leq \varepsilon_{\text{score}}^2.$$

Hence the estimator is accurate at the start and at convergence. The open question concerns the *intermediate* segment of the trajectory: does $x_t$ wander into regions where the prior score $\widehat{s}(x)$ is unreliable? Ideally, the time-marginal of $x_t$, averaged over $y$, remains close to $p(x)$ throughout.

### 3.3  Annealing via Mixing Steps

In fact, even though $x_0$ and $x_\infty$ both have marginal $p(x)$, so the score estimate $\widehat{s}(x)$ is accurate on average at those times, this is *not* true at intermediate times. In Figure 3, we illustrate this with a simple Gaussian example: $x_0$ and $x_\infty$ have distribution $\mathcal{N}(0, I)$ while $x_t$ has marginal $\mathcal{N}(0, cI)$ for a constant $c < 1$. An $L^p$ error bound under $x \sim \mathcal{N}(0, I)$ does not give an $L^2$ error bound under $x \sim \mathcal{N}(0, cI)$, which means Langevin dynamics may not converge to the right distribution. A very strong accuracy guarantee like the MGF bound is needed here.

However, consider the case where the target posterior $p(x \mid y)$ is very close to the initial prior $p(x)$, such as when the measurement noise $\eta$ is very large (low signal-to-noise ratio). Langevin dynamics between close distributions typically converges rapidly. This suggests a key insight: if the required convergence time $T$ is short, the process $x_t$ might not deviate substantially from its initial distribution $p(x_0)$. In such short-time regimes, an $L^2$ score error bound relative to $p(x_0)$ could potentially suffice to control the dynamics. While $p(x)$ itself is already a good approximation for $p(x \mid y)$ when $\eta$ is very large, this motivates a general strategy.

Instead of a single, potentially long Langevin run from $p(x)$ to $p(x \mid y)$, we introduce an annealing scheme using multiple *mixing steps*. Given the measurement parameters $(A, \eta, y)$, we construct a decreasing noise schedule $\eta_1 > \eta_2 > \cdots > \eta_N = \eta$. Correspondingly, we generate a sequence of auxiliary measurements $y_1, y_2, \ldots, y_N = y$ such that each $y_i$ is distributed as $Ax + \mathcal{N}(0, \eta_i^2 I_m)$ and $y_i$ is appropriately coupled to $y_{i+1}$ (specifically, $y_i \sim \mathcal{N}(y_{i+1}, (\eta_i^2 - \eta_{i+1}^2)I_m)$ conditional on $y_{i+1}$). This creates a sequence of intermediate posterior distributions $p(x \mid y_i)$.

An *admissible schedule* (formally defined in Definition D.1) ensures that:

- $\eta_1$ is sufficiently large, making $p(x \mid y_1)$ close to the prior $p(x)$.

- Consecutive $\eta_i$ and $\eta_{i+1}$ are sufficiently close, making $p(x \mid y_i)$ close to $p(x \mid y_{i+1})$.

Our algorithm proceeds as follows:

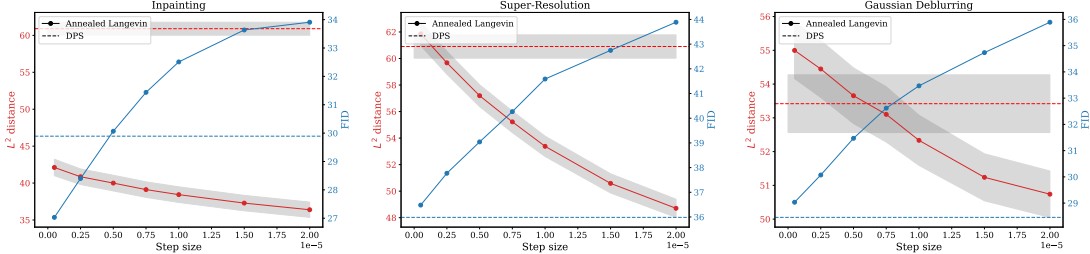

Figure 4: For each of the three settings (inpainting, super-resolution, and Gaussian deblurring), we plot the $L^2$ distance between samples obtained by our annealed Langevin method and the ground truth samples in red. We plot the FID of the distribution obtained by running annealed Langevin in blue. We plot the baseline $L^2$ distance and FID for samples obtained by the DPS algorithm using red and blue dashed lines.

1. Start with a sample $X_0 \sim p(x)$. Since $\eta_1$ is large, $p(x)$ is close to $p(x \mid y_1)$, so $X_0$ serves as an approximate sample $X_1 \sim \widehat{p}(x \mid y_1)$.

2. For $i = 1$ to $N - 1$: Run Langevin dynamics for a short time $T_i$, starting from the previous sample $X_i \sim \widehat{p}(x \mid y_i)$, targeting the next posterior $p(x \mid y_{i+1})$ using the score $\widehat{s}_{y_{i+1}}(x)$. Let the result be $X_{i+1} \sim \widehat{p}(x \mid y_{i+1})$.

3. The final sample $X_N \sim \widehat{p}(x \mid y_N)$ approximates a draw from the target posterior $p(x \mid y)$.

The core idea behind this annealing scheme is to actively control the process distribution $p(x_t)$, ensuring it remains on the manifold of the prior $p(x)$. By design, each mixing step $i \to i + 1$ connects two statistically close intermediate posteriors, $p(x \mid y_i)$ and $p(x \mid y_{i+1})$. This closeness guarantees that a short Langevin run $T_i$ can mix them, and this short duration prevents $p(x_t)$ from drifting significantly away from the step's starting distribution $\widehat{p}(x \mid y_i)$, and we can then argue that

$$\mathbb{E}_{y_i} \mathbb{E}_{x_t \sim \widehat{p}(x|y_i)} [\|s_{y_i}(x_t) - \widehat{s}_{y_i}(x_t)\|^2]] \approx \mathbb{E}_{y_i} \mathbb{E}_{x_t \sim p(x|y_i)} [\|s(x_t) - \widehat{s}(x_t)\|^2]] \leq \varepsilon_{\text{score}}^2.$$

This contrasts fundamentally with a single long Langevin run, where $x_t$ could venture far "off-manifold" into regions of poor score accuracy. By inserting frequent checkpoints that re-anchor the process, our annealing method substitutes such strong assumptions with structural control: the frequent "checkpoints" $p(x \mid y_i)$ ensure the process is repeatedly localized to regions where the $L^4$ accuracy suffices. While error is incurred in each step, maintaining proximity to the manifold keeps this error small. The overall approach hinges on demonstrating that these small, per-step errors accumulate controllably across all $N$ steps.

This strategy, however, requires rigorous analysis of three key technical challenges:

1. How to bound the required convergence time $T_i$ for the transition from $p(x \mid y_i)$ to $p(x \mid y_{i+1})$? In particular, what happens when $p$ only has local strong log-concavity?

2. How to bound the error incurred during a single mixing step of duration $T_i$, given the $L^4$ score error assumption on the prior score estimate?

3. How to ensure the total error accumulated across all $N$ mixing steps remains small?

Addressing these questions forms the core of our proof.

**Proof Organization.** In Section A, we show that for globally strongly log-concave distributions $p$, Langevin dynamics converges rapidly from $p(x \mid y_i)$ to $p(x \mid y_{i+1})$. We extend this convergence analysis to locally strongly log-concave distributions in Section B. In Section C, we provide bounds on the errors incurred by score errors and discretization in Langevin dynamics. In Section D, we show how to design the noise schedule to control the accumulated error of the full process. In Section E, we conclude the analysis for Algorithm 1, and apply it to establish the main theorems.

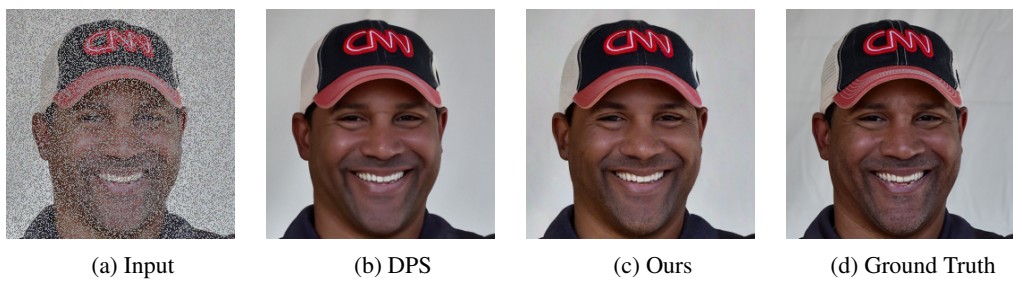

|                |               |              |                  |
| :------------: | :-----------: | :----------: | :--------------: |
| (a) Input      | (b) DPS       | (c) Ours     | (d) Ground Truth |

Figure 5: A set of samples for the inpainting task.

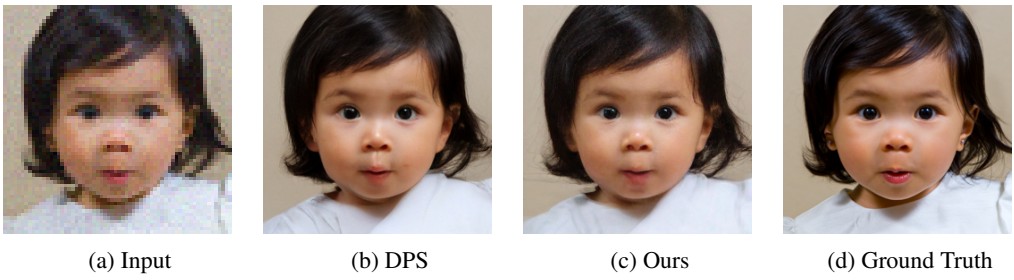

|                |               |              |                  |
| :------------: | :-----------: | :----------: | :--------------: |
| (a) Input      | (b) DPS       | (c) Ours     | (d) Ground Truth |

Figure 6: A set of samples for the super-resolution task.

## 4 Experiments

To validate our theoretical analysis and assess real-world performance, we study three inverse problems on FFHQ–256 [KLA21]: inpainting, $4\times$ super-resolution, and Gaussian deblurring. Experiments use 1k validation images and the pre-trained diffusion model from [CKM$^+$23]. Forward operators are specified as in [CKM$^+$23]: inpainting masks 30%–70% of pixels uniformly at random; super-resolution downsamples by a factor of 4; deblurring convolves the ground-truth with a Gaussian kernel of size $61 \times 61$ (std. 3.0). We first obtain initial reconstructions $x_0$ via Diffusion Posterior Sampling (DPS) [DS24], then refine them with our annealed Langevin sampler to draw samples close to $p(x \mid x_0, y)$. To control runtime, we sweep the step size while keeping the annealing schedule fixed.

For each step size, we report the per-image $L^2$ distance to the ground truth and the FID of the resulting sample distribution (Figure 4). Across all three tasks, increasing the time devoted to annealed Langevin decreases $L^2$ but increases FID; in the inpainting setting, when the step size is sufficiently small, our method surpasses DPS on both metrics. Qualitatively, our reconstructions better preserve ground-truth attributes compared to DPS (Figures 5 and 6). All experiments were run on a cluster with four NVIDIA A100 GPUs and required roughly two hours per task.

## Acknowledgments

This work is supported by the NSF AI Institute for Foundations of Machine Learning (IFML). ZX is supported by NSF Grant CCF-2312573 and a Simons Investigator Award (#409864, David Zuckerman).

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

# A  Langevin Convergence Between Strongly Log-concave Distributions

In this section, we study the following problem. Let $p$ be a probability distribution on $\mathbb{R}^d$, and let $A \in \mathbb{R}^{m \times d}$ be a matrix. For a sequence of parameters $\eta_i > \eta_{i+1}$ satisfying

$$\eta_i^2 = (1 + \gamma_i)\eta_{i+1}^2,$$

consider two random variables $y_i$ and $y_{i+1}$ defined as follows. First, draw $x \sim p$. Then, generate

$$y_{i+1} = Ax + N(0, \eta_{i+1}^2 I_m),$$

and further perturb it by

$$y_i = y_{i+1} + N(0, (\eta_i^2 - \eta_{i+1}^2)I_m).$$

Define the score function

$$s_{i+1}(x) = \nabla_x \log p(x \mid y_{i+1}).$$

We analyze the following SDE:

$$\mathrm{d}x_t = s_{i+1}(x_t)\,\mathrm{d}t + \sqrt{2}\,\mathrm{d}B_t, \quad x_0 \sim p(x \mid y_i). \tag{6}$$

This is the ideal (no discretization, no score estimation error) version of the process (2) that we actually run. Our goal is to establish the following lemma.

**Lemma A.1.** *Suppose the prior distribution $p(x)$ is $\alpha$-strongly log-concave. Then, running the process (6) for time*

$$T = O\left(\frac{m\gamma_i + \log(\lambda/\varepsilon)}{\alpha}\right)$$

*ensures that*

$$\Pr_{y_i, y_{i+1}}\left[\mathrm{TV}(x_T, p(x \mid y_{i+1})) \leq \varepsilon\right] \geq 1 - \frac{1}{\lambda}.$$

## A.1  $\chi^2$-divergence Between Distributions

In this section, our goal is to bound $\chi^2\left(p(x \mid y_i) \,\|\, p(x \mid y_{i+1})\right)$. Since the posterior distributions can be expressed as

$$p(x \mid y_i) = \frac{p(y_i \mid x)p(x)}{p(y_i)}, \quad p(x \mid y_{i+1}) = \frac{p(y_{i+1} \mid x)p(x)}{p(y_{i+1})}.$$

The $\chi^2$ divergence is

$$
\begin{aligned}
\chi^2\left(p(x \mid y_i) \,\|\, p(x \mid y_{i+1})\right) &= \mathop{\mathbb{E}}_{x \sim p(x|y_i)}\left[\frac{p(x \mid y_i)}{p(x \mid y_{i+1})}\right] - 1 \\
&= \mathop{\mathbb{E}}_{x \sim p(x|y_i)}\left[\frac{p(y_i \mid x)}{p(y_{i+1} \mid x)} \cdot \frac{p(y_{i+1})}{p(y_i)}\right] - 1 \\
&= \mathop{\mathbb{E}}_{x \sim p(x|y_i)}\left[\frac{p(y_i \mid x)}{p(y_{i+1} \mid x)}\right] \cdot \frac{p(y_{i+1})}{p(y_i)} - 1.
\end{aligned}
$$

We bound the term $\mathbb{E}_{x \sim p(x|y_i)}\left[\frac{p(y_i|x)}{p(y_{i+1}|x)}\right]$ first.

**Lemma A.2.** *We have*

$$\mathop{\mathbb{E}}_{x, y_i, y_{i+1}}\left[\frac{p(y_i \mid x)}{p(y_{i+1} \mid x)}\right] = 1$$

*Proof.* Let $Z_1 = y_{i+1} - Ax$, and let $Z_2 = y_i - Ax$. Then we have

$$\mathop{\mathbb{E}}_{x, y_i, y_{i+1}}\left[\frac{p(y_i \mid x)}{p(y_{i+1} \mid x)}\right] = \mathop{\mathbb{E}}_{Z_1, Z_2}\left[\frac{p(Z_2)}{p(Z_1)}\right] = \iint \frac{p_{Z_2}(z_2)}{p_{Z_1}(z_1)} \cdot p_{Z_1, Z_2}(z_1, z_2)\,\mathrm{d}z_1\,\mathrm{d}z_2.$$

Note that

$$p_{Z_1, Z_2}(z_1, z_2) = p_{Z_1}(z_1) \cdot f(z_2 - z_1),$$

where $f$ is the density function for $N(0, (\eta_i^2 - \eta_{i+1}^2)I_m)$. Therefore,

$$\iint \frac{p_{Z_2}(z_2)}{p_{Z_1}(z_1)} \cdot p_{Z_1, Z_2}(z_1, z_2) \, \mathrm{d}\, z_1 \, \mathrm{d}\, z_2 = \iint p_{Z_2}(z_2) \cdot f(z_2 - z_1) \, \mathrm{d}\, z_1 \, \mathrm{d}\, z_2$$

$$= \int p_{Z_2}(z_2) \left( \int f(z_2 - z_1) \, \mathrm{d}\, z_1 \right) \mathrm{d}\, z_2.$$

Since $f$ is a density function, its integral over $\mathbb{R}^m$ is 1. This gives that

$$\int p_{Z_2}(z_2) \left( \int f(z_2 - z_1) \, \mathrm{d}\, z_1 \right) \mathrm{d}\, z_2 = \int p_{Z_2}(z_2) \, \mathrm{d}\, z_2 = 1.$$

Hence,

$$\mathop{\mathbb{E}}_{x, y_i, y_{i+1}} \left[ \frac{p(y_i \mid x)}{p(y_{i+1} \mid x)} \right] = 1.$$

$\square$

**Corollary A.3.** *For any $\lambda > 1$, we have*

$$\mathop{\Pr}_{y_i, y_{i+1}} \left[ \mathop{\mathbb{E}}_{x \sim p(x \mid y_i)} \left[ \frac{p(y_i \mid x)}{p(y_{i+1} \mid x)} \right] \leq \lambda \right] \geq 1 - \frac{1}{\lambda}.$$

*Proof.* By Lemma A.2, we have

$$\mathop{\mathbb{E}}_{y_i, y_{i+1}} \left[ \mathop{\mathbb{E}}_{x \sim p(x \mid y_i)} \left[ \frac{p(y_i \mid x)}{p(y_{i+1} \mid x)} \right] \right] = \mathop{\mathbb{E}}_{x, y_i, y_{i+1}} \left[ \frac{p(y_i \mid x)}{p(y_{i+1} \mid x)} \right] = 1.$$

Applying Markov's inequality gives the result. $\square$

Now we bound $\frac{p(y_{i+1})}{p(y_i)}$. To make the lemma more self-contained, we abstract this a little bit.

**Lemma A.4.** *Let $\eta_1 > \eta_2$ be two positive numbers, and let $X \in \mathbb{R}^d$ be an arbitrary random variable. Define $Y_1 = X + Z_1$ and $Y_2 = Y_1 + Z_2$, where $Z_1 \sim N(0, \eta_1^2 I_d)$ and $Z_2 \sim N(0, \eta_2^2 I_d)$. Then,*

$$\mathop{\mathbb{E}}_{Y_1, Y_2 \mid \|Z_1\| \leq t} \left[ \frac{p(Y_1)}{p(Y_2)} \right] \leq \frac{1}{\Pr[\|Z_1\| \leq t]} \cdot \exp\left( O\left( \frac{d\eta_2^2}{\eta_1^2} + \frac{t^2 \eta_2^2}{\eta_1^4} \right) \right).$$

*where $p(Y_1)$ and $p(Y_2)$ are the densities of $Y_1$ and $Y_2$, respectively.*

*Proof.* First, we turn to bound

$$F_t(Y_1, Y_2) := \frac{p(Y_1)}{p(Y_2)} \cdot \Pr[\|Z_1\| \leq t \mid Y_1].$$

Note that

$$F_t(Y_1, Y_2) = \frac{\int_{\|s - Y_1\| \leq t} p(X = s)p(Y_1 \mid X = s) \, \mathrm{d}\, s}{\int_{\mathbb{R}^d} p(X = s)p(Y_2 \mid X = s) \, \mathrm{d}\, s} = \frac{\int_{\mathbb{R}^d} p_X(s)\phi_{\eta_1^2 I_d}(Y_1 - s) \cdot \mathbf{1}_{\{\|Y_1 - s\| \leq t\}} \, \mathrm{d}\, s}{\int_{\mathbb{R}^d} p_X(s)\phi_{(\eta_1^2 + \eta_2^2)I_d}(Y_2 - s) \, \mathrm{d}\, s}.$$

We have

$$F_t(Y_1, Y_2) \leq \sup_{s \in \mathbb{R}^d} \frac{\phi_{\eta_1^2 I_d}(Y_1 - s) \cdot \mathbf{1}_{\{\|Y_1 - s\| \leq t\}}}{\phi_{(\eta_1^2 + \eta_2^2)I_d}(Y_2 - s)} \leq \sup_{\|s - Y_1\| \leq t} \frac{\phi_{\eta_1^2 I_d}(Y_1 - s)}{\phi_{(\eta_1^2 + \eta_2^2)I_d}(Y_2 - s)}.$$

Write $Y_1 - s = e_1$, and note that $Y_2 - s = e_1 + Z_2$. Then define

$$G(e_1) = \frac{\phi_{\eta_1^2 I_d}(e_1)}{\phi_{(\eta_1^2 + \eta_2^2)I_d}(e_1 + Z_2)}, \qquad \|e_1\| \leq t.$$

This gives that for any $Y_1, Y_2$, and $t$,

$$F_t(Y_1, Y_2) \leq \sup_{\|e_1\| \leq t} G(e_1).$$

**Bounding $G(e_1)$** To bound $\sup_{\|e_1\|\leq t} G(e_1)$, we expand $\phi$ as the $d$-dimensional Gaussian probability density function:

$$G(e_1) = \left(\frac{\eta_1^2 + \eta_2^2}{\eta_1^2}\right)^{d/2} \exp\left(-\frac{\|e_1\|^2}{2\eta_1^2} + \frac{\|e_1 + Z_2\|^2}{2(\eta_1^2 + \eta_2^2)}\right).$$

Using the quadratic expansion $\|e_1 + Z_2\|^2 = \|e_1\|^2 + 2\langle e_1, Z_2\rangle + \|Z_2\|^2$, we rewrite:

$$G(e_1) = \left(\frac{\eta_1^2 + \eta_2^2}{\eta_1^2}\right)^{d/2} \exp\left(-\frac{\|e_1\|^2}{2\eta_1^2} + \frac{\|e_1\|^2 + 2\langle e_1, Z_2\rangle + \|Z_2\|^2}{2(\eta_1^2 + \eta_2^2)}\right).$$

Since $\|e_1\| \leq t$ and $\langle e_1, Z_2\rangle \leq \|e_1\|\|Z_2\|$, we bound

$$\frac{2\langle e_1, Z_2\rangle}{2(\eta_1^2 + \eta_2^2)} \leq \frac{t\|Z_2\|}{\eta_1^2 + \eta_2^2}.$$

Thus,

$$G(e_1) \leq \left(\frac{\eta_1^2 + \eta_2^2}{\eta_1^2}\right)^{d/2} \exp\left(\frac{\|Z_2\|^2}{2(\eta_1^2 + \eta_2^2)} + \frac{t\|Z_2\|}{\eta_1^2 + \eta_2^2}\right).$$

Therefore, for any $Y_1, Y_2$, and $t$, we have

$$F_t(Y_1, Y_2) \leq \left(\frac{\eta_1^2 + \eta_2^2}{\eta_1^2}\right)^{d/2} \exp\left(\frac{\|Z_2\|^2}{2(\eta_1^2 + \eta_2^2)} + \frac{t\|Z_2\|}{\eta_1^2 + \eta_2^2}\right).$$

This gives that

$$\mathop{\mathbb{E}}_{Y_1, Y_2 | \|Z_1\| \leq t} \left[\frac{p(Y_1)}{p(Y_2)}\right]$$

$$= \mathop{\mathbb{E}}_{Y_1, Y_2 | \|Z_1\| \leq t} \left[\frac{F_t(Y_1, Y_2)}{\Pr\left[\|Z_1\| \leq t \mid Y_1\right]}\right]$$

$$\leq \mathop{\mathbb{E}}_{Y_1, Y_2 | \|Z_1\| \leq t} \left[\frac{1}{\Pr\left[\|Z_1\| \leq t \mid Y_1\right]} \cdot \left(\frac{\eta_1^2 + \eta_2^2}{\eta_1^2}\right)^{d/2} \exp\left(\frac{\|Z_2\|^2}{2(\eta_1^2 + \eta_2^2)} + \frac{t\|Z_2\|}{\eta_1^2 + \eta_2^2}\right)\right]$$

$$= \left(\frac{\eta_1^2 + \eta_2^2}{\eta_1^2}\right)^{d/2} \mathop{\mathbb{E}}_{Y_1 | \|Z_1\| \leq t} \left[\frac{1}{\Pr\left[\|Z_1\| \leq t \mid Y_1\right]}\right] \cdot \mathop{\mathbb{E}}_{Z_2}\left[\exp\left(\frac{\|Z_2\|^2}{2(\eta_1^2 + \eta_2^2)} + \frac{t\|Z_2\|}{\eta_1^2 + \eta_2^2}\right)\right].$$

**Bounding expectation over $Z_2$.** We have

$$\mathop{\mathbb{E}}_{Z_2}\left[\exp\left(\frac{\|Z_2\|^2}{2(\eta_1^2 + \eta_2^2)} + \frac{t\|Z_2\|}{\eta_1^2 + \eta_2^2}\right)\right] = \mathop{\mathbb{E}}_{Z \sim \mathcal{N}(0, I_d)}\left[\exp\left(\frac{\eta_2^2\|Z\|^2}{2(\eta_1^2 + \eta_2^2)} + \frac{t\eta_2\|Z\|}{\eta_1^2 + \eta_2^2}\right)\right].$$

We can apply results on the Gaussian moment generating functions to bound this. Using Lemma A.10 by setting $\alpha = \frac{\eta_2^2}{2(\eta_1^2 + \eta_2^2)}$, $\beta = \frac{t\eta_2}{\eta_1^2 + \eta_2^2}$, and $\gamma = \frac{\eta_1^2}{4(\eta_1^2 + \eta_2^2)}$, we have

$$\mathop{\mathbb{E}}_{Z \sim \mathcal{N}(0, I_d)}\left[\exp\left(\frac{\eta_2^2\|Z\|^2}{2(\eta_1^2 + \eta_2^2)} + \frac{t\eta_2\|Z\|}{\eta_1^2 + \eta_2^2}\right)\right] \leq \exp\left(\frac{t^2\eta_2^2}{\eta_1^2(\eta_1^2 + \eta_2^2)}\right) \cdot \left(\frac{2(\eta_1^2 + \eta_2^2)}{\eta_1^2}\right)^{d/2}.$$

Finally, this gives

$$\mathop{\mathbb{E}}_{Y_1, Y_2 | \|Z_1\| \leq t}\left[\frac{p(Y_1)}{p(Y_2)}\right] \leq \frac{1}{\Pr\left[\|Z_1\| \leq t\right]} \cdot \exp\left(O\left(\frac{d\eta_2^2}{\eta_1^2} + \frac{t^2\eta_2^2}{\eta_1^4}\right)\right).$$

One need to verify that

$$\mathop{\mathbb{E}}_{Y_1 | \|Z_1\| \leq t}\left[\frac{1}{\Pr\left[\|Z_1\| \leq t \mid Y_1\right]}\right] \leq \frac{1}{\Pr\left[\|Z_1\| \leq t\right]}.$$

Also,

$$\mathop{\mathbb{E}}_{Z_2}[F_t(Y_1, Y_2)] \leq \exp\left(O\left(\frac{d\eta_2^2}{\eta_1^2} + \frac{t\eta_2\sqrt{d}}{\eta_1^2}\right)\right).$$

This gives the result. $\qquad\square$

**Lemma A.5.** *Let $\eta_1 > \eta_2$ be two positive numbers, and let $X \in \mathbb{R}^d$ be an arbitrary random variable. Define $Y_1 = X + Z_1$ and $Y_2 = Y_1 + Z_2$, where $Z_1 \sim N(0, \eta_1^2 I_d)$ and $Z_2 \sim N(0, \eta_2^2 I_d)$. There exists a constant $C > 0$ such that for any $\lambda > 1$,*

$$\Pr_{Y_1, Y_2}\left[\frac{p(Y_1)}{p(Y_2)} \leq \exp\left(C \cdot \left(\frac{d\eta_2^2}{\eta_1^2} + \ln \lambda\right)\right)\right] \geq 1 - \frac{1}{\lambda}.$$

*where $p(Y_1)$ and $p(Y_2)$ are the densities of $Y_1$ and $Y_2$, respectively.*

*Proof.* Let $t = (\sqrt{d} + \sqrt{2\ln(2\lambda)})\eta_1$. By applying Laurent-Massart bounds (Lemma A.11), we have

$$\Pr\left[\|Z_1\| \leq t\right] \geq 1 - \frac{1}{2\lambda}.$$

Taking these into Lemma A.4, we have

$$\mathbb{E}_{Y_1, Y_2 | \|Z_1\| \leq t}\left[\frac{p(Y_1)}{p(Y_2)}\right] \leq \exp\left(O\left(\frac{d\eta_2^2}{\eta_1^2} + \frac{t^2\eta_2^2}{\eta_1^4}\right)\right) \leq \exp\left(O\left(\frac{(d + \ln \lambda)\eta_2^2}{\eta_1^2}\right)\right).$$

By applying Markov's inequality, for a large enough constant $C > 0$, we have

$$\Pr_{Y_1, Y_2 | \|Z_1\| \leq t}\left[\frac{p(Y_1)}{p(Y_2)} \leq \lambda \exp\left(C \cdot \left(\frac{(d + \ln \lambda)\eta_2^2}{\eta_1^2}\right)\right)\right] \geq 1 - \frac{1}{2\lambda}.$$

Cleaning up the bound a little bit, this implies that for a large enough constant $C > 0$,

$$\Pr_{Y_1, Y_2 | \|Z_1\| \leq t}\left[\frac{p(Y_1)}{p(Y_2)} \leq \exp\left(C \cdot \left(\frac{d\eta_2^2}{\eta_1^2} + \ln \lambda\right)\right)\right] \geq 1 - \frac{1}{2\lambda}.$$

Combining this with the probability that $\|Z\| \leq t$, a union bound gives that

$$\Pr_{Y_1, Y_2}\left[\frac{p(Y_1)}{p(Y_2)} \leq \exp\left(C \cdot \left(\frac{d\eta_2^2}{\eta_1^2} + \ln \lambda\right)\right)\right] \geq 1 - \frac{1}{\lambda}.$$

$\square$

The $\chi^2$ divergence is

$$
\begin{aligned}
\chi^2\left(p(x \mid y_i) \,\|\, p(x \mid y_{i+1})\right) &= \mathbb{E}_{x \sim p(x|y_i)}\left[\frac{p(x \mid y_i)}{p(x \mid y_{i+1})}\right] - 1 \\
&= \mathbb{E}_{x \sim p(x|y_i)}\left[\frac{p(y_i \mid x)}{p(y_{i+1} \mid x)} \cdot \frac{p(y_{i+1})}{p(y_i)}\right] - 1 \\
&= \mathbb{E}_{x \sim p(x|y_i)}\left[\frac{p(y_i \mid x)}{p(y_{i+1} \mid x)}\right] \cdot \frac{p(y_{i+1})}{p(y_i)} - 1.
\end{aligned}
$$

Now we can bound the $\chi^2$-diversity.

**Lemma A.6.** *There exists a constant $C > 0$ such that for any $\lambda > 1$,*

$$\Pr_{y_i, y_{i+1}}\left[\chi^2\left(p(x \mid y_i) \,\|\, p(x \mid y_{i+1})\right) \leq \exp\left(C\left(\frac{m(\eta_i^2 - \eta_{i+1}^2)}{\eta_{i+1}^2} + \ln \lambda\right)\right)\right] \geq 1 - \frac{1}{\lambda}.$$

*Proof.* Note that

$$\chi^2\left(p(x \mid y_i) \,\|\, p(x \mid y_{i+1})\right) = \mathbb{E}_{x \sim p(x|y_i)}\left[\frac{p(y_i \mid x)}{p(y_{i+1} \mid x)}\right] \cdot \frac{p(y_{i+1})}{p(y_i)} - 1.$$

By Corollary A.3, we have

$$\Pr_{y_i, y_{i+1}}\left[\mathbb{E}_{x \sim p(x|y_i)}\left[\frac{p(y_i \mid x)}{p(y_{i+1} \mid x)}\right] \leq 2\lambda\right] \geq 1 - \frac{1}{2\lambda}.$$

By Lemma A.5, there exists a constant $C > 0$ such that

$$\Pr_{Y_1, Y_2}\left[\frac{p(Y_1)}{p(Y_2)} \leq \exp\left(C\left(\frac{m(\eta_i^2 - \eta_{i+1}^2)}{\eta_{i+1}^2} + \ln \lambda\right)\right)\right] \geq 1 - \frac{1}{2\lambda}.$$

A union bound over these two implies that with probability of $1 - 1/\lambda$,

$$\frac{p(y_i \mid x)}{p(y_{i+1} \mid x)} \cdot \frac{p(y_{i+1})}{p(y_i)} - 1 \leq 2\lambda \cdot \exp\left(C\left(\frac{m(\eta_i^2 - \eta_{i+1}^2)}{\eta_{i+1}^2} + \ln \lambda\right)\right) \leq \exp\left(C'\left(\frac{m(\eta_i^2 - \eta_{i+1}^2)}{\eta_{i+1}^2} + \ln \lambda\right)\right),$$

where $C'$ is a positive constant. This concludes the lemma. $\square$

### A.2 Convergence time of Langevin dynamics

We present the following result on the convergence of Langevin dynamics:

**Lemma A.7** ([Dal17])**.** *Let $p$ and $q$ be probability distributions such that $q$ is an $\alpha$-strong log-concave distribution. Consider the Langevin dynamics initialized with $p$ as the starting distribution. Then, for any $t \geq 0$, we have*

$$\mathrm{TV}(p_t, q) \leq \frac{1}{2}\chi^2(p \,\|\, q)^{1/2}e^{-t\alpha/2}.$$

This implies that

**Lemma A.8.** *Let $p$ and $q$ be probability distributions such that $q$ is an $\alpha$-strong log-concave distribution. Consider the Langevin dynamics initialized with $p$ as the starting distribution. By running the diffusion for time*

$$T = O\left(\frac{\log(1/\varepsilon) + \log\chi^2(p\|q)}{\alpha}\right),$$

*we have $\mathrm{TV}(p_T, q) \leq \varepsilon$.*

Now we show that the posterior distribution is even more strongly log-concave than prior distribution.

**Lemma A.9.** *Suppose that $p(x)$ is $\alpha$-strongly log-concave. Then, the posterior density*

$$p(x \mid Ax + N(\eta_i^2 I_m) = y_i)$$

*is $\alpha$-strongly log-concave.*

*Proof.* By Bayes' rule, the posterior density can be written (up to normalization) as

$$p(x \mid Ax + N(\eta_i^2 I_m) = y_i) \;\propto\; p(x)\exp\left(-\frac{1}{2\eta_i^2}\|Ax - y_i\|_2^2\right).$$

Define the negative log–posterior

$$\varphi(x) := -\log p(x) + \frac{1}{2\eta_i^2}\|Ax - y_i\|_2^2.$$

Since $p$ is $\alpha$-strongly log-concave, its negative log–density satisfies

$$\nabla^2(-\log p(x)) \;\succeq\; \alpha I.$$

Moreover, the Gaussian likelihood term has

$$\nabla^2\left(\frac{1}{2\eta_i^2}\|Ax - y_i\|_2^2\right) = \frac{1}{\eta_i^2}A^T A \;\succeq\; 0.$$

By the sum rule for Hessians,

$$\nabla^2\varphi(x) = \nabla^2(-\log p(x)) \;+\; \frac{1}{\eta_i^2}A^T A \;\succeq\; \alpha I.$$

Hence $\varphi$ is $\alpha$-strongly convex, and the posterior density $p(x \mid Ax + N(\eta_i^2 I_m) = y_i) \propto e^{-\varphi(x)}$ is $\alpha$-strongly log-concave. □

Now we are ready to prove Lemma A.1:

*Proof of Lemma A.1.* By Lemma A.9, $p(x \mid y_{i+1})$ is $alpha$-strongly log-concave. This allows us to apply Lemma A.8. Therefore, to achieve $\varepsilon$ TV error in convergence, we only need to run the process for

$$T = O\left(\frac{\log(1/\varepsilon) + \log\chi^2(p(x \mid y_i) \,\|\, p(x \mid y_{i+1}))}{\alpha}\right).$$

Taking in the result in Lemma A.6, we have with $1 - \frac{1}{\lambda}$ probability over $y_i$ and $y_{i+1}$, we only need

$$T = O\left(\frac{m\gamma_i + \log(\lambda/\varepsilon)}{\alpha}\right).$$

□

### A.3 Utility Lemmas.

**Lemma A.10.** *Let $Z \sim \mathcal{N}(0, I_d)$ be a $d$-dimensional standard Gaussian random vector, and let $\alpha, \beta \in \mathbb{R}$. For any $\gamma > 0$ satisfying $\alpha + \gamma < \frac{1}{2}$, we have*

$$\mathbb{E}\Big[\exp\Big(\alpha\|Z\|^2 + \beta\|Z\|\Big)\Big] \leq \exp\Big(\frac{\beta^2}{4\gamma}\Big)(1 - 2(\alpha + \gamma))^{-d/2}.$$

*Proof.* For all $r \geq 0$ and any $\gamma > 0$, it is easy to check that by AM-GM inequality,

$$\beta\, r \leq \gamma\, r^2 + \frac{\beta^2}{4\gamma}.$$

Taking $r = \|Z\|$ and exponentiating both sides, we obtain

$$\exp\Big(\beta\|Z\|\Big) \leq \exp\Big(\gamma\|Z\|^2 + \frac{\beta^2}{4\gamma}\Big).$$

Multiplying both sides by $\exp\Big(\alpha\|Z\|^2\Big)$ yields

$$\exp\Big(\alpha\|Z\|^2 + \beta\|Z\|\Big) \leq \exp\Big(\frac{\beta^2}{4\gamma}\Big)\exp\Big((\alpha + \gamma)\|Z\|^2\Big).$$

This gives that

$$\mathbb{E}\Big[\exp\Big(\alpha\|Z\|^2 + \beta\|Z\|\Big)\Big] \leq \exp\Big(\frac{\beta^2}{4\gamma}\Big)\mathbb{E}\Big[\exp\Big((\alpha + \gamma)\|Z\|^2\Big)\Big].$$

For $Z \sim \mathcal{N}(0, I_d)$, when $\alpha + \gamma < \frac{1}{2}$ we have

$$\mathbb{E}\Big[\exp\Big((\alpha + \gamma)\|Z\|^2\Big)\Big] = (1 - 2(\alpha + \gamma))^{-d/2},$$

Hence,

$$\mathbb{E}\Big[\exp\Big(\alpha\|Z\|^2 + \beta\|Z\|\Big)\Big] \leq \exp\Big(\frac{\beta^2}{4\gamma}\Big)(1 - 2(\alpha + \gamma))^{-d/2}.$$

$\square$

**Lemma A.11** (Laurent-Massart Bounds[LM00])**.** *Let $v \sim \mathcal{N}(0, I_m)$. For any $t > 0$,*

$$\Pr[\|v\|^2 - m \geq 2\sqrt{mt} + 2t] \leq e^{-t},$$
$$\Pr[\|v\|^2 - m \leq -2\sqrt{mt}] \leq e^{-t}.$$

## B  Convergence Between Locally Well-Conditioned Distributions

In the last section, we considered the convergence time between two posterior distributions of a globally strongly log-concave distribution. In this section, we will relax the assumption of global strong log-concavity and consider the convergence time between two distributions that are locally "well-behaved". We give the following formal definition:

**Definition B.1.** *For $\delta \in [0, 1)$ and $R, \widetilde{L}, \alpha \in (0, +\infty]$, we say that a distribution $p$ is $(\delta, r, R, \widetilde{L}, \alpha)$ mode-centered locally well-conditioned if there exists $\theta$ such that*

- $\nabla \log p(\theta) = 0$.

- $\Pr_{x \sim p}[x \in B(\theta, r)] \geq 1 - \delta$.

- *For $x, y \in B(\theta, R)$, we have that $\|s(x) - s(y)\| \leq \widetilde{L}\alpha\|x - y\|$.*

- *For $x, y \in B(\theta, R)$, we have that $\langle s(y) - s(x), x - y \rangle \geq \alpha\|x - y\|^2$.*

Again, we consider the following process $P$, which is identical to process (6) we considered in the last section:

$$dx_t = \left(s(x_t) + \frac{A^T y_{i+1} - A^T A x_t}{\eta_{i+1}^2}\right) dt + \sqrt{2} dB_t, \quad x_0 \sim p(x \mid y_i)$$

Our goal is to prove the following lemma:

**Lemma B.2.** *Suppose $p$ is a $(\delta, r, R, \widetilde{L}, \alpha)$ mode-centered locally well-conditioned distribution. Let $C > 0$ be a large enough constant. We consider the process $P$ running for time*

$$T \geq C\left(\frac{m\gamma_i + \log(\lambda/\varepsilon)}{\alpha}\right).$$

*Suppose that*

$$R \geq r + \frac{T\|A\|}{\eta_{i+1}^2}\left(\|A\|r + \eta_{i+1}(\sqrt{m} + \sqrt{2\ln(1/\delta)})\right) + 2\sqrt{dT \ln(2d/\delta)}.$$

*Then $x_T \sim P_T$ satisfies that*

$$\Pr_{y_i, y_{i+1}}\left[\mathrm{TV}(x_T, p(x \mid y_{i+1})) \leq \varepsilon + \lambda\delta\right] \geq 1 - O(\lambda^{-1}).$$

In this section, we will assume that $p$ is $(\delta, r, R, \widetilde{L}, \alpha)$ mode-centered locally well-conditioned. Without loss of generality, we assume that the mode of $p$ is at 0, i.e., $\theta = 0$.

## B.1 High Probability Boundedness of Langevin Dynamics

We consider the process $P'$ defined as the process $P$ conditioned on $x_t \in B(0, R)$ for $t \in [0, T]$.

Our goal is to prove the following lemma:

**Lemma B.3.** *Suppose the following holds:*

$$R \geq r + \frac{T\|A\|}{\eta_{i+1}^2}\left(\|A\|r + \eta_{i+1}(\sqrt{m} + \sqrt{2\ln(1/\delta)})\right) + 2\sqrt{dT \ln(2d/\delta)}.$$

*We have that*

$$\mathbb{E}\left[\mathrm{TV}(P, P')\right] \lesssim \delta.$$

We start by decomposing the total variation distance between $P$ and $P'$ as follows:

**Lemma B.4.** *We have that*

$$\mathbb{E}\left[\mathrm{TV}(P, P')\right] \leq \mathbb{E}_{y_i, y_{i+1}}\left[\Pr_P\left[\exists t \in [0, T] : \|x_t\| \geq R \,\middle|\, x_0 \in B(0, r)\right]\right] + \delta.$$

*Proof.* Recall that the process $P'$ is defined as the law of $P$ conditioned on the event

$$\mathcal{F} := \{x_t \in B(0, R) \text{ for all } t \in [0, T]\}.$$

Thus, for any fixed $y_i$ we have

$$\mathrm{TV}(P, P') = \mathrm{TV}\left(P, \, P(\cdot \mid \mathcal{F})\right) = 1 - P(\mathcal{F}) = P(\mathcal{F}^c),$$

where $\mathcal{F}^c = \{\exists t \in [0, T] : \|x_t\| \geq R\}$.

Let $\mathcal{E} := \{x_0 \in B(0, r)\}$ denote the event that the initial condition is "good." Then, by the law of total probability,

$$P(\mathcal{F}^c) = P(\mathcal{F}^c \cap \mathcal{E}) + P(\mathcal{F}^c \cap \mathcal{E}^c) \leq P(\mathcal{F}^c \mid \mathcal{E}) + P(\mathcal{E}^c).$$

Taking the expectation with respect to $y_i$ and $y_{i+1}$, we obtain

$$\mathbb{E}\left[\mathrm{TV}(P, P')\right] \leq \mathbb{E}\left[P(\mathcal{F}^c \mid \mathcal{E})\right] + \mathbb{E}\left[P(\mathcal{E}^c)\right].$$

Since
$$P(\mathcal{F}^c \mid \mathcal{E}) = \Pr_P\Big[\exists\, t \in [0,T] : \|x_t\| \ge R \,\Big|\, x_0 \in B(0,r)\Big],$$

and by the law of total probability, we have
$$\mathbb{E}\Big[P(\mathcal{E}^c)\Big] = \Pr_{x \sim p}(\|x\| \ge r) \le \delta,$$

it follows that
$$\mathbb{E}\Big[\mathrm{TV}(P, P')\Big] \le \mathbb{E}\Big[\Pr_P\Big[\exists\, t \in [0,T] : \|x_t\| \ge R \,\Big|\, x_0 \in B(0,r)\Big]\Big] + \delta.$$

This completes the proof. $\qquad\square$

Now we focus on bounding $\mathbb{E}_{y_i, y_{i+1}}\Big[\Pr_P\Big[\exists\, t \in [0,T] : \|x_t\| \ge R \,\Big|\, x_0 \in B(0,r)\Big]\Big]$. We start by observing the following lemma for log-concave distributions.

**Lemma B.5.** *Let $p$ be a log-concave distribution such that $p$ is continuously differentiable. Suppose the mode of $p$ is at 0. Then, for all $x \in \mathbb{R}^d$,*
$$\langle \nabla \log p(x), x \rangle \le 0.$$

*Proof.* Since $\log p$ is concave, for any $x, \theta \in \mathbb{R}^d$ the first-order condition for concavity yields
$$\log p(\theta) \le \log p(x) + \langle \nabla \log p(x), -x \rangle.$$

Rearrange this inequality to obtain
$$\langle \nabla \log p(x), -x \rangle \ge \log p(\theta) - \log p(x).$$

Because $\theta$ is a mode, $\log p(\theta) \ge \log p(x)$ for every $x \in \mathbb{R}^d$; hence,
$$\langle \nabla \log p(x), x \rangle \le 0.$$

$\qquad\square$

**Lemma B.6.** *Let $x_t$ be the stochastic process*
$$dx_t = (f(x_t) + g(x_t))\, dt + \sqrt{2}\, dB_t, \quad x_0 \in \mathbb{R}^d,$$

*where $B_t$ is a standard $\mathbb{R}^d$-valued Brownian motion and the functions $f, g : \mathbb{R}^d \to \mathbb{R}^d$ satisfy*
$$\|f(x)\| \le a \quad \text{and} \quad \langle g(x), x \rangle \le 0 \quad \text{for all } x \in \mathbb{R}^d,$$

*with $a \ge 0$. Then, for any time horizon $T > 0$ and $\delta \in (0,1)$,*
$$\Pr\left[\sup_{t \in [0,T]} \|x_t\| \le \|x_0\| + aT + 2\sqrt{T\, d\, \ln\Big(\frac{2d}{\delta}\Big)}\right] \ge 1 - \delta.$$

*Proof.* Define $r(t) = \|x_t\|$. Although the Euclidean norm is not smooth at the origin, an application of Itô's formula yields that, for $x_t \ne 0$, one has
$$dr(t) = \frac{\langle x_t,\, f(x_t) + g(x_t)\rangle}{\|x_t\|}\, dt + \sqrt{2}\, \langle u(t), dB_t\rangle + \frac{d-1}{\|x_t\|}\, dt,$$

where $u(t) = x_t / \|x_t\|$. Using the bound $\|f(x_t)\| \le a$ and the hypothesis $\langle g(x_t), x_t \rangle \le 0$, it follows by the Cauchy–Schwarz inequality that
$$\frac{\langle x_t,\, f(x_t)\rangle}{\|x_t\|} \le a \quad \text{and} \quad \frac{\langle x_t,\, g(x_t)\rangle}{\|x_t\|} \le 0.$$

Discarding the nonnegative Itô correction term $\frac{d-1}{\|x_t\|}\, dt$ (which can only increase the process), we deduce that
$$dr(t) \le a\, dt + \sqrt{2}\, \langle u(t), dB_t\rangle.$$

Introduce the one-dimensional process

$$y(t) = \|x_0\| + at + \sqrt{2}\,\beta(t), \quad \text{with} \quad \beta(t) = \int_0^t \langle u(s), dB_s \rangle.$$

Since $\|u(s)\| = 1$ for all $s$, the process $\beta(t)$ is a standard one-dimensional Brownian motion with quadratic variation $\langle \beta \rangle_t = t$. By a standard comparison theorem for one-dimensional stochastic differential equations, it follows that $r(t) \leq y(t)$ almost surely for all $t \geq 0$; hence,

$$\sup_{t \in [0,T]} \|x_t\| \leq \|x_0\| + aT + \sqrt{2} \sup_{t \in [0,T]} \beta(t).$$

A classical application of the reflection principle for one-dimensional Brownian motion shows that, for any $\rho > 0$,

$$\Pr\left[ \sup_{t \in [0,T]} \beta(t) \geq \rho \right] = 2\Pr(\beta(T) \geq \rho) \leq 2\exp\left(-\frac{\rho^2}{2T}\right).$$

To incorporate the $d$-dimensional nature of the noise, one may use a union bound over the $d$ coordinate processes of $B_t$, which yields that

$$\Pr\left[ \sqrt{2} \sup_{t \in [0,T]} \beta(t) \leq 2\sqrt{T\,d\,\ln\left(\frac{2d}{\delta}\right)} \right] \geq 1 - \delta.$$

Combining the foregoing estimates, we deduce that

$$\Pr\left[ \sup_{t \in [0,T]} \|x_t\| \leq \|x_0\| + aT + 2\sqrt{T\,d\,\ln\left(\frac{2d}{\delta}\right)} \right] \geq 1 - \delta,$$

which is the desired result. $\qquad\square$

**Lemma B.7.** *For any $\delta \in (0,1)$ and $T > 0$, it holds that*

$$\Pr_{x_t \sim P_t}\left[ \sup_{t \in [0,T]} \|x_t\| \geq r + T \cdot \frac{\|A^T y_{i+1}\|}{\eta_{i+1}^2} + 2\sqrt{T\,d\,\ln\left(\frac{2d}{\delta}\right)} \,\Big|\, x_0 \in B(0,r) \right] < \delta.$$

*Proof.* We first note that by Lemma B.5, for any $x \in \mathbb{R}^d$, we have

$$\left\langle s(x) - \frac{A^T A x}{\eta_{i+1}^2}, x \right\rangle \leq \langle s(x), x \rangle - \frac{1}{\eta_{i+1}^2}\|Ax\|^2 \leq 0.$$

By Lemma B.6, we have that

$$\Pr_{x_t \sim P}\left[ \sup_{t \in [0,T]} \|x_t\| \geq \|x_0\| + T \cdot \frac{\|A^T y_{i+1}\|}{\eta_{i+1}^2} + 2\sqrt{T\,d\,\ln\left(\frac{2d}{\delta}\right)} \right] < \delta,$$

This gives that

$$\Pr_{x_t \sim P_t}\left[ \sup_{t \in [0,T]} \|x_t\| \geq r + T \cdot \frac{\|A^T y_{i+1}\|}{\eta_{i+1}^2} + 2\sqrt{T\,d\,\ln\left(\frac{2d}{\delta}\right)} \,\Big|\, x_0 \in B(0,r) \right] < \delta.$$

$\qquad\square$

**Lemma B.8.** *For any $\delta \in (0,1)$, suppose*

$$R \geq r + \frac{T\|A\|}{\eta_{i+1}^2}\left( \|A\|r + \eta_{i+1}(\sqrt{m} + \sqrt{2\ln(1/\delta)}) \right) + 2\sqrt{dT\ln(2d/\delta)}.$$

*It holds that*

$$\mathbb{E}_{y_i, y_{i+1}}\left[ \Pr_{x_t \sim P}\left[ \sup_{t \in [0,T]} \|x_t\| \geq R \,\Big|\, x_0 \in B(0,r) \right] \right] \lesssim \delta.$$

*Proof.* Recall that
$$y_{i+1} = Ax + \eta_{i+1}z, \quad z \sim \mathcal{N}(0, I_m).$$
With probability at least $1 - \delta$
$$\|z\| \le \sqrt{m} + \sqrt{2\ln(1/\delta)}.$$
Since $\|x\| \le r$ with probability $1 - \delta$. Thus, with probability at least $1 - 2\delta$, it follows that
$$\|y_{i+1}\| \le \|Ax\| + \eta_{i+1}\|z\| \le \|A\|r + \eta_{i+1}\left(\sqrt{m} + \sqrt{2\ln(1/\delta)}\right).$$
Hence, with the $1 - 2\delta$ probability,
$$T \cdot \frac{\|A^T y_{i+1}\|}{\eta_{i+1}^2} \le \frac{T\|A\|\|y_{i+1}\|}{\eta_{i+1}^2} \le \frac{T\|A\|}{\eta_{i+1}^2}\left(\|A\|r + \eta_{i+1}(\sqrt{m} + \sqrt{2\ln(1/\delta)})\right).$$
Therefore, ensuring that
$$R \ge r + T \cdot \frac{\|A^T y_{i+1}\|}{\eta_{i+1}^2} + 2\sqrt{T\,d\,\ln\left(\frac{2d}{\delta}\right)}.$$
In this case, [Lemma B.7](#) guarantees that
$$\Pr_{x_t \sim P}\left[\sup_{t \in [0,T]} \|x_t\| \ge R \,\Big|\, x_0 \in B(0,r)\right] \lesssim \delta.$$
Since the probability satisfying the condition is at least $1 - 2\delta$, we have
$$\mathop{\mathbb{E}}_{y_i, y_{i+1}}\left[\Pr_{x_t \sim P}\left[\sup_{t \in [0,T]} \|x_t\| \ge R \,\Big|\, x_0 \in B(0,r)\right]\right] \lesssim \delta.$$
$\square$

Putting [Lemma B.4](#) and [Lemma B.8](#) together, we directly obtain [Lemma B.3](#).

## B.2 Concentration of Strongly Log-Concave Distributions

Before moving futher, we first prove that a strongly log-concave distribution is highly concentrated.

**Lemma B.9** (Norm Bound for $\alpha$-Strongly Logconcave Distributions). *Let $X$ be a random vector in $\mathbb{R}^d$ with density*
$$\pi(x) \propto \exp(-V(x)),$$
*where the potential $V : \mathbb{R}^d \to \mathbb{R}$ is $\alpha$-strongly convex; that is,*
$$\nabla^2 V(x) \succeq \alpha I \quad \text{for all } x \in \mathbb{R}^d.$$
*Denote by $\mu = \mathbb{E}[X]$ the mean of $X$. Then, for any $\delta \in (0,1)$, with probability at least $1 - \delta$ we have*
$$\|X - \mu\| \le \sqrt{\frac{d}{\alpha}} + \sqrt{\frac{2\ln(1/\delta)}{\alpha}}.$$

*Proof.* Since $V$ is $\alpha$-strongly convex, the density $\pi$ satisfies a logarithmic Sobolev inequality with constant $1/\alpha$. Consequently, for any 1-Lipschitz function $f : \mathbb{R}^d \to \mathbb{R}$ and any $t > 0$, one has the concentration inequality (via Herbst's argument)
$$\mathbb{P}\left(f(X) - \mathbb{E}[f(X)] \ge t\right) \le \exp\left(-\frac{\alpha t^2}{2}\right).$$
Noting that the function
$$f(x) = \|x - \mu\|$$
is 1-Lipschitz (by the triangle inequality), it follows that
$$\mathbb{P}\left(\|X - \mu\| - \mathbb{E}\|X - \mu\| \ge t\right) \le \exp\left(-\frac{\alpha t^2}{2}\right).$$

A standard calculation using the fact that the covariance matrix of $X$ satisfies $\mathrm{Cov}(X) \preceq \frac{1}{\alpha} I$ gives

$$\mathbb{E}\|X - \mu\| \leq \sqrt{\frac{d}{\alpha}}.$$

Thus, setting

$$t = \sqrt{\frac{2\ln(1/\delta)}{\alpha}},$$

we obtain

$$\mathbb{P}\left(\|X - \mu\| \geq \sqrt{\frac{d}{\alpha}} + \sqrt{\frac{2\ln(1/\delta)}{\alpha}}\right) \leq \delta.$$

This completes the proof. $\square$

**Lemma B.10** ([JCP24])**.** *Let $\mu$ and $\theta$ denote the mean and the mode of distribution $p$, respectively, where $p$ is $\alpha$-strongly log-concave and univariate. Then, $|\mu - \theta| \leq \frac{1}{\sqrt{\alpha}}$.*

This immediately gives us the following corollary.

**Corollary B.11.** *Let $p$ be a $\alpha$–strongly log-concave distribution on $\mathbb{R}^d$. Let $\theta$ be the mode of $p$. For every $0 < \delta < 1$, we have*

$$\Pr_{X \sim p}\left[\|X - \theta\| \leq 2\sqrt{\frac{d}{\alpha}} + \sqrt{\frac{2\log(1/\delta)}{\alpha}}\right] \geq 1 - \delta.$$

This also implies that every $\alpha$-strongly log-concave distribution is mode-centered locally well-conditioned.

**Lemma B.12.** *Let $p$ be an $\alpha$-strongly log-concave distribution. Suppose the score function of $p$ is $L$-Lipschitz. Then, for any $0 < \delta < 1$, we have that $p$ is $(\delta, 2\sqrt{\frac{d}{\alpha}} + \sqrt{\frac{2\log(1/\delta)}{\alpha}}, \infty, L/\alpha, \alpha)$ mode-centered locally well-conditioned.*

### B.3 Convergence to Target Distribution

Since $p$ is not globally strongly log-concave, we need to extend the distribution $p$ to a globally strongly log-concave distribution. We will use the following lemma to extend the distribution.

**Lemma B.13.** *Suppose $g : B(0, R) \to \mathbb{R}$ is continuously differentiable with gradient $s := \nabla g \in C(B(0, R); \mathbb{R}^d)$ and satisfies*

$$\langle s(y) - s(x), x - y\rangle \geq \alpha \|x - y\|^2, \qquad \forall\, x, y \in B(0, R). \tag{7}$$

*For every $z \in B(0, R)$ define*

$$\varphi_z(x) = g(z) + \langle s(z), x - z\rangle - \frac{\alpha}{2}\|x - z\|^2, \qquad x \in \mathbb{R}^d,$$

*and set*

$$\tilde{g}(x) = \begin{cases} g(x), & \|x\| \leq R, \\ \inf_{z \in B(0,R)} \varphi_z(x), & \|x\| > R. \end{cases} \tag{8}$$

*Then the density $\widetilde{p}(x) \propto e^{\tilde{g}(x)}$ is globally $\alpha$–strongly log–concave.*

*Proof.* For each fixed $z \in B(0, R)$ the mapping $\varphi_z$ has Hessian $-\alpha I_d$, hence is $\alpha$–strongly concave on the whole space. Because of (7) we have

$$g(x) \leq g(z) + \langle s(z), x - z\rangle - \frac{\alpha}{2}\|x - z\|^2 = \varphi_z(x), \qquad \forall\, x, z \in B(0, R),$$

with equality when $x = z$. Consequently $\tilde{g}$ defined in (8) agrees with $g$ on $B(0, R)$.

Fix $x \in \mathbb{R}^d$ and choose $z_x \in B(0, R)$ attaining the infimum in (8). Because $\varphi_{z_x}$ touches $\tilde{g}$ from above at $x$, the vector

$$\xi = \nabla\varphi_{z_x}(x) = s(z_x) - \alpha(x - z_x)$$

belongs to $\partial\widetilde{g}(x)$. By $\alpha$–strong concavity of $\varphi_{z_x}$,

$$\varphi_{z_x}(y) \leq \varphi_{z_x}(x) + \langle \xi,\, y - x \rangle - \frac{\alpha}{2}\,\|y - x\|^2, \qquad \forall\, y \in \mathbb{R}^d.$$

Taking the infimum over $z$ on the left and using $\widetilde{g}(x) = \varphi_{z_x}(x)$ gives that

$$\widetilde{g}(y) \leq \widetilde{g}(x) + \langle \xi,\, y - x \rangle - \frac{\alpha}{2}\,\|y - x\|^2, \qquad \forall\, x, y \in \mathbb{R}^d;$$

hence $\widetilde{g}$ is globally $\alpha$–strongly concave, and therefore $\widetilde{p}$ is $\alpha$–strongly log-concave. $\qquad\square$

**Lemma B.14.** *Let $p$ be a $d$-dimensional $(\delta, r, R, \widetilde{L}, \alpha)$ mode-centered locally well-conditioned probability distribution with $0 < \delta \leq 1/2$ and $\alpha > 0$. Assume*

$$R \geq 2\sqrt{\frac{d}{\alpha}} + \sqrt{\frac{2\log(1/\delta)}{\alpha}}.$$

*Then there exists an $\alpha$-strongly log-concave distribution $\widetilde{p}$ on $\mathbb{R}^d$ such that*

$$\mathrm{TV}(p, \widetilde{p}) \leq 3\delta.$$

*Proof.* Let $\theta$ be the point in [Definition B.1](#) and without loss of generality, we assume $\theta = 0$. Write $B := B(0, R)$ and $B^c := \mathbb{R}^d \setminus B$. By definition $p(B^c) \leq \delta$.

Set $g := \log p$, and let $\widetilde{g}$ be the function in [Lemma B.13](#). Then, $\rho(x) := e^{\widetilde{g}(x)}$ is $\alpha$-strongly log-concave and $\rho = p$ on $B$. Let $Z := \int_{\mathbb{R}^d} \rho$ and define $\widetilde{p} := \rho/Z$.

Now we bound

$$\mathrm{TV}(p, \widetilde{p}) = \frac{1}{2}\int_B |p - \widetilde{p}| + \frac{1}{2}\int_{B^c} |p - \widetilde{p}| =: I_B + I_{B^c}.$$

[Corollary B.11](#) implies that $\widetilde{p}(B^c) \leq \delta$. Therefore,

$$I_{B^c} \leq \frac{1}{2}[p(B^c) + \widetilde{p}(B^c)] \leq \delta.$$

Note that $\int_B \rho = p(B) \geq 1 - \delta$ and $\int_{B^c} \rho \leq \delta Z$ (since $\widetilde{p}(B^c) \leq \delta$). Thus,

$$1 - \delta \leq Z = p(B) + \int_{B^c} \rho \leq 1 + 2\delta.$$

Since $\widetilde{p} = p/Z$ on $B$, we have

$$\left| 1 - \frac{1}{Z} \right| \leq \left| \frac{Z - 1}{1 - \delta} \right| \leq \frac{2\delta}{1 - \delta} \leq 4\delta.$$

Therefore, $I_B \leq \frac{1}{2} \cdot 4\delta = 2\delta$.

Combining,

$$\mathrm{TV}(p, \widetilde{p}) \leq 2\delta + \delta = 3\delta.$$

$\qquad\square$

Now, we can consider process $\widetilde{P}$ defined as

$$dx_t = \left( \nabla \log \widetilde{p}(x_t) + \frac{A^T y_{i+1} - A^T A x_t}{\eta_{i+1}^2} \right) dt + \sqrt{2}\, dB_t, \quad x_0 \sim p(x \mid y_i).$$

Then, we have the following lemma.

**Lemma B.15.** *Suppose the following holds:*

$$R \geq r + \frac{T\|A\|}{\eta_{i+1}^2} \Big( \|A\| r + \eta_{i+1}(\sqrt{m} + \sqrt{2\ln(1/\delta)}) \Big) + 2\sqrt{dT \ln(2d/\delta)}.$$

*We have that*

$$\mathbb{E}\left[ \mathrm{TV}(P, \widetilde{P}) \right] \lesssim \delta.$$

*Proof.* Let
$$\mathcal{E} = \left\{ \sup_{t \in [0,T]} \|x_t\| \le R \right\} \quad \text{and} \quad P' = P(\cdot \mid \mathcal{E}), \widetilde{P}' = \widetilde{P}(\cdot \mid \mathcal{E}).$$

Because $s(x) = \nabla \log \widetilde{p}(x)$ for every $x \in B(0, R)$, the drift coefficients of $P$ and $\widetilde{P}$ coincide on the event $\mathcal{E}$, and hence conditioning on $\mathcal{E}$ gives $P' = \widetilde{P}'$.

Then, we have
$$\mathrm{TV}(P, \widetilde{P}) \le \mathrm{TV}(P, P') + \mathrm{TV}(\widetilde{P}, \widetilde{P}') = P(\mathcal{E}^c) + \widetilde{P}(\mathcal{E}^c).$$
Taking expectation over $(y_i, y_{i+1})$ gives
$$\mathbb{E}[\mathrm{TV}(P, \widetilde{P})] \le \mathbb{E}[P(\mathcal{E}^c)] + \mathbb{E}[\widetilde{P}(\mathcal{E}^c)]. \tag{9}$$

Lemma B.3 implies that $\mathbb{E}[P(\mathcal{E}^c)] \lesssim \delta$. Furthermore, the same argument also implies that $\mathbb{E}[\widetilde{P}(\mathcal{E}^c)] \lesssim \delta$. Therefore, we have
$$\mathbb{E}[\mathrm{TV}(P, \widetilde{P})] \lesssim \delta.$$
$\square$

*Proof of Lemma B.2.* We start by considering another process $\widetilde{P}^s$ defined as
$$dx_t = \left( \nabla \log \widetilde{p}(x_t) + \frac{A^T y_{i+1} - A^T A x_t}{\eta_{i+1}^2} \right) dt + \sqrt{2} dB_t, \quad x_0 \sim \widetilde{p}(x \mid y_i).$$

We can see that
$$\mathbb{E}\left[ \mathrm{TV}(\widetilde{P}, \widetilde{P}^s) \right] \le \mathbb{E}\left[ \mathrm{TV}(p(x \mid y_i), \widetilde{p}(x \mid y_i)) \right] \lesssim \delta.$$
Combining this with Lemma B.15, we have that
$$\mathbb{E}\left[ \mathrm{TV}(P, \widetilde{P}^s) \right] \lesssim \delta.$$

By Markov's inequality, we have that
$$\Pr_{y_i, y_{i+1}} \left[ \mathrm{TV}(P, \widetilde{P}^s) \ge \lambda \delta \right] \le O(\lambda^{-1}).$$

Furthermore, by Lemma A.1 and our constraint on $T$, we have that
$$\Pr_{y_i, y_{i+1}} \left[ \mathrm{TV}(\widetilde{P}_T^s, \widetilde{p}(x \mid y_{i+1})) \le \varepsilon \right] \ge 1 - O(\lambda^{-1}).$$

Therefore, we have that
$$\Pr_{y_i, y_{i+1}} \left[ \mathrm{TV}(P_T, \widetilde{p}(x \mid y_{i+1})) \le \varepsilon + \lambda \delta \right] \ge 1 - O(\lambda^{-1}).$$

Combining this with $\Pr\left[ \mathrm{TV}(\widetilde{p}(x \mid y_{i+1}), p(x \mid y_{i+1})) \le \lambda \delta \right] \ge 1 - O(\lambda^{-1})$, we conclude that for $x_T \sim P_T$,
$$\Pr_{y_i, y_{i+1}} \left[ \mathrm{TV}(x_T, p(x \mid y_{i+1})) \le \varepsilon + \lambda \delta \right] \ge 1 - O(\lambda^{-1}).$$
$\square$

## C   Control of Score Approximation and Discretization Errors

In this section, we consider these processes running for time $T$:

- Process $P$:
$$dx_t = \left( s(x_t) + \frac{A^T y_{i+1} - A^T A x_t}{\eta_{i+1}^2} \right) dt + \sqrt{2} dB_t, \quad x_0 \sim p(x \mid y_i)$$

- Process $\widehat{P}$: Let $0 = t_1 < \cdots < t_M = T$ be the $M$ discretization steps with step size $t_{j+1} - t_j = h$. For $t \in [t_j, t_{j+1}]$,
$$dx_t = \left( \widehat{s}(x_{t_j}) + \frac{A^T y_{i+1} - A^T A x_{t_j}}{\eta_{i+1}^2} \right) dt + \sqrt{2} dB_t, \quad x_0 \sim p(x \mid y_i)$$

Note that $\widehat{P}$ is exactly the process (2) we run in Algorithm 1, except that we start from $x_0 \sim p(x \mid y_i)$.

We have shown that the process $P$ will converge to the target distribution $p(x \mid y_{i+1})$. We will show that the process $\widehat{P}$ will also converge to $p(x \mid y_{i+1})$ with a small error

**Lemma C.1.** *Let $p$ be a $(\delta, r, R, \widetilde{L}, \alpha)$ mode-centered locally well-conditioned. Suppose the followings hold for a large enough constant $C > 0$:*

- $T > C \left( \frac{m\gamma_i + \log(\lambda/\varepsilon)}{\alpha} \right)$.

- $\|A\|^4 (T^2 m + TR^2) \leq \frac{\eta_i^4}{C\gamma_i^2}$.

- $R \geq r + \frac{T\|A\|}{\eta_{i+1}^2} \left( \|A\|r + \eta_{i+1}(\sqrt{m} + \sqrt{2\ln(1/\delta)}) \right) + 2\sqrt{dT\ln(2d/\delta)}$.

*Then running $\widehat{P}$ for time $T$ guarantees that with probability at least $1 - 1/\lambda$ over $y_i$ and $y_{i+1}$, we have:*

$$\mathrm{TV}(\widehat{P}_T, p(x \mid y_{i+1})) \lesssim \varepsilon + \lambda\delta + \lambda\sqrt{T} \cdot \left( \left( \widetilde{L}\alpha + \frac{\|A\|^2}{\eta_i^2} \right) \left( h\widetilde{L}\alpha R + \frac{h\|A\|^2 R + h\|A\|\sqrt{m}\eta_i}{\eta_i^2} + \sqrt{dh} \right) + \varepsilon_{score} \right).$$

In this section, we assume $p$ is $(\delta, r, R, \widetilde{L}, \alpha)$ mode-centered locally well-conditioned. Without loss of generality, we assume that the mode of $p$ is at 0, i.e., $\theta = 0$. Let $L := \widetilde{L}\alpha$, i.e., the Lipschitz constant inside the ball $B(0, R)$.

We will also consider the following stochastic processes:

- Process $Q$:
$$dx_t = \left( s(x_t) + \frac{A^T y_i - A^T A x_t}{\eta_i^2} \right) dt + \sqrt{2} dB_t, \quad x_0 \sim p(x \mid y_i)$$

- Process $Q'$ is the process $Q$ conditioned on $x_t \in B(0, R)$ for $t \in [0, T]$.
- Process $P'$ is the process $P$ conditioned on $x_t \in B(0, R)$ for $t \in [0, T]$.

We first note that following the same proof in Lemma B.3 that bounds $\mathrm{TV}(P, P')$, we can also bound $\mathrm{TV}(Q, Q')$.

**Lemma C.2.** *Suppose the following holds:*
$$R \geq r + \frac{T\|A\|}{\eta_i^2} \left( \|A\|r + \eta_i(\sqrt{m} + \sqrt{2\ln(1/\delta)}) \right) + 2\sqrt{dT\ln(2d/\delta)}.$$

*We have that*
$$\mathbb{E}\left[\mathrm{TV}(Q, Q')\right] \lesssim \delta.$$

**Lemma C.3.** *We have*
$$\mathop{\mathbb{E}}_{x_t \sim Q'} \left[ \|x_t - x_{t_j}\|^4 \right] \lesssim \left( hLR + \frac{h\|A\|\|y_i\| + h\|A\|^2 R}{\eta_i^2} \right)^4 + d^2 h^2$$

*Proof.*
$$\mathop{\mathbb{E}}_{x_t \sim Q'} \left[ \|x_t - x_{t_j}\|^4 \right]$$
$$= \mathop{\mathbb{E}}_{x_t \sim Q'} \left[ \left\| \int_{t_j}^t \left( s(x_s) + \frac{A^T y_i - A^T A x_s}{\eta_i^2} \right) ds + \sqrt{2} dB_s \right\|^4 \right]$$
$$\lesssim \mathop{\mathbb{E}}_{x_t \sim Q'} \left[ \left( \int_{t_j}^t \|s(x_s)\| ds \right)^4 \right] + \mathop{\mathbb{E}}_{x_t \sim Q'} \left[ \left( \int_{t_j}^t \left\| \frac{A^T y_i - A^T A x_s}{\eta_i^2} \right\| ds \right)^4 \right] + \mathop{\mathbb{E}}_{x_t \sim Q'} \left[ \left\| \int_{t_j}^t \sqrt{2} dB_s \right\|^4 \right]$$
$$\lesssim (hLR)^4 + \left( \frac{h\|A\|\|y_i\|}{\eta_i^2} \right)^4 + \left( \frac{h\|A\|^2 R}{\eta_i^2} \right)^4 + \mathbb{E}\left[ \left\| \int_{t_j}^t \sqrt{2} dB_s \right\|^4 \right].$$

Since $\int_{t_j}^{t} \sqrt{2} dB_s \sim \mathcal{N}(0, (t - t_j)I_d)$, we have that $\mathbb{E}\|\int_{t_j}^{t} \sqrt{2} dB_s\|^4 \lesssim d^2(t - t_j)^2 \lesssim d^2 h^2$. This gives that

$$\mathbb{E}_{x_t \sim Q'} \left[ \|x_t - x_{t_j}\|^4 \right] \lesssim \left( hLR + \frac{h\|A\|\|y_i\| + h\|A\|^2 R}{\eta_i^2} \right)^4 + d^2 h^2$$

$\square$

**Lemma C.4.** *Suppose* $\|A\|^4(T^2 m + TR^2) \leq \frac{\eta_i^4 \eta_{i+1}^4}{C(\eta_i^2 - \eta_{i+1}^2)^2}$ *for a large enough constant* $C$.

$$\mathbb{E}_{y_i, y_{i+1}, x_t \sim Q'} \left[ \left( \frac{\mathrm{d}\, P'}{\mathrm{d}\, Q'}(x_t) \right)^2 \right] = O(1).$$

*Proof.* By Girsanov's theorem, for any *trajectory* $x_{0,\dots,t}$,

$$\frac{dP'}{dQ'}(x_{0,\dots,t}) = \exp(M_t)$$

where the Girsanov exponent $M_t$ is given by

$$M_t = \frac{1}{\sqrt{2}} \int_0^t \Delta b_y(x_u) \cdot dB_u - \frac{1}{4} \int_0^t \|\Delta b_y(x_u)\|^2 du$$

for

$$\Delta b_y(x_u) = \frac{A^T y_{i+1} - A^T A x_u}{\eta_{i+1}^2} - \frac{A^T y_i - A^T A x_u}{\eta_i^2}$$
$$= \frac{\eta_i^2 A^T y_{i+1} - \eta_{i+1}^2 A^T y_i - A^T A x_u(\eta_i^2 - \eta_{i+1}^2)}{\eta_{i+1}^2 \eta_i^2}.$$

Since $Q'$ is supported in $B(0, R)$,

$$\|\Delta b_y(x_u)\| \leq O\left( \frac{\|A\|\|\eta_i^2 y_{i+1} - \eta_{i+1}^2 y_i\| + \|A\|^2(\eta_i^2 - \eta_{i+1}^2)R}{\eta_{i+1}^2 \eta_i^2} \right) := \kappa_y$$

Now, for $\zeta_y := \int_0^t \|\Delta b_y(x_u)\|^2 du$, we have that $M_t \sim \mathcal{N}\left( -\frac{1}{4}\zeta_y, \frac{1}{2}\zeta_y \right)$
So,

$$\mathbb{E}\left[ \exp(2M_t) \right] \leq \exp(\zeta_y/2) \leq \exp(\kappa_y^2 t/2)$$

Note that $\|\eta_i^2 y_{i+1} - \eta_{i+1}^2 y_i\|^2$ has mean $\|(\eta_i^2 - \eta_{i+1}^2)Ax\|^2$ and is subgamma with variance $m\left(\eta_{i+1}^2 \eta_i^4 - \eta_{i+1}^4 \eta_i^2\right)^2$ and scale $\eta_{i+1}^2 \eta_i^4 - \eta_{i+1}^4 \eta_i^2$. Thus, for $t\|A\|^2 \leq \frac{\eta_{i+1}^2 \eta_i^2}{C(\eta_i^2 - \eta_{i+1}^2)}$ we have

$$\mathbb{E}_{x, y_{i+1}, y_i} [\exp(2M_t)] \leq \mathbb{E}\left[ \exp\left( t \frac{\|A\|^2 \|\eta_i^2 y_{i+1} - \eta_{i+1}^2 y_i\|^2 + (\eta_i^2 - \eta_{i+1}^2)^2\|A\|^4 R^2}{\eta_{i+1}^4 \eta_i^4} \right) \right]$$
$$\lesssim \exp\left( 2\left( \frac{t^2\|A\|^4(\eta_{i+1}^2 \eta_i^4 - \eta_{i+1}^4 \eta_i^2)^2 m}{\eta_{i+1}^8 \eta_i^8} + \frac{(\eta_i^2 - \eta_{i+1}^2)^2\|A\|^4 t R^2}{\eta_{i+1}^4 \eta_i^4} \right) \right)$$
$$= \exp\left( 2\left( \frac{t^2\|A\|^4(\eta_i^2 - \eta_{i+1}^2)^2 m + (\eta_i^2 - \eta_{i+1}^2)^2\|A\|^4 t R^2}{\eta_{i+1}^4 \eta_i^4} \right) \right)$$
$$= \exp\left( \frac{\|A\|^4(\eta_i^2 - \eta_{i+1}^2)^2 \cdot (t^2 m + t R^2)}{\eta_{i+1}^4 \eta_i^4} \right)$$
$$\lesssim 1.$$

$\square$

**Lemma C.5.** *Let $E$ be the event on $y_i$ such that $\mathrm{TV}(Q, Q') \leq \frac{1}{2}$. Suppose*

$$\|A\|^4 (T^2 m + T R^2) \leq \frac{\eta_i^4 \eta_{i+1}^4}{C(\eta_i^2 - \eta_{i+1}^2)^2}.$$

*Then,*

$$\mathop{\mathbb{E}}_{y_i, y_{i+1}} \left[ \mathrm{TV}(P', \widehat{P}) \right] \lesssim 1 - \Pr\left[E\right] + \sqrt{T} \cdot \left( \left( L + \frac{A^T A}{\eta_i^2} \right) \left( hLR + \frac{h \|A\|^2 R + h\|A\|\sqrt{m}\eta_i}{\eta_i^2} + \sqrt{dh} \right) + \varepsilon_{score} \right).$$

*Proof.* Note that the bound is trivial when $\Pr[E] < 1/2$. Therefore, we can use the fact that $\mathbb{E}[\cdot \mid E] \lesssim \mathbb{E}[\cdot]$ throughout the proof. We have, for any $t \in [t_j, t_{j+1}]$, .

$$\mathop{\mathbb{E}}_{y_i, y_{i+1}|E} \mathop{\mathbb{E}}_{x_t \sim P'} \left[ \|s(x_t) - \widehat{s}(x_{t_j})\|^2 + \left\| \frac{A^T A}{\eta_i^2}(x_t - x_{t_j}) \right\|^2 \right]$$

$$= \mathop{\mathbb{E}}_{y_i, y_{i+1}|E} \mathop{\mathbb{E}}_{x_t \sim Q'} \left[ \frac{\mathrm{d}P'}{\mathrm{d}Q'} \cdot \left( \|s(x_t) - \widehat{s}(x_{t_j})\|^2 + \left\| \frac{A^T A}{\eta_i^2}(x_t - x_{t_j}) \right\|^2 \right) \right]$$

$$\lesssim \sqrt{ \mathop{\mathbb{E}}_{y_i, y_{i+1}, x_t \sim Q'} \left[ \left( \frac{\mathrm{d}P'}{\mathrm{d}Q'}(x_t) \right)^2 \right] \cdot \mathop{\mathbb{E}}_{y_i|E} \mathop{\mathbb{E}}_{x_t \sim Q'} \left[ \|s(x_t) - \widehat{s}(x_{t_j})\|^4 + \left\| \frac{A^T A}{\eta_i^2}(x_t - x_{t_j}) \right\|^4 \right] }$$

The first term can be bounded using Lemma C.4. Now we focus on the second term. Note that

$$\mathop{\mathbb{E}}_{y_i|E} \left[ \mathop{\mathbb{E}}_{x_t \sim Q'} \left[ \|s(x_t) - \widehat{s}(x_{t_j})\|^4 \right] \right] \leq \mathop{\mathbb{E}}_{y_i|E} \left[ \mathop{\mathbb{E}}_{x_t \sim Q'} \left[ \|s(x_t) - s(x_{t_j})\|^4 \right] \right] + \mathop{\mathbb{E}}_{y_i|E} \left[ \mathop{\mathbb{E}}_{x_t \sim Q'} \left[ \|s(x_{t_j}) - \widehat{s}(x_{t_j})\|^4 \right] \right].$$

Since $s$ is $L$-Lipschitz in $B(0, R)$, and using Lemma C.3, we have

$$\mathop{\mathbb{E}}_{y_i|E} \left[ \mathop{\mathbb{E}}_{x_t \sim Q'} \left[ \|s(x_t) - s(x_{t_j})\|^4 + \left\| \frac{A^T A}{\eta_i^2}(x_t - x_{t_j}) \right\|^4 \right] \right]$$

$$\lesssim \mathop{\mathbb{E}}_{y_i} \left[ \left( L + \frac{A^T A}{\eta_i^2} \right)^4 \mathop{\mathbb{E}}_{x_t \sim Q'} \left[ \|x_t - x_{t_j}\|^4 \right] \right]$$

$$\lesssim \left( L + \frac{A^T A}{\eta_i^2} \right)^4 \mathop{\mathbb{E}}_{y_i} \left[ \left( hLR + \frac{h \|A\| y_i + h\|A\|^2 R}{\eta_i^2} \right)^4 + d^2 h^2 \right]$$

$$\lesssim \left( L + \frac{A^T A}{\eta_i^2} \right)^4 \left( hLR + \frac{h\|A\|^2 R + h\|A\|\sqrt{m}\eta_i}{\eta_i^2} + \sqrt{dh} \right)^4.$$

Since $Q'$ is a conditional measure of $Q$, conditioned on $E$, we have $\frac{\mathrm{d}Q'}{\mathrm{d}Q} \leq \frac{1}{1 - \mathrm{TV}(Q', Q)} \leq 2$. Therefore,

$$\mathop{\mathbb{E}}_{y_i|E} \left[ \mathop{\mathbb{E}}_{x_{t_j} \sim Q'} \left[ \|s(x_{t_j}) - \widehat{s}(x_{t_j})\|^4 \right] \right] \leq \mathop{\mathbb{E}}_{y_i|E} \left[ 2 \cdot \mathop{\mathbb{E}}_{x_t \sim Q} \left[ \|s(x_{t_j}) - \widehat{s}(x_{t_j})\|^4 \right] \right]$$

$$\lesssim \mathop{\mathbb{E}}_{y_i} \left[ \mathop{\mathbb{E}}_{x_t \sim Q} \left[ \|s(x_{t_j}) - \widehat{s}(x_{t_j})\|^4 \right] \right]$$

$$\leq \varepsilon_{score}^4$$

This gives that

$$\mathop{\mathbb{E}}_{y_i, y_{i+1}|E} \mathop{\mathbb{E}}_{x_t \sim P'} \left[ \|s(x_t) - \widehat{s}(x_{t_j})\|^2 + \left\| \frac{A^T A}{\eta_i^2}(x_t - x_{t_j}) \right\|^2 \right]$$

$$\lesssim \left( L + \frac{A^T A}{\eta_i^2} \right)^2 \left( hLR + \frac{h\|A\|^2 R + h\|A\|\sqrt{m}\eta_i}{\eta_i^2} + \sqrt{dh} \right)^2 + \varepsilon_{score}^2.$$

Thus, by Girsanov's theorem,

$$\mathop{\mathbb{E}}_{y_i,y_{i+1}|E}\left[\mathrm{KL}\left(P'\,\|\,\widehat{P}\right)\right] \lesssim \sum_{j=0}^{M-1}\int_{t_j}^{t_{j+1}}\mathop{\mathbb{E}}_{y_i,y_{i+1},x_t\sim P'}\left[\|s(x_t)-\widehat{s}(x_{t_j})\|^2+\left\|\frac{A^T A}{\eta_i^2}(x_t-x_{t_j})\right\|^2\right]$$

$$\lesssim T\cdot\left(\left(L+\frac{\|A\|^2}{\eta_i^2}\right)^2\left(hLR+\frac{h\,\|A\|^2\,R+h\|A\|\sqrt{m}\eta_i}{\eta_i^2}+\sqrt{dh}\right)^2+\varepsilon_{score}^2\right).$$

By Pinsker's inequality,

$$\mathop{\mathbb{E}}_{y_i,y_{i+1}|E}\left[\mathrm{TV}(P',\widehat{P})\right]\lesssim \sqrt{T}\cdot\left(\left(L+\frac{\|A\|^2}{\eta_i^2}\right)\left(hLR+\frac{h\,\|A\|^2\,R+h\|A\|\sqrt{m}\eta_i}{\eta_i^2}+\sqrt{dh}\right)+\varepsilon_{score}\right)$$

Hence,

$$\mathop{\mathbb{E}}_{y_i,y_{i+1}}\left[\mathrm{TV}(P',\widehat{P})\right]\leq 1-\Pr\left[E\right]+\mathop{\mathbb{E}}_{y_i,y_{i+1}|E}\left[\mathrm{TV}(P',\widehat{P})\right]$$

$$\lesssim 1-\Pr\left[E\right]+\sqrt{T}\cdot\left(\left(L+\frac{\|A\|^2}{\eta_i^2}\right)\left(hLR+\frac{h\,\|A\|^2\,R+h\|A\|\sqrt{m}\eta_i}{\eta_i^2}+\sqrt{dh}\right)+\varepsilon_{score}\right).$$

$\square$

Then have the following as a corollary:

**Corollary C.6.** *Suppose*

$$\|A\|^4(T^2 m+TR^2)\leq\frac{\eta_i^4\eta_{i+1}^4}{C(\eta_i^2-\eta_{i+1}^2)^2}.$$

*Then,*

$$\mathop{\mathbb{E}}_{y_i,y_{i+1}}\left[\mathrm{TV}(P,\widehat{P})\right]\lesssim \mathbb{E}\left[\mathrm{TV}(P,P')\right]+\mathbb{E}\left[\mathrm{TV}(Q,Q')\right]$$

$$+\sqrt{T}\cdot\left(\left(L+\frac{\|A\|^2}{\eta_i^2}\right)\left(hLR+\frac{h\,\|A\|^2\,R+h\|A\|\sqrt{m}\eta_i}{\eta_i^2}+\sqrt{dh}\right)+\varepsilon_{score}\right).$$

*Proof.* We have that

$$\mathop{\mathbb{E}}_{y_i,y_{i+1}}\left[\mathrm{TV}(P,\widehat{P})\right]\leq \mathop{\mathbb{E}}_{y_i,y_{i+1}}\left[\mathrm{TV}(P,P')\right]+\mathop{\mathbb{E}}_{y_i,y_{i+1}}\left[\mathrm{TV}(P',\widehat{P})\right].$$

Furthermore,

$$\mathop{\mathbb{E}}_{y_i,y_{i+1}}\left[\mathrm{TV}(P',\widehat{P})\right]$$

$$\lesssim \Pr\left[\mathrm{TV}(Q,Q')>\frac{1}{2}\right]+\sqrt{T}\cdot\left(\left(L+\frac{\|A\|^2}{\eta_i^2}\right)\left(hLR+\frac{h\,\|A\|^2\,R+h\|A\|\sqrt{m}\eta_i}{\eta_i^2}+\sqrt{dh}\right)+\varepsilon_{score}\right)$$

$$\lesssim \mathbb{E}\left[\mathrm{TV}(Q,Q')\right]+\sqrt{T}\cdot\left(\left(L+\frac{\|A\|^2}{\eta_i^2}\right)\left(hLR+\frac{h\,\|A\|^2\,R+h\|A\|\sqrt{m}\eta_i}{\eta_i^2}+\sqrt{dh}\right)+\varepsilon_{score}\right),$$

where the last line follows from Markov's inequality. The gives the result. $\square$

*Proof of Lemma C.1.* We note that by our definition of $\gamma_i$,

$$\|A\|^4(T^2 m+TR^2)\leq\frac{\eta_i^4\eta_{i+1}^4}{C(\eta_i^2-\eta_{i+1}^2)^2}\iff\|A\|^4(T^2 m+TR^2)\leq\frac{\eta_i^4}{C\gamma_i^2}$$

Then, combining Corollary C.6 with Lemmas B.3 and C.2, we have

$$\mathbb{E}_{y_i, y_{i+1}} \left[ \mathrm{TV}(P, \widehat{P}) \right] \lesssim \mathbb{E} \left[ \mathrm{TV}(P, P') \right] + \mathbb{E} \left[ \mathrm{TV}(Q, Q') \right]$$

$$+ \sqrt{T} \cdot \left( \left( L + \frac{\|A\|^2}{\eta_i^2} \right) \left( hLR + \frac{h \|A\|^2 R + h\|A\|\sqrt{m}\eta_i}{\eta_i^2} + \sqrt{dh} \right) + \varepsilon_{score} \right)$$

$$\lesssim \delta + \sqrt{T} \cdot \left( \left( L + \frac{\|A\|^2}{\eta_i^2} \right) \left( hLR + \frac{h \|A\|^2 R + h\|A\|\sqrt{m}\eta_i}{\eta_i^2} + \sqrt{dh} \right) + \varepsilon_{score} \right)$$

The conditions in Lemmas B.3 and C.2 are satisfied by our assumptions, noting that $\eta_{i+1} < \eta_i$ implies the bound on $R$ holds for both processes.

Applying Markov's inequality and combining Lemma B.2 with the above, we conclude the proof.
□

# D   Admissible Noise Schedule

Recall that we can define process $\widehat{P}_i$ that converges from $p(x \mid y_i)$ to $p(x \mid y_{i+1})$: Let $0 = t_1 < \cdots < t_M = T$ be the $M$ discretization steps with step size $t_{j+1} - t_j = h$. For $t \in [t_j, t_{j+1}]$,

$$dx_t = \left( \widehat{s}(x_{t_j}) + \frac{A^T y_{i+1} - A^T A x_{t_j}}{\eta_{i+1}^2} \right) dt + \sqrt{2} dB_t, \quad x_0 \sim p(x \mid y_i) \tag{10}$$

We have already proven that we can converge the process from $p(x \mid y_i)$ to $p(x \mid y_{i+1})$ with good probability, as long as some conditions are satisfied. Those conditions actually depend on the choice of the schedule of $\eta_i$ and $T_i$. In this section, we will specify the schedule of $\eta_i$ and $T_i$.

Now we specify the schedule of $\eta_i$ and $T_i$.

**Definition D.1.** *We say a noise schedule $\eta_1 > \cdots > \eta_N$ together with running times $T_1, \cdots, T_{N-1}$ is* admissible *(for a set of parameters $C, \alpha, \lambda, A, d, \varepsilon, \eta, R$) if:*

- *$\eta_N = \eta$;*

- *$\eta_1 \geq \frac{\lambda\|A\|}{\varepsilon} \sqrt{\frac{d}{\alpha}}$;*

- *For all $\gamma_i = (\eta_i/\eta_{i+1})^2 - 1$, we have $\gamma_i \leq 1$ and*

$$T_i \geq C \left( \frac{m\gamma_i + \log(\lambda/\varepsilon)}{\alpha} \right).$$

  *Furthermore,*

$$\|A\|^4 (T_i^2 m + T_i R^2) \leq \frac{\eta_i^4}{C\gamma_i^2}.$$

The reason we need to satisfy the last inequality is to satisfy the conditions in Lemma C.1. We formalize this in the following lemma.

**Lemma D.2.** *Let $C > 0$ be a sufficiently large constant and $p$ be a $(\delta, r, R, \widetilde{L}, \alpha)$ mode-centered locally well-conditioned distribution. For any $\delta, \varepsilon \in (0, 1)$ and $\lambda > 1$, suppose*

$$R \geq r + C \left( \frac{(m + \log\frac{\lambda}{\varepsilon})\|A\|}{\alpha\eta^2} \left( \|A\| r + \eta\sqrt{m + \log(1/\delta)} \right) + \sqrt{\frac{d \log(d/\delta)(m + \log(\lambda/\varepsilon))}{\alpha}} \right).$$

*For any admissible schedule $(\eta_i)_{i \in [N]}$ and $(T_i)_{i \in [N-1]}$, running the process $\widehat{P}_i$ for time $T_i$ guarantees that with probability at least $1 - 1/\lambda$ over $y_i$ and $y_{i+1}$:*

$$\mathrm{TV}(x_{T_i}, p(x \mid y_{i+1})) \lesssim \varepsilon + \lambda\delta + \lambda\sqrt{\frac{m + \log(\lambda/\varepsilon)}{\alpha}} \cdot (\varepsilon_{dis} + \varepsilon_{score}),$$

*where*

$$\varepsilon_{dis} := \left( \widetilde{L}\alpha + \frac{\|A\|^2}{\eta^2} \right) \left( h\widetilde{L}\alpha R + \frac{h \|A\|^2 R + h\|A\|\sqrt{m}\eta}{\eta^2} + \sqrt{dh} \right).$$

*Proof.* It is straightforward to verify that an admissible schedule satisfies the first two conditions of Lemma C.1.

For the third condition regarding $R$, our assumption states:

$$R \geq r + C\left(\frac{(m + \log\frac{\lambda}{\varepsilon})\|A\|}{\alpha\eta^2}\left(\|A\|\, r + \eta\sqrt{m + \log(1/\delta)}\right) + \sqrt{\frac{d\log(d/\delta)(m + \log(\lambda/\varepsilon))}{\alpha}}\right)$$

Given that $T_i \lesssim \frac{m + \log(\lambda/\varepsilon)}{\alpha}$, this choice of $R$ is sufficient to satisfy the third condition in Lemma C.1.

Therefore, applying Lemma C.1 at each step $i$, we obtain that with probability at least $1 - 1/\lambda$ over $y_i$ and $y_{i+1}$:

$$\mathrm{TV}(x_{T_i}, p(x \mid y_{i+1}))$$

$$\lesssim \varepsilon + \lambda\delta + \lambda\sqrt{T_i} \cdot \left(\left(\widetilde{L}\alpha + \frac{\|A\|^2}{\eta_i^2}\right)\left(h\widetilde{L}\alpha R + \frac{h\|A\|^2 R + h\|A\|\sqrt{m}\eta_i}{\eta_i^2} + \sqrt{dh}\right) + \varepsilon_{score}\right)$$

$$\lesssim \varepsilon + \lambda\delta + \lambda\sqrt{\frac{m + \log(\lambda/\varepsilon)}{\alpha}} \cdot (\varepsilon_{dis} + \varepsilon_{score}).$$

$\square$

We also want to prove the following two lemmas:

**Lemma D.3.** *Let $p$ be a $d$-dimensional $(\delta, r, R, \widetilde{L}, \alpha)$ mode-centered locally well-conditioned distribution. For any $\delta \in (0, 1)$, suppose*

$$R \geq 2\sqrt{\frac{d}{\alpha}} + \sqrt{\frac{2\log(1/\delta)}{\alpha}}.$$

*Then, suppose $\eta_1 \geq \frac{\lambda\|A\|}{\varepsilon}\sqrt{\frac{d}{\alpha}}$, with probability at least $1 - \frac{1}{\lambda}$ over $y_1$,*

$$\mathrm{TV}(p(x \mid y_1),\, p(x)) \lesssim \varepsilon + \lambda\delta.$$

**Lemma D.4.** *There exists an admissible noise such that*

$$N \lesssim \rho^2\sqrt{m}\log(\lambda/\varepsilon) + \frac{\rho^2\alpha R^2}{\sqrt{m}} + \frac{m^2}{m\log(\lambda/\varepsilon) + \alpha R^2} + \log\left(2 + \frac{\lambda\sqrt{d}\rho}{\varepsilon}\right),$$

*where $\rho = \frac{\|A\|}{\eta\sqrt{\alpha}}$.*

## D.1 The Closeness Between $p(x \mid y_1)$ and $p(x)$

In this part, we prove Lemma D.3, showing that any admissible schedule has a large enough $\eta_1$, enabling us to use $p(x)$ to approximate $p(x \mid y_1)$.

We have the following standard information-theoretic result.

**Lemma D.5.** *Let $X \in \mathbb{R}^m$ be a random variable, and $Y = X + \mathcal{N}(0, \eta^2 I_m)$. Then,*

$$I(X; Y) \leq \frac{1}{2}\log\det\left(I_m + \frac{\mathrm{Cov}(X)}{\eta^2}\right).$$

**Lemma D.6.** *For any distribution $p$ with $\mathbb{E}_{x \sim p}\left[\|x - \mathbb{E}\,x\|^2\right] = m_2^2$, we have*

$$\mathbb{E}\left[\mathrm{TV}(p(x \mid y_1), p(x))\right] \leq \frac{\|A\|m_2}{2\eta_1}.$$

*Proof.* Note that $\mathbb{E}\left[\mathrm{KL}(p(x \mid y_i) \,\|\, p(x))\right]$ is exactly the mutual information between $x$ and $y_i$. In addition, we have

$$\mathbb{E}\left[\mathrm{KL}(p(x \mid y_i) \,\|\, p(x))\right] = I(x; y_i) \leq I(Ax; y_i) \leq \frac{1}{2}\log\det\left(I_m + \frac{\mathrm{Cov}(Ax)}{\eta_i^2}\right) \leq \frac{\|A\|^2 m_2^2}{2\eta_i^2}.$$

By Pinsker's inequality, we have

$$\mathbb{E}\left[\mathrm{TV}(p(x \mid y_1), p(x))\right] \leq \frac{\|A\| m_2}{2\eta_1}.$$

$\square$

**Lemma D.7.** *Let $p$ be a $d$-dimensional $(\delta, r, R, \widetilde{L}, \alpha)$ mode-centered locally well–conditioned probability distribution. Assume*

$$R \geq 2\sqrt{\frac{d}{\alpha}} + \sqrt{\frac{2\log(1/\delta)}{\alpha}}.$$

*Then*

$$\mathbb{E}_{y_1}[\mathrm{TV}(p(x \mid y_1), p(x))] \lesssim \frac{\|A\|}{\eta_1}\sqrt{\frac{d}{\alpha}} + \delta.$$

*Proof.* Lemma B.14 provides an $\alpha$-strongly log–concave density $\widetilde{p}$ satisfying

$$\mathrm{TV}(p, \widetilde{p}) \leq 3\delta.$$

For an $\alpha$-strongly log–concave law the Brascamp–Lieb inequality yields $\mathrm{Cov}_{\widetilde{p}} \preceq \alpha^{-1} I_d$; hence

$$m_2(\widetilde{p}) := (\mathbb{E}_{\widetilde{p}}\|x - \mathbb{E}_{\widetilde{p}} x\|^2)^{1/2} \leq \sqrt{\frac{d}{\alpha}}.$$

Applying Lemma D.6 to $\widetilde{p}$ gives

$$\mathbb{E}_{y_1}[\mathrm{TV}(\widetilde{p}(x \mid y_1), \widetilde{p}(x))] \leq \frac{\|A\|}{2\eta_1}\sqrt{\frac{d}{\alpha}}.$$

Note that

$$\mathrm{TV}(p(x \mid y_1), p(x)) \leq \mathrm{TV}(p(x \mid y_1), \widetilde{p}(x \mid y_1)) + \mathrm{TV}(\widetilde{p}(x \mid y_1), \widetilde{p}(x)) + \mathrm{TV}(\widetilde{p}(x), p(x)).$$

Integrating in $y_1$ and using the elementary fact

$$\mathbb{E}_{y_1}[\mathrm{TV}(p(x \mid y_1), \widetilde{p}(x \mid y_1))] \leq \mathrm{TV}(p, \widetilde{p}),$$

together with the above calculaion, yields

$$\mathbb{E}_{y_1}[\mathrm{TV}(p(x \mid y_1), p(x))] \leq 3\delta + \frac{\|A\|}{2\eta_1}\sqrt{\frac{d}{\alpha}} + 3\delta.$$

This proves the stated bound. $\square$

Now we prove Lemma D.3.

*Proof of Lemma D.3.* By Lemma D.7, we have

$$\mathbb{E}_{y_1}[\mathrm{TV}(p(x \mid y_1), p(x))] \lesssim \frac{\|A\|}{\eta_1}\sqrt{\frac{d}{\alpha}} + \delta.$$

Since all admissible noise schedules satisfy $\eta_1 \geq \frac{\lambda\|A\|}{\varepsilon}\sqrt{\frac{d}{\alpha}}$. This implies

$$\frac{\|A\|}{\eta_1}\sqrt{\frac{d}{\alpha}} \leq \frac{\varepsilon}{\lambda}.$$

Consequently,

$$\mathbb{E}_{y_1}[\mathrm{TV}(p(x \mid y_1), p(x))] \lesssim \frac{\varepsilon}{\lambda} + \delta.$$

By Markov's inequality, with probability at least $1 - \frac{1}{\lambda}$ over $y_1$,

$$\mathrm{TV}(p(x \mid y_1), p(x)) \lesssim \varepsilon + \lambda\delta,$$

which proves the lemma. $\square$

## D.2 Bound for $N$ Mixing Steps

In this part, we prove Lemma D.4.

**Lemma D.8.** *Let $a, x_0 > 0$, and let $c > 0$. Consider the number sequence*

$$x_{i+1} = (1 + \min((ax_i)^c, 1))x_i.$$

*For every $B > 0$, let $k(B)$ be the minimum integer $i$ such that $x_i \geq B$. Then*

$$k(B) = O\left((ax_0)^{-c} + \log\left(1 + \frac{B}{x_0}\right)\right).$$

*Proof.* We show in two steps that the time to go from $x_0$ to $1/a$, then to $B$. Define

$$k_1 = \min\{i \in \mathbb{N} : x_i \geq 1/a\},$$

**Bound for $k_1$.** We first show that $k_1 \lesssim (ax_0)^{-c}$. Consider the quantities

$$N_j = \min\{i \in \mathbb{N} : x_i \geq 2^j x_0\},$$

and let $j^*$ be the smallest $j$ such that $x_{N_j} \geq 1/a$. If instead $x_0 \geq 1/a$ already, then $k_1 = 0$ and there is nothing to prove.

Assume $x_0 < 1/a$. For each $j < j^*$ define

$$t_j = (2^j ax_0)^{-c}.$$

We claim that

$$N_{j+1} - N_j \leq t_j.$$

Indeed, for each $j < j^*$,

$$x_{N_j + t_j} \geq x_{N_j} \prod_{i=N_j}^{N_j + t_j - 1} \left(1 + (ax_i)^c\right)$$

$$\geq x_{N_j} \prod_{i=N_j}^{N_j + t_j - 1} \left(1 + (ax_{N_j})^c\right) = x_{N_j} \left(1 + (ax_{N_j})^c\right)^{t_j}.$$

Since

$$(ax_{N_j})^c \geq (a \cdot 2^j x_0)^c = \frac{1}{t_j},$$

we get

$$x_{N_j + t_j} \geq \left(1 + \frac{1}{t_j}\right)^{t_j} x_{N_j} \geq 2 x_{N_j} \geq 2^{j+1} x_0.$$

By monotonicity of the sequence $(x_i)$, it follows that $N_{j+1} \leq N_j + t_j$. Summing over $j$ up to $j^* - 1$ gives

$$N_{j^*} = \sum_{j=0}^{j^*-1}(N_{j+1} - N_j) \leq \sum_{j=0}^{j^*-1}(2^j ax_0)^{-c} \lesssim (ax_0)^{-c}.$$

By definition, $N_{j^*}$ is the first index $i$ such that $x_i \geq 1/a$, so $k_1 = N_{j^*} \lesssim (ax_0)^{-c}$.

**Bound to achieve $B$.** If $B \leq 1/a$, the bound already holds. Now we analyze how many steps Note that for every $i \geq k_1$,

$$x_{i+1} = (1 + \min((ax_i)^c, 1))x_i = 2x_i.$$

Therefore, we have

$$x_{k_1 + \log_2(B)} \geq 2^{\log_2(B/x_{k_1})} x_{k_1} \geq B.$$

This proves that

$$k(B) \leq k_1 + \log_2\left(1 + \frac{B}{x_{k_1}}\right) \leq k_1 + \log_2\left(1 + \frac{B}{x_0}\right) \lesssim (ax_0)^{-c} + \log\left(1 + \frac{B}{x_0}\right).$$

$\square$

**Lemma D.9.** *Given parameters $x_0, a, b > 0$, consider sequence inductively defined by $x_{i+1} = (1 + \gamma_i)x_i$, where*

$$\gamma_i = \min\left\{\gamma_i \leq 1 \ : \ a\gamma^2 + b\gamma \leq 2x_i\right\}.$$

*Given $B$, let $k(B)$ be the minimum integer $i$ such that $x_i \geq B$. Then,*

$$k(B) \lesssim \frac{b}{x_0} + \frac{a}{b} + \log\left(1 + \frac{B}{x_0}\right).$$

*Proof.* We do case analysis.

**Case 1:** $x_0 \geq b^2/a$. We always choose $\gamma_i = \sqrt{x_i/a}$. We can verify that

$$a\left(\frac{x_i}{a}\right) + b\sqrt{\frac{x_i}{a}} \leq x_i + \sqrt{\frac{b^2}{a} \cdot x_i} \leq 2x_i,$$

and this satisfies the requirement for $\gamma_i$. By applying Lemma D.8, we have that

$$k(B) \lesssim \left(\frac{x_0}{a}\right)^{-1/2} + \log\left(1 + \frac{B}{x_0}\right) \leq \frac{a}{b} + \log\left(1 + \frac{B}{x_0}\right).$$

**Case 2:** $x_0 \leq B \leq b^2/a$. We always choose $\gamma_i = \min(x_i/b, 1)$. We can verify that

$$a\left(\frac{x_i}{b}\right)^2 + b\left(\frac{x_i}{b}\right) \leq x_i\left(\frac{ax_i}{b^2}\right) + x_i \leq 2x_i,$$

and this satisfies the requirement for $\gamma_i$. By applying Lemma D.8, we have that

$$k(B) \lesssim (x_0/b)^{-1} + \log\left(1 + \frac{B}{x_0}\right).$$

**Case 3:** $x_0 \leq b^2/a \leq B$. We combine the bound for the first two cases, where we first go from $x_0$ to $b^2/a$, then go from $b^2/a$ to $B$. Then we have

$$k(B) \lesssim \left((x_0/b)^{-1} + \log\left(1 + \frac{B}{x_0}\right)\right) + \left(\frac{a}{b} + \log\left(1 + \frac{B}{x_0}\right)\right) \lesssim \frac{b}{x_0} + \frac{a}{b} + \log\left(1 + \frac{B}{x_0}\right).$$

$\square$

*Proof of Lemma D.4.* Now we describe how we construct an admissible noise schedule. Consider we start from $\eta_1' = \eta$, and for each $i$, we iteratively choose $\gamma_i'$ to be the maximum $\gamma \leq 1$ such that

$$\|A\|^4(f_T^2(\gamma)m + f_T(\gamma)R^2) \leq \frac{(\eta_i')^4}{C\gamma^2},$$

and then set $\eta_{i+1}' = \sqrt{(1 + \gamma_i')(\eta_i')^2}$. We continue this process until we reach $\eta_N' \geq \frac{\lambda\|A\|}{\varepsilon}\sqrt{\frac{d}{\alpha}}$. It is easy to verify that $(\eta_N', \eta_{N-1}', \ldots, \eta_1')$ is an admissible noise schedule. Now we bound the number of iterations $N$.

Since for all $\gamma$, we have $\|A\|^4(f_T^2(\gamma)m + f_T(\gamma)R^2) \leq \|A\|^4(\sqrt{m}f_T(\gamma) + \frac{R^2}{2\sqrt{m}})^2$, a sufficient condition for $\|A\|^4(f_T^2(\gamma)m + f_T(\gamma)R^2) \leq \frac{(\eta_i')^4}{C\gamma^2}$ is that

$$\|A\|^4(\sqrt{m}f_T(\gamma) + \frac{R^2}{2\sqrt{m}})^2 \leq \frac{(\eta_i')^4}{C\gamma^2} \iff \|A\|^2(\sqrt{m}f_T(\gamma) + \frac{R^2}{2\sqrt{m}}) \leq \frac{(\eta_i')^2}{C\gamma}.$$

Therefore, fixing $\eta_i'$, we have that $\gamma_i'$ is at least

$$\max\left\{\gamma \leq 1 \ : \ \frac{\|A\|^2 m^{1.5}}{\alpha}\gamma^2 + \left(\frac{\|A\|^2\sqrt{m}\log(\lambda/\varepsilon)}{\alpha} + \frac{\|A\|^2 R^2}{\sqrt{m}}\right)\gamma \leq \frac{(\eta_i')^2}{C}\right\}.$$

Now we look at the inductive sequence starting from $x_1 = \eta^2$, and $x_{i+1} = (1 + \widetilde{\gamma}_i)x_i$, where

$$\widetilde{\gamma}_i = \max\left\{\gamma \leq 1 \; : \; \frac{\|A\|^2 \, m^{1.5}}{\alpha}\gamma^2 + \left(\frac{\|A\|^2 \, \sqrt{m}\log(\lambda/\varepsilon)}{\alpha} + \frac{\|A\|^2 \, R^2}{\sqrt{m}}\right)\gamma \leq \frac{x_i}{C}\right\}.$$

By Lemma D.9, we know that for any $\eta_{goal} > 0$, we can achieve $x_N \geq \eta_{goal}^2$ within

$$N \lesssim \frac{\|A\|^2 \, \sqrt{m}\log(\lambda/\varepsilon)}{\alpha\eta^2} + \frac{\|A\|^2 \, R^2}{\sqrt{m}\eta^2} + \frac{m^2}{m\log(\lambda/\varepsilon) + \alpha R^2} + \log\left(2 + \frac{\eta_{goal}}{\eta}\right).$$

Taking in $\eta_{goal} = \frac{\lambda\|A\|}{\varepsilon}\sqrt{\frac{d}{\alpha}}$, we conclude the lemma. $\qquad\square$

# E    Theoretical Analysis of Algorithm 1

In this section, we analyze the algorithm presented in Algorithm 1. In Algorithm 1, the algorithm initializes by drawing a sample from the prior distribution $p(x)$ via the diffusion SDE, which introduces sampling error. [CCL+22] demonstrated that this diffusion sampling error is polynomially small, with the exact magnitude depending on the discretization scheme chosen for the diffusion SDE. Since the focus of this paper is on enabling an unconditional diffusion sampling model to perform posterior sampling, the choice of diffusion discretization and its associated error are not not the focus of our analysis. Consequently, we omit the diffusion sampling error in the error analysis presented in this section. This omission does not impact the rigor of the theorems in the main paper, as the error is polynomially small.

We start with the following lemma:

**Lemma E.1.** *Let $C > 0$ be a large enough constant. Let $p$ be a $(\delta, r, R, \widetilde{L}, \alpha)$ mode-centered locally well-conditioned distribution. For every $\delta, \varepsilon \in (0, 1)$ and $\lambda > 1$, suppose*

$$R \geq r + C\left(\frac{(m + \log\frac{\lambda}{\varepsilon})\|A\|}{\alpha\eta^2}\left(\|A\|\,r + \eta\sqrt{m + \log(1/\delta)}\right) + \sqrt{\frac{d\log(d/\delta)(m + \log(\lambda/\varepsilon))}{\alpha}}\right).$$

*Then running Algorithm 1 will guarantee that*

$$\Pr_{y_1,\ldots,y_N}\left[\mathrm{TV}(X_N, p(x \mid y)) \lesssim N\left(\varepsilon + \lambda\delta + \lambda\sqrt{\frac{m + \log(\lambda/\varepsilon)}{\alpha}} \cdot (\varepsilon_{dis} + \varepsilon_{score})\right)\right] \geq 1 - \frac{N}{\lambda},$$

*where*

$$\varepsilon_{dis} := \left(\widetilde{L}\alpha + \frac{\|A\|^2}{\eta^2}\right)\left(h\widetilde{L}\alpha R + \frac{h\|A\|^2 \, R + h\|A\|\sqrt{m}\eta}{\eta^2} + \sqrt{dh}\right).$$

*Proof.* Let $\varepsilon_{\mathrm{step}} := C_0\left(\varepsilon + \lambda\delta + \lambda\sqrt{\frac{m + \log(\lambda/\varepsilon)}{\alpha}} \cdot (\varepsilon_{\mathrm{dis}} + \varepsilon_{\mathrm{score}})\right)$, where $C_0$ is a constant large enough to absorb the implicit constants in Lemma D.3 and Lemma D.2.

We prove by induction that for each $i \in [N]$:

$$\Pr_{y_1,\ldots,y_i}\left[\mathrm{TV}(X_i, p(x \mid y_i)) \leq i \cdot \varepsilon_{\mathrm{step}}\right] \geq 1 - \frac{i}{\lambda}. \tag{11}$$

For the base case ($i = 1$), since $X_1 \sim p(x)$, Lemma D.3 gives that $\mathrm{TV}(p(x), p(x \mid y_1)) \leq \varepsilon_{\mathrm{step}}$ with probability at least $1 - 1/\lambda$ over $y_1$.

For the inductive step, assume the statement holds for some $i < N$. Let $\mathcal{E}_i$ be the event that $\mathrm{TV}(X_i, p(x \mid y_i)) \leq i \cdot \varepsilon_{\mathrm{step}}$, so $\Pr[\mathcal{E}_i^c] \leq i/\lambda$.

Let $X_i^* \sim p(x \mid y_i)$ and let $X_{i+1}^*$ be the result of evolving $X_i^*$ for time $T_i$ using the SDE in Equation (2). By Lemma D.2, the event $\mathcal{F}_{i+1}$ that $\mathrm{TV}(X_{i+1}^*, p(x \mid y_{i+1})) \leq \varepsilon_{\mathrm{step}}$ has probability at least $1 - 1/\lambda$ over $y_i, y_{i+1}$ and the SDE path.

By the triangle inequality and data processing inequality:

$$\text{TV}(X_{i+1}, p(x \mid y_{i+1})) \leq \text{TV}(X_i, p(x \mid y_i)) + \text{TV}(X_{i+1}^*, p(x \mid y_{i+1})). \qquad (12)$$

If both $\mathcal{E}_i$ and $\mathcal{F}_{i+1}$ occur, then $\text{TV}(X_{i+1}, p(x \mid y_{i+1})) \leq (i+1)\varepsilon_{\text{step}}$. The probability that this bound fails is at most:

$$\Pr[\mathcal{E}_i^c \cup \mathcal{F}_{i+1}^c] \leq \Pr[\mathcal{E}_i^c] + \mathbb{E}_{y_1, \ldots, y_i}[\mathbf{1}_{\mathcal{E}_i} \Pr[\mathcal{F}_{i+1}^c \mid y_1, \ldots, y_i]]$$
$$\leq \frac{i}{\lambda} + \frac{1}{\lambda} = \frac{i+1}{\lambda}.$$

Thus, the induction holds for $i+1$, and the lemma follows for $i = N$. $\qquad \square$

**Lemma E.2.** *Let $S_1$ and $S_2$ be two random variables such that*

$$\Pr_{y_1, \ldots, y_N} [\text{TV}((S_1 \mid y_1, \ldots, y_N), (S_2 \mid y_1, \ldots, y_N)) \leq \varepsilon] \geq 1 - \delta.$$

*Then we have*

$$\Pr_{y_N} [\text{TV}((S_1 \mid y_N), (S_2 \mid y_N)) \leq 2\varepsilon] \geq 1 - \frac{\delta}{\varepsilon}.$$

*Proof.* Let $E(y_1, \ldots, y_N)$ be the event such that $\text{TV}((S_1 \mid E), p((S_2 \mid E)) \leq \varepsilon$. Then, we have that

$$\text{TV}((S_1 \mid y_N), (S_2 \mid y_N)) \leq \Pr[\overline{E} \mid y_N] + \varepsilon.$$

Since $\Pr[E] \geq 1 - \delta$, we apply Markov's inequality, and have

$$\Pr_{y} [\Pr[\overline{E} \mid y_N] \geq \varepsilon] \leq \frac{\mathbb{E}_y [\Pr[\overline{E} \mid y_N]]}{\varepsilon} = \frac{\Pr[\overline{E}]}{\varepsilon} \leq \frac{\delta}{\varepsilon}.$$

Hence, we have with probability $1 - \frac{\delta}{\varepsilon}$ over $y$,

$$\text{TV}((S_1 \mid y_N), (S_2 \mid y_N)) \leq 2\varepsilon.$$

$\qquad \square$

Applying Lemma E.2 on Lemma E.1 gives the following corollary.

**Corollary E.3.** *Let $C > 0$ be a large enough constant. Let $p$ be a $(\delta, r, R, \widetilde{L}, \alpha)$ mode-centered locally well-conditioned distribution. For every $\delta, \varepsilon \in (0, 1)$ and $\lambda > 1$, suppose*

$$R \geq r + C \left( \frac{(m + \log \frac{\lambda}{\varepsilon}) \|A\|}{\alpha \eta^2} \left( \|A\| r + \eta \sqrt{m + \log(1/\delta)} \right) + \sqrt{\frac{d \log(d/\delta)(m + \log(\lambda/\varepsilon))}{\alpha}} \right).$$

*Define Then running Algorithm 1 will guarantee that*

$$\Pr_{y} [\text{TV}(X_N, p(x \mid y)) \leq \varepsilon_{error}] \geq 1 - \frac{N}{\lambda \varepsilon_{error}},$$

*with*

$$\varepsilon_{error} \lesssim N \left( \varepsilon + \lambda \delta + \lambda \sqrt{\frac{m + \log(\lambda/\varepsilon)}{\alpha}} \cdot (\varepsilon_{dis} + \varepsilon_{score}) \right),$$

*where*

$$\varepsilon_{dis} := \left( \widetilde{L} \alpha + \frac{\|A\|^2}{\eta^2} \right) \left( h\widetilde{L}\alpha R + \frac{h \|A\|^2 R + h\|A\| \sqrt{m}\eta}{\eta^2} + \sqrt{dh} \right).$$

**Lemma E.4** (Main Analysis Lemma for Algorithm 1). *Let $\rho = \frac{\|A\|}{\eta \sqrt{\alpha}}$. For all $0 < \varepsilon, \delta < 1$, there exists*

$$K \leq \widetilde{O} \left( \frac{1}{\varepsilon \delta} \left( \frac{\rho^2 \left( (m^2 \rho^4 + 1) \widetilde{r}^2 + m^3 \rho^2 + dm \right)}{\sqrt{m}} + \frac{m}{d} + \log d \right) \right)$$

such that: suppose distribution $p$ is a $(\frac{\varepsilon}{K^2}, \widetilde{r}/\sqrt{\alpha}, R, \widetilde{L}, \alpha)$ mode-centered locally well-conditioned distribution with $R \geq \frac{\sqrt{K\sqrt{m}/\alpha}}{\rho}$, and $\varepsilon_{score} \leq \frac{\sqrt{\alpha/m}}{K^2\delta}$; then *Algorithm 1* samples from a distribution $\widehat{p}(x \mid y)$ such that

$$\Pr_y [\mathrm{TV}(\widehat{p}(x \mid y), p(x \mid y)) \leq \varepsilon] \geq 1 - \delta.$$

*Furthermore, the total iteration complexity can be bounded by*

$$\widetilde{O}\left(K^3(K^2 m^2 d + m^3\rho + m^{1.5}(m\rho^2 + 1)\widetilde{r})(\widetilde{L} + \rho^2)^2 + K^3 m^2\rho(\widetilde{L} + \rho^2)\right).$$

*Proof.* To distinguish the $\varepsilon$ and $\delta$ in the lemma and the one in Corollary E.3, we will use $\varepsilon_{error}$ and $\delta_{error}$ to denote the $\varepsilon$ and $\delta$ in our lemma statement. We need to set parameters in Corollary E.3. For any given $0 < \delta_{error}, \varepsilon_{error}$, we set

$$\varepsilon = \frac{1}{\lambda\delta_{error}}, \qquad \delta = \frac{\varepsilon_{error}}{\lambda^2},$$

and we set $\lambda$ to be the minimum $\lambda$ that satisfies

$$\rho^2\sqrt{m}\log(\lambda/\varepsilon) + \frac{\rho^2\alpha R^2}{\sqrt{m}} + \frac{m^2}{m\log(\lambda/\varepsilon) + \alpha R^2} + \log\left(2 + \frac{\lambda\sqrt{d}\rho}{\varepsilon}\right) \leq \lambda\delta_{error}\varepsilon_{error}.$$

Now we verify the correctness. Taking in the bound for $N$ in Lemma D.4, we have

$$N \lesssim \rho^2\sqrt{m}\log(\lambda/\varepsilon) + \frac{\rho^2\alpha R^2}{\sqrt{m}} + \frac{m^2}{m\log(\lambda/\varepsilon) + \alpha R^2} + \log\left(2 + \frac{\lambda\sqrt{d}\rho}{\varepsilon}\right) \leq \lambda\delta_{error}\varepsilon_{error}.$$

By the setting of our parameters, we have $N\varepsilon \lesssim \varepsilon_{error}$, $\lambda\delta \lesssim \varepsilon_{error}$, and $N/\lambda\varepsilon_{error} \lesssim \delta_{error}$. This guarantees that

$$\Pr_y\left[\mathrm{TV}(\widetilde{X}_N, p(x \mid y)) \lesssim \varepsilon_{error} + \lambda N\sqrt{\frac{m + \log(\lambda/\varepsilon)}{\alpha}} \cdot (\varepsilon_{dis} + \varepsilon_{score})\right] \geq 1 - \delta_{error}.$$

It is easy to verify our bound on $R$ satisfies the condition in Corollary E.3. Note that if a distribution is $(\delta, r, R, \widetilde{L}, \alpha)$ mode-centered locally well-conditioned, then it is also $(\delta, r, R', \widetilde{L}, \alpha)$ mode-centered locally well-conditioned for any $R' \leq R$. Therefore, we can set $R$ to be the minimum $R$ that satisfies the condition.

$$\begin{aligned}
\lambda &= \widetilde{O}\left(\frac{1}{\varepsilon_{error}\delta_{error}}\left(\rho^2\sqrt{m} + \frac{\rho^2\alpha R^2}{\sqrt{m}} + \frac{m^2}{m + \alpha R^2} + \log d\right)\right) \\
&= \widetilde{O}\left(\frac{1}{\varepsilon_{error}\delta_{error}}\left(\frac{\rho^2\left((m^2\rho^4 + 1)\widetilde{r}^2 + m^3\rho^2 + dm\right)}{\sqrt{m}} + \frac{m}{d} + \log d\right)\right) \\
&\lesssim K.
\end{aligned}$$

Therefore, we only need $\lambda N\sqrt{\frac{m+\log(\lambda/\varepsilon)}{\alpha}}(\varepsilon_{dis} + \varepsilon_{score}) \lesssim \varepsilon_{error}$. This can be satisfied when

$$\varepsilon_{dis} + \varepsilon_{score} \lesssim \frac{1}{\lambda^2\delta_{error}}\sqrt{\frac{\alpha}{\log(\lambda/\varepsilon) + m}} \lesssim \frac{\sqrt{\alpha/m}}{K^2\delta_{error}}.$$

Recall that

$$\begin{aligned}
\varepsilon_{dis} &= \left(\widetilde{L}\alpha + \frac{\|A\|^2}{\eta^2}\right)\left(h\widetilde{L}\alpha R + \frac{h\|A\|^2 R + h\|A\|\sqrt{m}\eta}{\eta^2} + \sqrt{dh}\right) \\
&\leq \alpha\left(\widetilde{L} + \rho^2\right)\left(h\widetilde{L}\alpha R + h\rho^2\alpha R + h\rho\sqrt{m\alpha} + \sqrt{dh}\right).
\end{aligned}$$

Therefore, we need to set

$$h = \widetilde{\Omega}\left(\min\left\{\frac{1}{K^2\delta_{error}}\frac{\sqrt{\alpha/m}}{\alpha(\widetilde{L} + \rho^2)[\alpha R(\widetilde{L} + \rho^2) + \rho\sqrt{m\alpha}]}, \frac{1}{K^4\delta_{error}^2\alpha md(\widetilde{L} + \rho^2)^2}\right\}\right).$$

Note that the bound for the sum of $N$ mixing times can be bounded by

$$\sum_{i=1}^{N-1} T_i \lesssim \frac{N(\log(\lambda/\varepsilon) + m)}{\alpha} \leq \widetilde{O}\left(\frac{Km\delta_{error}\varepsilon_{error}}{\alpha}\right).$$

Therefore, the total iteration complexity is bounded by $\widetilde{O}(\frac{Km\delta_{error}\varepsilon_{error}}{\alpha h})$,

$$\widetilde{O}\left(K^3 m(\widetilde{L} + \rho^2)[\alpha R(\widetilde{L} + \rho^2) + \rho\sqrt{m\alpha}]\sqrt{m/\alpha}\varepsilon_{error}^2\delta_{error} + K^5 m^2 d(\widetilde{L} + \rho^2)^2\varepsilon_{error}\delta_{error}^3\right).$$

We can relax it and make the bound be

$$\widetilde{O}\left(K^3(K^2 m^2 d + \sqrt{m^3\alpha}R)(\widetilde{L} + \rho^2)^2 + K^3 m^2\rho(\widetilde{L} + \rho^2)\right).$$

Take in $R$, and we have

$$\widetilde{O}\left(K^3(K^2 m^2 d + m^3\rho + m^{1.5}(m\rho^2 + 1)\widetilde{r})(\widetilde{L} + \rho^2)^2 + K^3 m^2\rho(\widetilde{L} + \rho^2)\right).$$

$\square$

## E.1 Application on Strongly Log-concave Distributions

By Lemma B.12, any $\alpha$-strongly log-concave distribution that has $L$-Lipschitz score is locally well-conditioned distribution $p$ is $(\delta, 2\sqrt{\frac{d}{\alpha}} + \sqrt{\frac{2\log(1/\delta)}{\alpha}}, \infty, L/\alpha, \alpha)$ mode-centered locally well-conditioned. Therefore, take this into Lemma E.4, we have the following result.

**Lemma E.5.** *Let $p(x)$ be an $\alpha$-strongly log-concave distribution over $\mathbb{R}^d$ with $L$-Lipschitz score. Let $\rho = \frac{\|A\|}{\eta\sqrt{\alpha}}$. For all $0 < \varepsilon, \delta < 1$, there exists*

$$K \leq \widetilde{O}\left(\frac{1}{\varepsilon\delta}\left(\frac{\rho^2\left((m^2\rho^4 + m)d + m^3\rho^2\right)}{\sqrt{m}} + \frac{m}{d} + \log d\right)\right)$$

*such that: suppose $\varepsilon_{score} \leq \frac{\sqrt{\alpha/m}}{K^2\delta}$, then Algorithm 1 samples from a distribution $\widehat{p}(x \mid y)$ such that*
$$\Pr_y[\mathrm{TV}(\widehat{p}(x \mid y), p(x \mid y)) \leq \varepsilon] \geq 1 - \delta.$$

*Furthermore, the total iteration complexity can be bounded by*

$$\widetilde{O}\left(K^3(K^2 m^2 d + m^3\rho + m^{1.5}(m\rho^2 + 1)\sqrt{d})(L/\alpha + \rho^2)^2 + K^3 m^2\rho(L/\alpha + \rho^2)\right).$$

To enhance clarity, we state our result in terms of expectation and established the following theorem:

**Theorem E.6** (Posterior sampling with global log-cancavity). *Let $p(x)$ be an $\alpha$-strongly log-concave distribution over $\mathbb{R}^d$ with $L$-Lipschitz score. Let $\rho = \frac{\|A\|}{\eta\sqrt{\alpha}}$. For all $0 < \varepsilon < 1$, there exists*

$$K \leq \widetilde{O}\left(\frac{1}{\varepsilon^2}\left(\frac{\rho^2\left((m^2\rho^4 + m)d + m^3\rho^2\right)}{\sqrt{m}} + \frac{m}{d} + \log d\right)\right)$$

*such that: suppose $\varepsilon_{score} \leq \frac{\sqrt{\alpha/m}}{K^2\varepsilon}$, then Algorithm 1 samples from a distribution $\widehat{p}(x \mid y)$ such that*
$$\mathbb{E}_y[\mathrm{TV}(\widehat{p}(x \mid y), p(x \mid y))] \leq \varepsilon.$$

*Furthermore, the total iteration complexity can be bounded by*

$$\widetilde{O}\left(K^3(K^2 m^2 d + m^3\rho + m^{1.5}(m\rho^2 + 1)\sqrt{d})(L/\alpha + \rho^2)^2 + K^3 m^2\rho(L/\alpha + \rho^2)\right).$$

This gives Theorem 1.1.

**Theorem 1.1** (Posterior sampling with global log-concavity)**.** *Let $p(x)$ be an $\alpha$-strongly log-concave distribution over $\mathbb{R}^d$ with $L$-Lipschitz score. For any $0 < \varepsilon < 1$, there exist $K_1 = \mathrm{poly}(d, m, \frac{\|A\|}{\eta\sqrt{\alpha}}, \frac{1}{\varepsilon})$ and $K_2 = \mathrm{poly}(d, m, \frac{\|A\|}{\eta\sqrt{\alpha}}, \frac{1}{\varepsilon}, \frac{L}{\alpha})$ such that: if $\varepsilon_{score} \leq \frac{\sqrt{\alpha}}{K_1}$, then there exists an algorithm that takes $K_2$ iterations to sample from a distribution $\widehat{p}(x \mid y)$ with*

$$\mathbb{E}\left[\mathrm{TV}(\widehat{p}(x \mid y), p(x \mid y))\right] \leq \varepsilon.$$

**Remark E.7.** *The analysis above is restricted to strongly log-concave distributions, where $\nabla^2 \log p(x) \prec 0$. However, this directly implies that we can use our algorithm to perform posterior sampling on log-concave distributions, for which $\nabla^2 \log p(x) \preceq 0$.*

*Specifically, for any log-concave distribution $p$, we can define a distribution $q(x) \propto p(x) \cdot \exp\left(-\frac{\varepsilon^2 \|x - \theta\|^2}{2m_2^2}\right)$, where $\theta$ is the mode of $p$ and $m_2^2$ is the variance of $p$. It is straightforward to verify that $\mathrm{TV}(p, q) \lesssim \varepsilon$, and $q$ is $(\varepsilon^2/m_2^2)$-strongly log-concave. Therefore, by sampling from $q(x \mid y)$, we can approximate $p(x \mid y)$, incurring an additional expected TV error of $\varepsilon$.*

## E.2 Gaussian Measurement

In this section, we prove Theorem 1.2. In Algorithm 2, we describe how to make Algorithm 1 work on the Gaussian case.

We first verify that suppose Assumption 1 holds, we can also have $L^4$-accurate estimates for the smoothed scores of $p_{x_0}$, so this satisfies the requirement of running Algorithm 1. We need to use the following lemma, with proof deferred to Section E.5.

**Lemma E.8.** *Let $X$, $Y$, and $Z$ be random vectors in $\mathbb{R}^d$, where $Y = X + N(0, \sigma_1^2 I_d)$ and $Z = X + N(0, \sigma_2^2 I_d)$. The conditional density of $Z$ given $Y$, denoted $p(Z \mid Y)$, is a multivariate normal distribution with mean*

$$\mu_{Z|Y} = \sigma_2^2(\sigma_1^2 + \sigma_2^2)^{-1} Y$$

*and covariance matrix*

$$\Sigma_{Z|Y} = \sigma_2^2(\sigma_1^2 + \sigma_2^2)^{-1} \sigma_1^2.$$

*Then, the gradient of the log-likelihood $\log p(Z \mid Y)$ with respect to $Y$ is given by*

$$\nabla_Y \log p(Z \mid Y) = -\frac{1}{\sigma_1^2}\left(Z - \sigma_2^2(\sigma_1^2 + \sigma_2^2)^{-1} Y\right).$$

Using this, we can calculate the *smoothed* conditional score given $x_0$:

**Lemma E.9.** *For any smoothing level $t \geq 0$, suppose we have score estimate $\widehat{s}_{t^2}(x)$ of the smoothed distributions $p_{t^2}(x) = p(x) * \mathcal{N}(0, t^2 I_d)$ that satisfies*

$$\mathbb{E}_{p_{t^2}(x)}[\|\widehat{s}_{t^2}(x) - s_{t^2}(x)\|^4] \leq \varepsilon_{score}^4.$$

*Then we can calculate a score estimate $\widehat{s}_{x_0,t^2}(x)$ of the distribution $p_{x_0,t^2}(x) = p_{x_0}(x) * \mathcal{N}(0, t^2 I_d)$ such that*

$$\mathbb{E}_{x_0}\left[\mathbb{E}_{p_{x_0,t^2}(x)}[\|\widehat{s}_{x_0,t^2}(x) - s_{x_0,t^2}(x)\|^4]\right] \leq \varepsilon_{score}^4.$$

*Proof.* Let $x^{(t)} \sim p_{t^2}$. Then, for any value of $x^{(t)}$, we have

$$\begin{aligned}
s_{x_0,t^2}(x^{(t)}) &= \nabla_{x^{(t)}} \log p(x^{(t)} \mid x_0) \\
&= \nabla_{x^{(t)}} \log p(x^{(t)}) + \nabla_{x^{(t)}} \log p(x_0 \mid x^{(t)}) \\
&= s_{t^2}(x^{(t)}) + \nabla_{x^{(t)}} \log p(x_0 \mid x^{(t)}).
\end{aligned}$$

Note that the second term is exactly in the form of Lemma E.8, so we can calculate this exaclty. For the first term, we use our score estimate $\widehat{s}_{t^2}(x^{(t)})$ for it. In this way, we have that for any $x$,

$$\|\widehat{s}_{x_0,t^2}(x) - s_{x_0,t^2}(x)\| = \|\widehat{s}_{t^2}(x) - s_{t^2}(x)\|.$$

Therefore,

$$\mathbb{E}_{x_0}\left[\mathbb{E}_{p_{x_0,t^2}(x)}[\|\widehat{s}_{x_0,t^2}(x) - s_{x_0,t^2}(x)\|^4]\right] = \mathbb{E}_{p_{t^2}(x)}[\|\widehat{s}_{x_0,t^2}(x) - s_{x_0,t^2}(x)\|^4] \leq \varepsilon_{score}^4.$$

$\square$

Applying Markov's inequality, we have:

**Corollary E.10.** *Suppose Assumption 1 holds for our prior distribution p. Then with $1 - \delta$ probability over $x_0$: we have smoothed score estimates for $p_{x_0}$ with $L^4$ error bounded by $\varepsilon_{score}^4/\delta$; in other words, Assumption 1 holds for $p_{x_0}$, where $\varepsilon_{score}$ is substituted with $\varepsilon_{score}/\delta^{1/4}$.*

To capture the behavior of a Gaussian measurement more accurately, we first define a relaxed version of mode-centered locally well-conditioned distribution.

**Definition E.11.** *For $\delta \in [0, 1)$ and $R, \widetilde{L}, \alpha \in (0, +\infty]$, we say that a distribution p is $(\delta, r, R, \widetilde{L}, \alpha)$ locally well-conditioned if there exists $\theta$ such that*

- $\Pr_{x \sim p}[x \in B(\theta, r)] \geq 1 - \delta$.

- *For $x, y \in B(\theta, R)$, we have that $\|s(x) - s(y)\| \leq \widetilde{L}\alpha \|x - y\|$.*

- *For $x, y \in B(\theta, R)$, we have that $\langle s(y) - s(x), x - y \rangle \geq \alpha \|x - y\|^2$.*

Note that this definition can still imply that the distribution is mode-centered local well-conditioned, due to the following fact:

**Lemma E.12.** *Let p be a probability density on $\mathbb{R}^d$. Fix $0 < r < R$ and $\theta \in \mathbb{R}^d$ such that*

$$\Pr_{x \sim p}[x \in B(\theta, r)] \geq 0.9, \qquad \nabla^2(-\log p(x)) \succeq \alpha I_d \quad (x \in B(\theta, R)), \ \alpha > 0.$$

*If $R > 4dr$, then there exists $\theta' \in B(\theta, 4dr)$ with $\nabla \log p(\theta') = 0$.*

We defer its proof to Section E.5. This implies the following lemma:

**Lemma E.13.** *Let p be a $(\delta, r, R, \widetilde{L}, \alpha)$ locally well conditioned distribution with $R > 9dr$ and $\delta < 0.1$. Then p is $(\delta, (4d + 1)r, R - 4dr, \widetilde{L}, \alpha)$ mode-centered locally well conditioned.*

This gives a version of Lemma E.4 for locally well-conditioned distributions as a corollary:

**Lemma E.14.** *Let $\rho = \frac{\|A\|}{\eta\sqrt{\alpha}}$. For all $0 < \varepsilon, \delta < 1$, there exists*

$$K \leq \widetilde{O}\left(\frac{1}{\varepsilon\delta}\left(\frac{\rho^2\left((m^2\rho^4 + 1)d^2\widetilde{r}^2 + m^3\rho^2 + dm\right)}{\sqrt{m}} + \frac{m}{d} + \log d\right)\right)$$

*such that: suppose distribution p is a $(\frac{\varepsilon}{K^2}, \widetilde{r}/\sqrt{\alpha}, R, \widetilde{L}, \alpha)$ mode-centered locally well-conditioned distribution with $R \geq \frac{\sqrt{K\sqrt{m}/\alpha}}{\rho}$, and $\varepsilon_{score} \leq \frac{\sqrt{\alpha/m}}{K^2\delta}$. Then Algorithm 1 samples from a distribution $\widehat{p}(x \mid y)$ such that*

$$\Pr_y[\mathrm{TV}(\widehat{p}(x \mid y), p(x \mid y)) \leq \varepsilon] \geq 1 - \delta.$$

*Furthermore, the total iteration complexity can be bounded by*

$$\widetilde{O}\left(K^3(K^2m^2d + m^3\rho + m^{1.5}(m\rho^2 + 1)\widetilde{r})(\widetilde{L} + \rho^2)^2 + K^3m^2\rho(\widetilde{L} + \rho^2)\right).$$

The reason we want this relaxed notion of locally well-conditioned is that, this captures the behavior of a Gaussian measurement. First note that:

**Lemma E.15.** *Let p be a distribution on $\mathbb{R}^d$. Let $\widetilde{x} = x_{true} + N(0, \sigma^2 I_d)$ be a Gaussian measurement of $x_{true} \sim p$. Let $p_{\widetilde{x}}(x)$ be the posterior distribution of x given $\widetilde{x}$. Then, for any $\delta \in (0, 1)$ and $\delta' \in (0, 1)$, with probability at least $1 - \delta'$ over $\widetilde{x}$,*

$$\Pr_{x \sim p_{\widetilde{x}}}[x \in B(\widetilde{x}, r)] \geq 1 - \delta$$

*for $r = \sigma(\sqrt{d} + \sqrt{2\log\frac{1}{\delta\delta'}})$.*

---

**Algorithm 2** Sampling from $p(x \mid x_0, y)$ given an extra Gaussian measurement $x_0$

---

1: **function** GAUSSIANSAMPLER($p : \mathbb{R}^d \to \mathbb{R}$, $x_0 \in \mathbb{R}^d$ , $y \in \mathbb{R}^m$, $A \in \mathbb{R}^{m \times d}$, $\eta, \sigma \in \mathbb{R}$)

2:      Let $p_{x_0}(x) := p(x \mid x + \mathcal{N}(0, \sigma^2 I_d) = x_0)$.

3:      Use Algorithm 1, return

$$\text{POSTERIORSAMPLER}(p_{x_0}, y, A, \eta).$$

4: **end function**

---

Again, we defer its proof to Section E.5. This implies the following lemma.

**Lemma E.16.** *For $\delta \in (0, 1)$, suppose $p$ is a distribution over $\mathbb{R}^d$ such that*

$$\Pr_{x' \sim p} \left[ \forall x \in B(x', R) : -L I_d \preceq \nabla^2 \log p(x) \preceq (\tau^2/R^2) I_d \right] \geq 1 - \delta.$$

*Given a Gaussian measurement $x_0 = x + \mathcal{N}(0, \sigma^2 I_d)$ of $x \sim p$ with*

$$\sigma \leq \frac{R}{2\sqrt{d} + \sqrt{2 \log(1/\delta)} + 2\tau}.$$

*Let $x_0 = x + N(0, \sigma^2 I_d)$, where $x \sim p$. Then, suppose $R$. with probability at least $1 - 3\delta$ probability over $x_0$, $p_{x_0}$ is $(\delta, \sigma(\sqrt{d} + \sqrt{4 \log \frac{1}{\delta}}), R/2, 2L\sigma^2 + 2, \frac{1}{2\sigma^2})$ locally well-conditioned.*

*Proof.* Let us check the locally well-conditioned conditions with $\theta = x_0$ one by one. The concentration follows directly from Lemma E.15, incurring an error probability of $\delta$.

By our choice of $\sigma$, we have that

$$\Pr \left[ \|x_0 - x\| \leq \frac{R}{2} \right] \geq 1 - \delta.$$

Therefore,

$$\Pr \left[ \forall x \in B(x_0, R/2) : -L I_d \preceq \nabla^2 \log p(x) \preceq (\tau^2/R^2) I_d \right] \geq 1 - 2\delta.$$

By direct calculation, we have that

$$-L I_d \preceq \nabla^2 \log p(x) \preceq (\tau^2/R^2) I_d \implies -(L + 1/\sigma^2) I_d \preceq \nabla^2 \log p(x) \preceq (\tau^2/R^2 - \frac{1}{\sigma^2}) I_d$$

By our choice of $\sigma$, we have that whenever $-L I_d \preceq \nabla^2 \log p(x) \preceq (\tau^2/R^2) I_d$,

$$-(2L\sigma^2 + 2)\frac{1}{2\sigma^2} I_d \preceq \nabla^2 \log p(x) \preceq -\frac{1}{2\sigma^2} I_d$$

This satisfies the Lipschitzness and the strong log-concavity condition by giving an additional error probability of $2\delta$. $\qquad\square$

This gives us the main lemma for our local log-concavity case:

**Lemma E.17.** *For any $\delta, \varepsilon, \tau, \sigma, R, L > 0$, suppose $p(x)$ is a distribution over $\mathbb{R}^d$ such that*

$$\Pr_{x' \sim p} \left[ \forall x \in B(x', R) : -L I_d \preceq \nabla^2 \log p(x) \preceq (\tau^2/R^2) I_d \right] \geq 1 - \delta.$$

*Let $\rho = \frac{\|A\|\sigma}{\eta}$. There exists*

$$K \leq \widetilde{O} \left( \frac{1}{\varepsilon \delta} \left( \frac{\rho^2 \left( (m^2 \rho^4 + 1) d^3 + m^3 \rho^2 + dm \right)}{\sqrt{m}} + \frac{m}{d} + \log d \right) \right).$$

*such that: suppose $R^2 \geq (\frac{K\sqrt{m}}{\rho^2} + 4\tau)\sigma^2$ and $\varepsilon_{score} \leq \frac{1}{K^2 \sqrt{m}\sigma}$, then Algorithm 2 samples from a distribution $\widehat{p}(x \mid x_0, y)$ such that*

$$\Pr_{x_0, y} \left[ \text{TV}(\widehat{p}(x \mid x_0, y), p(x \mid x_0, y)) \leq \varepsilon \right] \geq 1 - O(\delta).$$

*Furthermore, the total iteration complexity can be bounded by*

$$\widetilde{O} \left( K^3 (K^2 m^2 d + m^3 \rho + m^{1.5} (m\rho^2 + 1)\sqrt{d})(L\sigma^2 + \rho^2 + 1)^2 + K^3 m^2 \rho (L\sigma^2 + \rho^2 + 1) \right).$$

**Algorithm 3** Competitive Compressed Sensing Algorithm Given a Rough Estimation

---

1: **function** COMPRESSEDSENSING($p : \mathbb{R}^d \to \mathbb{R}$, $x_0 \in \mathbb{R}^d$, $y \in \mathbb{R}^m$, $A \in \mathbb{R}^{m \times d}$, $\eta, R \in \mathbb{R}$)
2:     Let $\sigma = R/\delta$.
3:     Sample $x_0' = x_0 + \mathcal{N}(0, \sigma^2 I_d)$.
4:     Use Algorithm 2, return

$$\text{GAUSSIANSAMPLER}(p, x_0', y, A, \eta, \sigma)$$

5: **end function**

---

*Proof.* Combining Corollary E.10 with Lemma E.16 enables us to apply Lemma E.14 and proves the lemma. $\qquad\square$

Expressing this in expectation, we have the following theorem.

**Theorem E.18** (Posterior sampling with local log-concavity). *For any $\varepsilon, \tau, R, L > 0$, suppose $p(x)$ is a distribution over $\mathbb{R}^d$ such that*

$$\Pr_{x' \sim p} \left[ \forall x \in B(x', R) : -L I_d \preceq \nabla^2 \log p(x) \preceq (\tau^2/R^2) I_d \right] \geq 1 - \varepsilon.$$

*Let $\rho = \frac{\|A\|\sigma}{\eta}$. There exists*

$$K \leq \widetilde{O}\left( \frac{1}{\varepsilon\delta} \left( \frac{\rho^2 \left( \left(m^2\rho^4 + 1\right) d^3 + m^3\rho^2 + dm \right)}{\sqrt{m}} + \frac{m}{d} + \log d \right) \right).$$

*such that: given a Gaussian measurement $x_0 = x + \mathcal{N}(0, \sigma^2 I_d)$ of $x \sim p$ with $R^2 \geq (\frac{K\sqrt{m}}{\rho^2} + 4\tau)\sigma^2$, and $\varepsilon_{score} \leq \frac{1}{K^2\sqrt{m}\sigma}$; then Algorithm 2 samples from a distribution $\widehat{p}(x \mid x_0, y)$ such that*

$$\mathbb{E}_{x_0, y} \left[ \text{TV}(\widehat{p}(x \mid x_0, y), p(x \mid x_0, y)) \right] \lesssim \varepsilon.$$

*Furthermore, the total iteration complexity can be bounded by*

$$\widetilde{O}\left( K^3(K^2 m^2 d + m^3\rho + m^{1.5}(m\rho^2 + 1)\sqrt{d})(L\sigma^2 + \rho^2 + 1)^2 + K^3 m^2 \rho (L\sigma^2 + \rho^2 + 1) \right).$$

This gives us Theorem 1.2:

**Theorem 1.2** (Posterior sampling with local log-concavity). *For any $\varepsilon, \tau, R, L > 0$, suppose $p(x)$ is a distribution over $\mathbb{R}^d$ such that*

$$\Pr_{x' \sim p} \left[ \forall x \in B(x', R) : -L I_d \preceq \nabla^2 \log p(x) \preceq (\tau^2/R^2) I_d \right] \geq 1 - \varepsilon.$$

*Then, there exist $K_1, K_2 = \text{poly}(d, m, \frac{\|A\|\sigma}{\eta}, \frac{1}{\varepsilon})$ and $K_3 = \text{poly}(d, m, \frac{\|A\|\sigma}{\eta}, \frac{1}{\varepsilon}, L\sigma^2)$ such that: Given a Gaussian measurement $x_0 = x + \mathcal{N}(0, \sigma^2 I_d)$ of $x \sim p$ with $\sigma \leq R/(K_1 + 2\tau)$. If $\varepsilon_{score} \leq \frac{1}{K_2\sigma}$, then there exists an algorithm that takes $K_3$ iterations to sample from a distribution $\widehat{p}(x \mid x_0, y)$ such that*

$$\mathbb{E}_{y, x_0} \left[ \text{TV}(\widehat{p}(x \mid x_0, y), p(x \mid x_0, y)) \right] \lesssim \varepsilon.$$

### E.3   Compressed Sensing

In this section, we prove Corollary 1.3. We first describe the sampling procedure in Algorithm 3. Now we verify its correctness.

**Lemma E.19.** *For any $\delta, \tau, R, R', L > 0$, suppose $p(x)$ is a distribution over $\mathbb{R}^d$ such that*

$$\Pr_{x' \sim p} \left[ \forall x \in B(x', R') : -L I_d \preceq \nabla^2 \log p(x) \preceq (\tau/R')^2 I_d \right] \geq 1 - \delta.$$

*Let $\rho = \frac{\|A\|R}{\eta}$. There exists*

$$K \leq \widetilde{O}\left( \frac{1}{\delta^2} \left( \frac{\rho^2 \left( \left(m^2\rho^4 + 1\right) d^3 + m^3\rho^2 + dm \right)}{\sqrt{m}} + \frac{m}{d} + \log d \right) \right).$$

*such that: suppose $(R')^2 \geq (\frac{K\sqrt{m}}{\rho^2} + 4\tau)R^2$ and $\varepsilon_{score} \leq \frac{1}{K^2\sqrt{m}R}$, then conditioned on $\|x_0 - x\| \leq R$, Algorithm 3 of Algorithm 3 samples from a distribution $\widehat{p}$ (depending on $x'_0$ and $y$) such that*

$$\Pr_{x'_0, y} \left[ \mathrm{TV}(\widehat{p}, p(x \mid x + \mathcal{N}(0, \sigma^2 I_d) = x'_0, Ax + \xi = y)) \leq \delta \right] \geq 1 - O(\delta).$$

*Furthermore, the total iteration complexity can be bounded by*

$$\widetilde{O}\left(K^3(K^2 m^2 d + m^3 \rho + m^{1.5}(m\rho^2 + 1)\sqrt{d})(L\sigma^2 + \rho^2 + 1)^2 + K^3 m^2 \rho (L\sigma^2 + \rho^2 + 1)\right).$$

*Proof.* This is a direct application of Lemma E.17. The sole difference is that $x'_0$ follows $x_0 + \mathcal{N}(0, \sigma^2 I_d)$ instead of $x + \mathcal{N}(0, \sigma^2 I_d)$. Because $\|x_0 - x\| \leq R$, $x'_0$ remains sufficiently close to $x$ for the local Hessian condition to hold, so the proof of Lemma E.17 carries over verbatim. $\square$

Now we explain why we want to sample from $p(x \mid x + \mathcal{N}(0, \sigma^2 I_d) = x'_0, Ax + \xi = y)$. Essentially, the extra Gaussian measurement won't hurt the concentration of $p(x \mid y)$ itself. We abstract it as the following lemma:

**Lemma E.20.** *Let $(X, Y)$ be jointly distributed random variables with $X \in \mathbb{R}^d$. Assume that for some $r > 0$ and $0 < \delta < 1$*

$$\Pr_{Y, \widehat{X} \sim p(X|Y)}[\|X - \widehat{X}\| \leq r] \geq 1 - \delta.$$

*Define $Z = X + \varepsilon$ where $\varepsilon \sim \mathcal{N}(0, \sigma^2 I_d)$ is independent of $(X, Y)$. If*

$$\sigma \geq \frac{r}{2\delta},$$

*then for $\widehat{X} \sim p(X \mid Y, Z)$ one has*

$$\Pr_{Y, Z, \widehat{X}}[\|X - \widehat{X}\| \leq r] \geq 1 - 3\delta.$$

*Proof.* Fix $Y$ and draw an auxiliary point $\widetilde{X} \sim p(X \mid Y)$. Let $Z' = \widetilde{X} + \varepsilon'$ with $\varepsilon' \sim \mathcal{N}(0, \sigma^2 I_d)$ independent of everything else. On the event

$$E = \{\|X - \widetilde{X}\| \leq r\},$$

$Z$ and $Z'$ are Gaussians with the same covariance $\sigma^2 I_d$ and means $X$ and $\widetilde{X}$. Pinsker's inequality combined with the KL divergence between the two Gaussians gives

$$\mathrm{TV}(\mathcal{N}(X, \sigma^2 I_d), \mathcal{N}(\widetilde{X}, \sigma^2 I_d)) \leq \frac{\|X - \widetilde{X}\|}{2\sigma} \leq \frac{r}{2\sigma} \leq \delta.$$

Hence

$$\mathrm{TV}(\mathcal{L}(Y, X, Z), \mathcal{L}(Y, X, Z')) \leq \Pr[E^c] + \delta \leq 2\delta,$$

because $\Pr[E^c] \leq \delta$ by the hypothesis on $p(X \mid Y)$.

By construction,

$$p(X \mid Y) = \mathbb{E}_{Z'|Y}[p(X \mid Y, Z')],$$

so

$$\Pr_{Y, Z', \widehat{X} \sim p(X|Y, Z')}[\|X - \widehat{X}\| \leq r] \geq 1 - \delta.$$

For the set $A = \{(Y, Z, \widehat{X}) : \|X - \widehat{X}\| > r\}$ the total-variation bound gives

$$\left| \Pr_{Y, Z, \widehat{X}}[A] - \Pr_{Y, Z', \widehat{X}}[A] \right| \leq 2\delta,$$

whence

$$\Pr_{Y, Z, \widehat{X}}[\|X - \widehat{X}\| \leq r] \geq 1 - \delta - 2\delta = 1 - 3\delta. \qquad \square$$

This implies the following lemma:

**Lemma E.21.** *Consider the random variables in [Algorithm 3](#). Suppose that*

- *Information theoretically, it is possible to recover $\widehat{x}$ from $y$ satisfying $\|\widehat{x} - x\| \le r$ with probability $1 - \delta$ over $x \sim p$ and $y$.*

- $\Pr\left[\|x_0 - x\| \le R\right] \ge 1 - \delta.$

*Then drawing sample $\widehat{x} \sim p(x \mid x + \mathcal{N}(0, \sigma^2 I_d) = x_0', Ax + \xi = y)$ would give that*

$$\Pr\left[\|x - \widehat{x}\| \le 2r\right] \ge 1 - O(\delta).$$

*Proof.* By [JAD$^+$21], the first condition implies that,

$$\Pr_{x,y,\widehat{x} \sim p(x|y)}\left[\|x - \widehat{x}\| \le 2r\right] \ge 1 - 2\delta.$$

Then by [Lemma E.20](#), suppose we have $x' = x + \mathcal{N}(0, \sigma^2 I_d)$, then

$$\Pr_{x,y,\widehat{x} \sim p(x|y,x+\mathcal{N}(0,\sigma^2 I_d)=x')}\left[\|x - \widehat{x}\| \le 2r\right] \ge 1 - 6\delta.$$

Note that whenever $\|x - x_0\| \le r$, we have

$$\mathrm{TV}(x' \mid x, x_0, x_0' \mid x, x_0) \le \delta.$$

This proves that

$$\Pr_{x,y,\widehat{x} \sim p(x|y,x+\mathcal{N}(0,\sigma^2 I_d)=x_0')}\left[\|x - \widehat{x}\| \le 2r\right] \ge 1 - 6\delta.$$

$\square$

**Lemma E.22.** *Consider attempting to accurately reconstruct $x$ from $y = Ax + \xi$. Suppose that:*

- *Information theoretically, it is possible to recover $\widehat{x}$ from $y$ satisfying $\|\widehat{x} - x\| \le r$ with probability $1 - \delta$ over $x \sim p$ and $y$.*

- *We have access to a "naive" algorithm that recovers $x_0$ from $y$ satisfying $\|x_0 - x\| \le R$ with probability $1 - \delta$ over $x \sim p$ and $y$.*

*Let $\rho = \frac{\|A\|R}{\eta\delta}$. There exists*

$$K \le \widetilde{O}\left(\frac{1}{\delta^2}\left(\frac{\rho^2\left(\left(m^2\rho^4 + 1\right)d^3 + m^3\rho^2 + dm\right)}{\sqrt{m}} + \frac{m}{d} + \log d\right)\right).$$

*such that: suppose for $R' = (R/\delta) \cdot \sqrt{\frac{K\sqrt{m}}{\rho^2} + 4\tau}$,*

$$\Pr_{x' \sim p}\left[\forall x \in B(x', R') : -LI_d \preceq \nabla^2 \log p(x) \preceq (\tau/R')^2 I_d\right] \ge 1 - \delta.$$

*Then we give an algorithm that recovers $\widehat{x}$ satisfying $\|\widehat{x} - x\| \le 2r$ with probability $1 - O(\delta)$, in $\mathrm{poly}(d, m, \frac{\|A\|R}{\eta}, \frac{1}{\delta})$ time, under Assumption [1](#) with $\varepsilon_{score} < \frac{1}{K^2\sqrt{m}(R/\delta)}$.*

*Proof.* By our assumption and [Lemma E.19](#), we have that we are sampling from $p(x \mid x + \mathcal{N}(0, \sigma^2 I_d) = x_0', Ax + \xi = y)$ with $\delta$ TV error with $1 - O(\delta)$ probability. By [Lemma E.21](#), this would recover $x$ within distance $2r$ with $1 - O(\delta)$ probaility. Combining the two gives the result. $\square$

Setting $\tau = 0$ would give [Corollary 1.3](#) as a corollary.

**Corollary 1.3** (Competitive compressed sensing)**.** *Consider attempting to accurately reconstruct $x$ from $y = Ax + \xi$. Suppose that:*

- *Information theoretically (but possibly requiring exponential time or using exact knowledge of $p(x)$), it is possible to recover $\widehat{x}$ from $y$ satisfying $\|\widehat{x} - x\| \le r$ with probability $1 - \delta$ over $x \sim p$ and $y$.*

- *We have access to a "naive" algorithm that recovers $x_0$ from $y$ satisfying $\|x_0 - x\| \le R$ with probability $1 - \delta$ over $x \sim p$ and $y$.*

- *For $R' = R \cdot \mathrm{poly}(d, m, \frac{\|A\|R}{\eta}, \frac{1}{\delta})$,*

$$\Pr_{x' \sim p} \left[ \forall x \in B(x', R') : -LI_d \preceq \nabla^2 \log p(x) \preceq 0 \right] \ge 1 - \delta.$$

*Then we give an algorithm that recovers $\widehat{x}$ satisfying $\|\widehat{x} - x\| \le 2r$ with probability $1 - O(\delta)$, in $\mathrm{poly}(d, m, \frac{\|A\|R}{\eta}, \frac{1}{\delta})$ time, under Assumption 1 with $\varepsilon_{score} < \frac{1}{\mathrm{poly}(d, m, \frac{\|A\|R}{\eta}, \frac{1}{\delta}, LR^2)R}$.*

### E.4   Ring example

Let $w \in (0, 0.01)$ and let $p_0$ be the uniform probability measure on the unit circle $S^1 = \{x \in \mathbb{R}^2 : \|x\| = 1\}$. Define the *circle–Gaussian mixture*

$$p(x) \;=\; (p_0 * \mathcal{N}(0, w^2 I_2))(x) \;=\; \frac{1}{2\pi} \int_0^{2\pi} \frac{1}{2\pi w^2} \exp\!\Big(-\frac{\|x - (\cos\theta, \sin\theta)\|^2}{2w^2}\Big)\, d\theta, \qquad x \in \mathbb{R}^2.$$

**Lemma E.23.** *For any $x \in \mathbb{R}^2$ with radius $r = \|x\| > 0$, the Hessian of the log–density satisfies*

$$\nabla^2 \log p(x) \;\preceq\; \begin{cases} (\dfrac{1}{2w^4} - \dfrac{1}{w^2})I_2, & 0 < r \le w^2, \\[2mm] (\dfrac{1}{w^2 r} - \dfrac{1}{w^2})I_2, & w^2 < r \le 1, \\[2mm] 0, & r > 1. \end{cases}$$

*Proof.* Rotational invariance gives $p(x) = p(r)$ with

$$p(r) = \frac{1}{2\pi w^2} \exp\!\Big(-\frac{r^2 + 1}{2w^2}\Big) I_0\!\Big(\frac{r}{w^2}\Big), \qquad r \ge 0.$$

Write $f(r) = \log p(r)$ and set $z = r/w^2 > 0$. Using $I_0'(z) = I_1(z)$, we get the first and second derivatives:

$$f'(r) = \frac{-r + I_1(z)/I_0(z)}{w^2}, \qquad f''(r) = -\frac{1}{w^2} + \frac{I_0(z)I_2(z) - I_1(z)^2}{w^4 I_0(z)^2}.$$

For $r > 0$, the eigenvalues of $\nabla^2 \log p$ are

$$\lambda_r(r) = f''(r), \qquad \lambda_t(r) = \frac{f'(r)}{r}.$$

The Turán inequality $I_1(z)^2 - I_0(z)I_2(z) \ge 0$ implies $\lambda_r(r) \le -1/w^2$; thus, the largest eigenvalue is $\lambda_t(r)$.

Since $I_1(z)/I_0(z) \le 1$ for all $z > 0$ and $I_1(z)/I_0(z) \le z/2$ for $0 < z \le 1$,

$$\lambda_t(r) = -\frac{1}{w^2} + \frac{1}{w^2 r} \frac{I_1(z)}{I_0(z)} \;\le\; \begin{cases} -\dfrac{1}{w^2} + \dfrac{1}{2w^4}, & 0 < r \le w^2, \\[2mm] -\dfrac{1}{w^2} + \dfrac{1}{w^2 r}, & w^2 < r \le 1, \\[2mm] 0, & r > 1. \end{cases}$$

$\square$

**Lemma E.24.** *For every $x \in \mathbb{R}^2$, we have*

$$\nabla^2 \log p(x) \;\succeq\; -\frac{1}{w^2} I_2.$$

*Proof.* Write $u = (\cos\theta, \sin\theta)$ and

$$p(x) = \frac{1}{2\pi}\int_0^{2\pi}\frac{1}{2\pi w^2}e^{-\|x-u\|^2/(2w^2)}\,d\theta.$$

Differentiating under the integral gives

$$\nabla p(x) = \int\left(-\frac{x-u}{w^2}\right)\frac{1}{2\pi}\frac{1}{2\pi w^2}e^{-\|x-u\|^2/(2w^2)}\,d\theta = -\frac{1}{w^2}\,p(x)\,(x - \mathbb{E}[u \mid x]),$$

so

$$\nabla\log p(x) = -\frac{x - \mathbb{E}[u \mid x]}{w^2}.$$

Differentiating once more,

$$\nabla^2\log p(x) = -\frac{I_2}{w^2} + \frac{1}{w^2}\,\nabla\,\mathbb{E}[u \mid x].$$

A standard score–covariance identity shows

$$\nabla\,\mathbb{E}[u \mid x] = \mathrm{Cov}_{\,u|x}(u, \frac{x-u}{w^2}) = \frac{1}{w^2}\,\mathrm{Cov}_{\,u|x}(u),$$

hence

$$\nabla^2\log p(x) = \frac{\mathrm{Cov}_{\,u|x}(u)}{w^4} - \frac{I_2}{w^2}.$$

Since $\mathrm{Cov}_{\,u|x}(u) \succeq 0$, it follows that

$$\nabla^2\log p(x) \succeq -\frac{1}{w^2}\,I_2,$$

as claimed. $\qquad\square$

**Lemma E.25.** *For any $w \in (0, 1/2)$, we have that*

$$\Pr_{x'\sim p}\left[\forall x \in B(x', 1/2) : -\frac{1}{w^2}I_d \preceq \nabla^2\log p(x) \preceq \frac{1}{2w^2}I_d\right] \geq 1 - e^{-\Omega(1/w^2)}.$$

*Proof.* Note that

$$\Pr_{x\sim p}\left[\|x\| > 3/4\right] \geq 1 - e^{-\Omega(1/w^2)}.$$

The rest follows by combining Lemma E.23 and Lemma E.24. $\qquad\square$

Hence, we can apply Theorem 1.2 on our ring distribution $p$ and get the following corollary:

**Corollary E.26.** *Let $A \in \mathbb{R}^{C\times 2}$ be a matrix for some constant $C > 0$. Consider $x \sim p$ with two measurements given by*

$$x_0 = x + N(0, \sigma^2 I_2) \quad and \quad y = Ax + N(0, \eta^2 I_2).$$

*Suppose $\|A\|w/\eta = O(1)$. Then, if $\sigma \leq cw$ and $\varepsilon_{score} \leq cw^{-1}$ for sufficiently small constant $c > 0$, Algorithm 2 takes a constant number of iterations to sample from a distribution $\widehat{p}(x \mid x_0, y)$ such that*

$$\mathbb{E}_{x_0,y}\left[\mathrm{TV}(\widehat{p}(x \mid x_0, y), p(x \mid x_0, y))\right] < 0.01.$$

## E.5 Deferred Proof

**Lemma E.8.** *Let $X$, $Y$, and $Z$ be random vectors in $\mathbb{R}^d$, where $Y = X + N(0, \sigma_1^2 I_d)$ and $Z = X + N(0, \sigma_2^2 I_d)$. The conditional density of $Z$ given $Y$, denoted $p(Z \mid Y)$, is a multivariate normal distribution with mean*

$$\mu_{Z|Y} = \sigma_2^2(\sigma_1^2 + \sigma_2^2)^{-1}Y$$

*and covariance matrix*

$$\Sigma_{Z|Y} = \sigma_2^2(\sigma_1^2 + \sigma_2^2)^{-1}\sigma_1^2.$$

*Then, the gradient of the log-likelihood $\log p(Z \mid Y)$ with respect to $Y$ is given by*

$$\nabla_Y\log p(Z \mid Y) = -\frac{1}{\sigma_1^2}\left(Z - \sigma_2^2(\sigma_1^2 + \sigma_2^2)^{-1}Y\right).$$

*Proof.* Since $Z \mid Y \sim \mathcal{N}(\mu_{Z|Y}, \Sigma_{Z|Y})$, the log-likelihood function is

$$\log p(Z \mid Y) = -\frac{1}{2} \left( (Z - \mu_{Z|Y})^T \Sigma_{Z|Y}^{-1} (Z - \mu_{Z|Y}) + \log \det(\Sigma_{Z|Y}) + d \log(2\pi) \right).$$

To compute the gradient with respect to $Y$, we focus on the term involving $\mu_{Z|Y}$:

$$-\frac{1}{2} \left( (Z - \mu_{Z|Y})^T \Sigma_{Z|Y}^{-1} (Z - \mu_{Z|Y}) \right).$$

Differentiating with respect to $Y$ gives:

$$\nabla_Y \left[ (Z - \mu_{Z|Y})^T \Sigma_{Z|Y}^{-1} (Z - \mu_{Z|Y}) \right] = -2\Sigma_{Z|Y}^{-1} (Z - \mu_{Z|Y}) \cdot \nabla_Y \mu_{Z|Y}.$$

Since $\mu_{Z|Y} = \sigma_2^2 (\sigma_1^2 + \sigma_2^2)^{-1} Y$, we have

$$\nabla_Y \mu_{Z|Y} = \sigma_2^2 (\sigma_1^2 + \sigma_2^2)^{-1}.$$

Thus, the gradient becomes

$$\nabla_Y \log p(Z \mid Y) = -\Sigma_{Z|Y}^{-1} (Z - \mu_{Z|Y}) \cdot \sigma_2^2 (\sigma_1^2 + \sigma_2^2)^{-1}.$$

Substituting the inverse of the covariance matrix $\Sigma_{Z|Y}$, we get

$$\Sigma_{Z|Y}^{-1} = \frac{1}{\sigma_1^2} \left( \sigma_1^2 + \sigma_2^2 \right),$$

and the final expression for the gradient is

$$\nabla_Y \log p(Z \mid Y) = -\frac{1}{\sigma_1^2} \left( Z - \sigma_2^2 (\sigma_1^2 + \sigma_2^2)^{-1} Y \right).$$

$\square$

**Lemma E.12.** *Let $p$ be a probability density on $\mathbb{R}^d$. Fix $0 < r < R$ and $\theta \in \mathbb{R}^d$ such that*
$$\Pr_{x \sim p} [x \in B(\theta, r)] \geq 0.9, \qquad \nabla^2(-\log p(x)) \succeq \alpha I_d \quad (x \in B(\theta, R)), \ \alpha > 0.$$

*If $R > 4dr$, then there exists $\theta' \in B(\theta, 4dr)$ with $\nabla \log p(\theta') = 0$.*

*Proof.* By Lemma B.13, there is a normalised density $q$ satisfying $\nabla \log q = \nabla \log p$ on $B(\theta, R)$ and such that $\log q$ is $\alpha$-strongly concave on $\mathbb{R}^d$. The difference $\log p - \log q$ is therefore constant on $B(\theta, R)$; hence
$$p(x) = C\, q(x) \qquad (x \in B(\theta, R))$$
for some $C > 0$.

Let $\mu = \arg\max q$; strong concavity gives $\nabla \log q(\mu) = 0$ and uniqueness of $\mu$. Assume for contradiction that $\|\mu - \theta\| \geq 4dr$. Set $\lambda = 2r/\|\mu - \theta\| \leq 1/(2d)$ and define
$$\tau(x) = (1 - \lambda)x + \lambda\mu.$$

Then $\det D\tau = (1 - \lambda)^d$ and $\tau(B(\theta, r)) = B(\theta', (1 - \lambda)r)$ with $\theta' = \tau(\theta) \subset B(\theta, R)$. Along any ray starting at $\mu$ the function $t \mapsto \log q(\mu + t(x - \mu))$ is strictly decreasing for $t \geq 0$; hence $q(\tau(x)) \geq q(x)$ for every $x$.

A change of variables yields
$$\Pr_q[B(\theta', (1 - \lambda)r)] = \int_{B(\theta, r)} q(\tau(x))(1 - \lambda)^d \, dx \geq (1 - \lambda)^d \Pr_q[B(\theta, r)].$$

Because $\lambda \leq 1/(2d)$, $(1 - \lambda)^d \geq e^{-1/2} > 0.6$. Multiplying by $C$ and using $p = Cq$ on $B(\theta, R)$ gives
$$\Pr_p[B(\theta', (1 - \lambda)r)] \geq 0.6 \Pr_p[B(\theta, r)] \geq 0.54.$$

The two balls $B(\theta, r)$ and $B(\theta', (1 - \lambda)r)$ are disjoint, so $1 \geq 0.9 + 0.54$, a contradiction. Thus $\|\mu - \theta\| < 4dr$.

Because $4dr < R$ we have $\mu \in B(\theta, R)$ and here $\nabla \log p = \nabla \log q$; consequently $\nabla \log p(\mu) = 0$. Putting $\theta' = \mu$ completes the proof. $\square$

**Lemma E.15.** *Let $p$ be a distribution on $\mathbb{R}^d$. Let $\widetilde{x} = x_{true} + N(0, \sigma^2 I_d)$ be a Gaussian measurement of $x_{true} \sim p$. Let $p_{\widetilde{x}}(x)$ be the posterior distribution of $x$ given $\widetilde{x}$. Then, for any $\delta \in (0, 1)$ and $\delta' \in (0, 1)$, with probability at least $1 - \delta'$ over $\widetilde{x}$,*

$$\Pr_{x \sim p_{\widetilde{x}}}[x \in B(\widetilde{x}, r)] \geq 1 - \delta$$

*for $r = \sigma(\sqrt{d} + \sqrt{2 \log \frac{1}{\delta \delta'}})$.*

*Proof.* Let $Q(\widetilde{x}) = \Pr_{x \sim p_{\widetilde{x}}}[\|x - \widetilde{x}\| > r]$. We want to show that with probability at least $1 - \delta'$ over $\widetilde{x}$, $Q(\widetilde{x}) \leq \delta$. This is equivalent to showing that $\Pr_{\widetilde{x}}[Q(\widetilde{x}) > \delta] \leq \delta'$.

We use Markov's inequality. For any $\delta > 0$:

$$\Pr_{\widetilde{x}}[Q(\widetilde{x}) > \delta] \leq \frac{\mathbb{E}_{\widetilde{x}}[Q(\widetilde{x})]}{\delta}.$$

Thus, it suffices to show that $\mathbb{E}_{\widetilde{x}}[Q(\widetilde{x})] \leq \delta \delta'$.

Let's compute $\mathbb{E}_{\widetilde{x}}[Q(\widetilde{x})]$:

$$\mathbb{E}_{\widetilde{x}}[Q(\widetilde{x})] = \mathbb{E}_{\widetilde{x}} \left[ \int_{\|x_1 - \widetilde{x}\| > r} p(x_1 \mid \widetilde{x}) dx_1 \right]$$

$$= \int p(\widetilde{x}) \left( \int_{\|x_1 - \widetilde{x}\| > r} p(x_1 \mid \widetilde{x}) dx_1 \right) d\widetilde{x}$$

$$= \int \left( \int_{\|x_1 - \widetilde{x}\| > r} p(x_1, \widetilde{x}) dx_1 \right) d\widetilde{x}.$$

Using $p(x_1, \widetilde{x}) = p(\widetilde{x} \mid x_1) p(x_1)$, we can change the order of integration:

$$\mathbb{E}_{\widetilde{x}}[Q(\widetilde{x})] = \int p(x_1) \left( \int_{\|\widetilde{x} - x_1\| > r} p(\widetilde{x} \mid x_1) d\widetilde{x} \right) dx_1.$$

Given $x_1$, the distribution of $\widetilde{x}$ is $N(x_1, \sigma^2 I_d)$. Let $Z = \widetilde{x} - x_1$. Then $Z \sim N(0, \sigma^2 I_d)$. The inner integral is $\Pr_{Z \sim N(0, \sigma^2 I_d)}[\|Z\| > r]$. Let $W = Z/\sigma$. Then $W \sim N(0, I_d)$. The inner integral becomes $P_G(r/\sigma) = \Pr_{W \sim N(0, I_d)}[\|W\| > r/\sigma]$. So, $\mathbb{E}_{\widetilde{x}}[Q(\widetilde{x})] = \int p(x_1) P_G(r/\sigma) dx_1 = P_G(r/\sigma)$.

We need to show $P_G(r/\sigma) \leq \delta \delta'$. We use the standard Gaussian concentration inequality: for $W \sim N(0, I_d)$ and $t \geq 0$,

$$\Pr[\|W\| \geq \sqrt{d} + t] \leq e^{-t^2/2}.$$

We want $P_G(r/\sigma) \leq \delta \delta'$. So we set $e^{-t^2/2} = \delta \delta'$. This implies $t^2/2 = \log(1/(\delta \delta'))$, so $t = \sqrt{2 \log(1/(\delta \delta'))}$. This choice of $t$ is real and non-negative since $\delta, \delta' \in (0, 1)$ implies $\delta \delta' \in (0, 1)$, so $\log(1/(\delta \delta')) \geq 0$. We set $r/\sigma = \sqrt{d} + t = \sqrt{d} + \sqrt{2 \log(1/(\delta \delta'))}$. Thus, for $r = \sigma \left( \sqrt{d} + \sqrt{2 \log \frac{1}{\delta \delta'}} \right)$, we have $P_G(r/\sigma) \leq \delta \delta'$.

With this choice of $r$, we have $\mathbb{E}_{\widetilde{x}}[Q(\widetilde{x})] \leq \delta \delta'$. By Markov's inequality,

$$\Pr_{\widetilde{x}}[Q(\widetilde{x}) > \delta] \leq \frac{\mathbb{E}_{\widetilde{x}}[Q(\widetilde{x})]}{\delta} \leq \frac{\delta \delta'}{\delta} = \delta'.$$

This means that $\Pr_{\widetilde{x}}[Q(\widetilde{x}) \leq \delta] \geq 1 - \delta'$, which is the desired statement:

$$\Pr_{\widetilde{x}} \left[ \Pr_{x \sim p_{\widetilde{x}}}[\|x - \widetilde{x}\| \leq r] \geq 1 - \delta \right] \geq 1 - \delta'.$$

$\square$

# F   Why Standard Langevin Dynamics Fails

As discussed in Section 3, after we get an initial sample $X_0 \sim p$ on the manifold, a natural attempt to get a sample from $p_y$ is to simply run vanilla Langevin SDE starting from $X_0$:

$$\mathrm{d}\, X_t \;=\; \Big(\widehat{s}(X_t) + \eta^{-2} A^{\mathsf{T}}(y - AX_t)\Big)\mathrm{d}\, t + \sqrt{2}\, \mathrm{d}\, B_t, \qquad X_0 \sim p \tag{13}$$

where $\widehat{s}(x)$ is an approximation to the true score $\nabla \log p(x)$. We now show that under any $L^p$ score accuracy assumption, the score error could get exponentially large as the dynamics evolves.

**Averaging over $y$ does not preserve the prior law.** We first consider the simplest one–dimensional Gaussian case of (13). Suppose $p = \mathcal{N}(0,1)$, $A = 1$, and noise $\xi = \mathcal{N}(0, \eta^2)$; so $y \sim \mathcal{N}(0, 1+\eta^2)$. Then with the perfect score estimator $\widehat{s}(X_t) = \nabla \log p(X_t) = -X_t$, (13) reduces to

$$\mathrm{d}\, X_t \;=\; \Big(-X_t + \eta^{-2}(y - X_t)\Big)\mathrm{d}\, t + \sqrt{2}\, \mathrm{d}\, B_t, \qquad X_0 \sim \mathcal{N}(0,1). \tag{14}$$

Recall that the hope of guaranteeing the robustness using only an $L^p$ guarantee is that at any time $t$, averaging $X_t$ over $y$ will preserve the original law $p$. We now show that this hope is unfounded even in this simplest case.

**Lemma F.1.** *Let $X_t$ follow* (14). *Averaging over $y \sim \mathcal{N}(0, 1+\eta^2)$, $X_t$ is Gaussian with mean $0$ and variance*

$$\mathrm{Var}(X_t) = e^{-2\alpha t} + \frac{1 - e^{-2\alpha t}}{\alpha} + \frac{(1 - e^{-\alpha t})^2}{1 + \eta^2} \leq 1,$$

*where $\alpha := \frac{1+\eta^2}{\eta^2} > 1$. In particular, $\mathrm{Var}(X_t) = 1 - \frac{1}{2(1+\eta^2)}$ at time $t^\star := \frac{\eta^2 \ln 2}{1 + \eta^2}$.*

*Proof.* Write the mild solution of (14):

$$X_t = X_0 e^{-\alpha t} + \eta^{-2} y \int_0^t e^{-\alpha(t-s)}\, \mathrm{d}\, s + \sqrt{2}\int_0^t e^{-\alpha(t-s)}\, \mathrm{d}\, B_s = X_0 e^{-\alpha t} + \frac{y}{\eta^2}\frac{1 - e^{-\alpha t}}{\alpha} + \sqrt{2}\int_0^t e^{-\alpha(t-s)}\, \mathrm{d}\, B_s.$$

Because $X_0, B$ are independent of $y$, conditional moments are

$$\mathbb{E}[X_t \mid y] = \frac{y}{\eta^2}\frac{1 - e^{-\alpha t}}{\alpha}, \quad \mathrm{Var}(X_t \mid y) = e^{-2\alpha t} + \frac{1 - e^{-2\alpha t}}{\alpha}.$$

Applying the law of total variance with $\mathrm{Var}(y) = 1 + \eta^2$ gives the stated formula.

Since $X_0$ and $B$ are independent of $y$, conditioning on $y$ gives

$$\mathbb{E}[X_t \mid y] \;=\; \frac{y}{\eta^2}\frac{1 - e^{-\alpha t}}{\alpha}, \qquad \mathrm{Var}(X_t \mid y) \;=\; e^{-2\alpha t} + \frac{1 - e^{-2\alpha t}}{\alpha}.$$

By the law of total variance and $\mathrm{Var}(y) = 1 + \eta^2$,

$$\mathrm{Var}(X_t) \;=\; e^{-2\alpha t} + \frac{1 - e^{-2\alpha t}}{\alpha} + \frac{(1 - e^{-\alpha t})^2}{1 + \eta^2}.$$

Using $\alpha = (1+\eta^2)/\eta^2$ and simple algebra, this simplifies to

$$\mathrm{Var}(X_t) \;=\; 1 - \frac{2\, e^{-\alpha t}(1 - e^{-\alpha t})}{1 + \eta^2},$$

which is at most $1$ and attains $1 - 1/[2(1+\eta^2)]$ when $e^{-\alpha t} = 1/2$, that is at $t^\star$. $\qquad\square$

Thus $\mathrm{Var}(X_t)$ first *shrinks* below $1$ (by a constant factor bounded away from $1$ when $\eta$ is small) before relaxing back to equilibrium. The phenomenon is harmless in one dimension but is catastrophic in high dimension.

**High-dimensional amplification.** Let $p = \mathcal{N}(0, I_d)$, take $A = I_d$, and set $\eta^2 = 0.1$. Then with the perfect score estimator, (13) reduces to

$$\mathrm{d}X_t = \left(-X_t + \eta^{-2}(y - X_t)\right)\mathrm{d}t + \sqrt{2}\,\mathrm{d}B_t, \qquad X_0 \sim \mathcal{N}(0, I_d). \tag{15}$$

By Lemma F.1 applied coordinatewise, at time $t^\star := \dfrac{\eta^2 \ln 2}{1 + \eta^2}$, averaging over $y$ yields

$$X_{t^\star} \sim p_{t^\star} := \mathcal{N}(0, \sigma^2 I_d) \quad \text{with} \quad \sigma^2 = 1 - \frac{1}{2(1+\eta^2)} = \frac{6}{11}.$$

Hence $X_{t^\star}$ is exponentially more concentrated in high dimension. We next show that this concentration amplifies score-estimation errors exponentially with the dimension.

**Lemma F.2.** *Let $p = \mathcal{N}(0, I_d)$ and let $p_{t^\star} = \mathcal{N}(0, \sigma^2 I_d)$ with $\sigma^2 = \dfrac{6}{11}$. For any finite $k > 1$ and $0 < \varepsilon < 1$, there exists a score estimate $\widehat{s} : \mathbb{R}^d \to \mathbb{R}^d$ such that*

$$\mathbb{E}_{x \sim p}\left[\|\widehat{s}(x) - \nabla \log p(x)\|^k\right] \leq \varepsilon^k,$$

*yet*

$$\Pr_{x \sim p_{t^\star}}\left(\|\widehat{s}(x) - \nabla \log p(x)\| \geq e^{cd}\varepsilon\right) \geq 1 - 2e^{-cd}$$

*for some constant $c > 0$ depending only on $k$.*

*Proof.* Fix $k > 1$ and $0 < \varepsilon < 1$. Let $\sigma^2 = \dfrac{6}{11} \in (0, 1)$ and choose $\rho \in (0, \min\{1/2, 1/\sigma^2 - 1\})$. Define the shell

$$S_\rho := \left\{x \in \mathbb{R}^d : \left|\|x\|^2 - \sigma^2 d\right| \leq \rho\,\sigma^2 d\right\}.$$

Write $m := \Pr_{x \sim p}[x \in S_\rho]$ and $q := \Pr_{x \sim p_{t^\star}}[x \in S_\rho]$. Since $\|X\|^2/\sigma^2 \sim \chi_d^2$ under $p_{t^\star}$, the chi-square concentration inequality Lemma A.11 gives

$$q \geq 1 - 2\exp\left(-\frac{\rho^2 d}{8}\right).$$

Since $(1 + \rho)\sigma^2 < 1$, the Chernoff left-tail bound for $\chi_d^2$ yields

$$m \leq \Pr_{x \sim p}\left[\|x\|^2 \leq (1 + \rho)\sigma^2 d\right] \leq \exp(-Id), \qquad I := \frac{1}{2}\left((1+\rho)\sigma^2 - 1 - \ln((1+\rho)\sigma^2)\right) > 0.$$

Choose any unit vector $u$ and set

$$e(x) := M\,\mathbf{1}_{S_\rho}(x)\,u, \qquad \widehat{s}(x) := \nabla \log p(x) + e(x), \qquad M := \varepsilon\,m^{-1/k}.$$

Then

$$\mathbb{E}_{x \sim p}\left[\|\widehat{s}(x) - \nabla \log p(x)\|^k\right] = \mathbb{E}_{x \sim p}\left[\|e(x)\|^k\right] = M^k m = \varepsilon^k.$$

Moreover $\|e(x)\| \equiv M$ on $S_\rho$, hence

$$\Pr_{x \sim p_{t^\star}}\left[\|\widehat{s}(x) - \nabla \log p(x)\| \geq M\right] = \Pr_{x \sim p_{t^\star}}[x \in S_\rho] \geq 1 - 2e^{-\rho^2 d/8}.$$

Using $m \leq e^{-Id}$ we have $M = \varepsilon\,m^{-1/k} \geq \varepsilon\,e^{(I/k)d}$. Setting

$$c := \min\left\{\frac{I}{k}, \frac{\rho^2}{8}\right\} > 0,$$

which depends only on $\sigma$ and $k$, gives

$$\Pr_{x \sim p_{t^\star}}\left[\|\widehat{s}(x) - \nabla \log p(x)\| \geq e^{cd}\varepsilon\right] \geq 1 - 2e^{-cd}.$$

This completes the proof. $\qquad\square$

