# OpenReview forum: "Posterior Sampling by Combining Diffusion Models with Annealed Langevin Dynamics"
_NeurIPS.cc/2025/Conference — NeurIPS 2025 poster_

### Official Review · Reviewer_afB1 · 2025-06-22

**Clarity:** 3
**Significance:** 3
**Originality:** 3
**Rating:** 4
**Confidence:** 4

**Summary:**

The authors propose running annealed Langevin dynamics to sample from the posterior distribution when solving inverse problems of the form $y=Ax+\xi$. The authors also propose a number of theoretical results with respect to this approach, namely deriving an upper bound on the estimation error of the conditional score.

**Questions:**

See Strengths and Weaknesses.

**Ethical Concerns:**

["NO or VERY MINOR ethics concerns only"]

**Final Justification:**

After reviewing the paper again, I am swayed by the authors' arguments. I did not properly evaluate this work in my initial review, and after doing so I feel that it is worthy of acceptance for its' theoretical contributions.

**Limitations:**

yes

**Paper Formatting Concerns:**

I did not notice any formatting concerns.

**Quality:**

3

**Strengths And Weaknesses:**

**Strengths**

S1) The paper is reasonably well-written and easy to follow.

S2) The theoretical results are interesting and (as far as I know) novel.

**Weaknesses**

This paper suffers from three major weaknesses: novelty, discussion of related literature, and experimental evaluation. I outline these issues below.

**W1) Novelty**

- While the theoretical results are novel (to my knowledge), the proposed method - using annealed Langevin dynamics for posterior sampling in inverse problems - is not. However, the authors suggest that they are the first to propose this approach. This is problematic, as it impacts the novelty of this work significantly.
- [1] was the first modern diffusion paper (to the best of my knowledge) to propose sampling with annealed Langevin dynamics in the unconditional case, and [2] directly extends this approach to conditional sampling, using the same conditional score proposed by the authors of this work (see the equation on the bottom of page 6 of the submitted paper and Eq. 4 in [2]).  I fail to see a meaningful difference between the proposed work and [2].

**W2) Discussion of Related Literature**
- The current discussion of related diffusion literature is insufficient. While I can appreciate trying to have a focused discussion on related methods (due to the breadth of diffusion methods for inverse problems), the current discussion is insufficient. Major methods such as DDRM [3], $\Pi$GDM [4], and, perhaps most important, DAPS [5] are absent from the discussion (to name just a few methods). Consequently, the discussion of related work needs to be fundamentally revised to acknowledge the broader diffusion-for-inverse-problems literature.
- One very noticeable missing citation is [6]. [6] is the foundation on top of which most modern diffusion literature is built, and Langevin dynamics is also discussed/applied in that paper (mostly based on [1]).

**W3) Experimental Evaluation**
- The experimental evaluation of the proposed method is completely insufficient and inconsistent with the broader literature. There is no evaluation table for the selected problems! The evaluation should be rebuilt from the ground up with quantitative results (nominally, PSNR, LPIPS, FID) for each problem listed in a well-formatted table. I would advise the authors to look at how DPS [7] presents their results in Tables 1 and 2 of that paper for inspiration.
- In addition to presenting experimental results more clearly, the authors need to compare with more relevant baselines (e.g., [2, 3, 4, 5]) to better understand how their approach performs.

**Some Minor Weaknesses**

These things do not impact my score, but I hope that the authors will take them under advisement.

- It is good practice to differentiate scalar and vector quantities in the text. I would suggest something simple, like boldface to represent vectors/matrices. This makes reading and understanding the math much easier.
- Figure 5 has a lot of wasted white space. I would suggest reformatting the figure to be more space efficient. Image grid figures also look nicer (generally) with less white space.

[1] Y. Song and S. Ermon. 'Generative modeling by estimating gradients of the data
distribution', in Advances in Neural Information Processing Systems, 2019.

[2] A. Jalal, et al., ‘Robust Compressed Sensing MRI with Deep Generative Priors,’ in Advances in Neural Information Processing Systems, 2021.

[3] B. Kawar et al., ‘Denoising Diffusion Restoration Models,’ in Advances in Neural Information Processing Systems, 2022.

[4] J. Song et al., 'Pseudoinverse-Guided Diffusion Models for Inverse Problems,' in International Conference on Learning Representations, 2023.

[5] B. Zhang et al, 'Improving Diffusion Inverse Problem Solving with Decoupled Noise Annealing,' in Proceedings of the Computer Vision and Pattern Recognition Conference (CVPR), 2025.

[6] Y. Song et al, 'Score-Based Generative Modeling through Stochastic Differential Equations,' in International Conference on Learning Representations, 2021.

---

> ### Author Rebuttal · Authors · 2025-07-30
>
> Thank you for the thorough review of our submission. We appreciate your feedback, though we believe the assessment overlooks the core contributions of our work.
> This is a theory paper, and we hope it can be assessed based on the theoretical questions it addresses, the insights it provides, and the theorems it proves. We respectfully ask you to reconsider your evaluation based on these criteria.
>
> Below, we respond to your specific concerns.
>
>
> ### **Novelty**
>
> We’re glad you found the theoretical results novel.
> You expressed concern that the methods we used to obtain these results are too similar to existing approaches, which you feel significantly reduces the novelty of our work. We **strongly disagree** with this view.
>
> Our goal is to find a posterior sampling method that is provably robust. In this context, simplicity is a strength. Demonstrating that a relatively simple method (annealed Langevin dynamics) satisfies strong theoretical guarantees makes the result **more** meaningful, not less. In fact, if we could prove that even vanilla Langevin dynamics is a robust posterior sampling algorithm, that would be an even better result!  Except that such a statement is false, as we show in Appendix F, so we need to use annealing.
>
> > *I fail to see a meaningful difference between the proposed work and [2].*
>
> We find this comment puzzling. Paper [2] doesn’t provide any robustness theorem, or any theoretical analysis of annealing, or guidance on choosing a schedule.  We also want to clarify that our proposed algorithm is quite different from [2] in design and purpose.
>
>
> Specifically, [2] uses the same observation $y$ at all times, and just anneals down the measurement variance $\sigma_t$.  In contrast, our method simulates a measurement process by also adding variance to $y$, so that at stage $t$ the measurement $y_t$ being considered is drawn according to the annealed variance $\sigma_t$.  Our version ensures that on average, the marginal distribution of the intermediate samples $x_t \sim p(x \mid y_t)$ matches the original marginal $p(x)$; this is a key property to be able to bound the error in each stage.  We do not believe the algorithm in [2] could achieve our Theorem 1.1 for any annealing schedule.
>
> What also sets our work apart is that we don’t just use annealed Langevin dynamics — we analyze why and when it works, and offer a principled annealing schedule based on this analysis. That goes beyond what previous works have offered, which typically rely on heuristics or hand-tuned parameters.
>
>
>
> To sum up, **the fact that annealed Langevin dynamics has had empirical success only strengthens the impact of our results**. We
> * identify failure modes of existing approaches,
> * introduce an alternative annealing design that circumvents those issues, and
> * provide formal theoretical guarantees.
>
>
> > *However, the authors suggest that they are the first to propose this approach.*
>
> Where do we suggest this?  We never intended to suggest that "annealed Langevin dynamics" is a new idea, it's a pretty natural one.  We do believe that we are the first to prove a theorem about any version of annealed Langevin dynamics for posterior sampling.  Furthermore, our version of annealing does not appear in prior work.
>
> #### **Novelty of Techniques**
>  You acknowledged the novelty of our theoretical results, but we also want to highlight the novelty of our proof techniques. Our paper is the first  to introduce the idea of “controlling the expectation of intermediate distributions” in posterior sampling (see Section 3 for a detailed discussion). We also provide a complete proof framework for formalizing this idea, as laid out in the appendix and in the proof organization paragraph on page 9.
>
> Taken together, we are confident that our work is both theoretically significant and methodologically novel.
>
>
> ### **Related Literature**
>
> We really appreciate your suggestion on including more related papers. We were mostly focusing on papers that have theoretical guarantees, but we will cite and discuss more related works in the final version of the paper.
>
> ### **Experiments**
> Look, this is a theory paper and should be judged as such.  We view
> the experimental section as "better than nothing": it demonstrates
> that the algorithm can be implemented and perform OK, but not much
> more.  We don't discuss our experiments until the last page of our
> paper.
>
> Plenty of theory papers at NeurIPS have no experiments at all; please
> judge our paper as if it didn't have them, rather than rejecting 55
> pages of theory over 1 page of unconvincing experiments.
>  Turning theoretical ideas into SOTA  systems that beat all baselines is a substantial effort in itself and goes beyond the intended scope of this paper.
>
> ### **Formatting suggestions**
> We appreciate your formatting and clarity suggestions. We’ll boldface vectors/matrices and tighten Figure 5 in the final version of the paper.
>
>
> ---
>
> We hope this clarifies the novelty and significance of our work. We’d be grateful if you could reconsider your evaluation in light of these points.

---

> > ### Comment · Reviewer_afB1 · 2025-08-05
> >
> > My thanks to the authors for their thorough reply to my concerns. I will admit that I do not feel that I properly evaluated this work as primarily theoretical; this is my mistake. To fix this, I opted to review the paper again.
> >
> > After looking at the paper again, and reading the other reviews and the authors' replies to them, I agree with the points made by the authors in their rebuttal to me and have decided to raise my overall score to a 4 (borderline accept). I still feel strongly about my points made with respect to the discussion of related literature. The way new work is positioned in the broader literature is highly important. Even if some of the citations I suggested do not provide theoretical results, they are certainly related and worth discussion. If the authors could provide this revised discussion, I would be comfortable raising my score further to a 5. Even so, after taking another look at the paper over the last few days, I can conclude that I initially reviewed it incorrectly and that I do think that theoretical contributions warrant acceptance.

---

> > > ### Author Response · Authors · 2025-08-07
> > >
> > > Thank you so much for taking the time to look at our paper again and for your thoughtful feedback. We truly appreciate your willingness to re-evaluate the submission and your recognition of its theoretical contributions.
> > >
> > > We completely agree that a thorough discussion of the related literature is essential. In the revision, we will expand the discussion on related works:
> > >
> > > > Naturally, researchers have looked to diffusion processes for more general and robust posterior sampling methods. The main difficulty is that the smoothed score of the posterior involves \$\nabla\_x \log p(y \mid x\_{\sigma\_t^2})\$ rather than the tractable unsmoothed term \$\nabla\_x \log p(y \mid x)\$. Because the smoothed score is hard to evaluate exactly, a range of approximation techniques has been proposed [1, 2, 3, 4, 5, 6]. One prominent example is the DPS algorithm [2]. Other methods include Monte Carlo approximations [7, 8, 9], singular value decomposition [10, 11, 6], and schemes that combine corrector steps with standard diffusion updates [12, 13, 14, 15, 16, 17, 18, 19, 20]. These approaches have shown promising empirical results, but none offer guarantees for fast and robust sampling.
> > >
> > > > Several recent studies [21, 22, 10] use various annealed versions of the Langevin SDE as a key component in their diffusion-based posterior sampling method and achieve strong empirical results. Still, these methods provide no theoretical guidance on two key aspects: how to design the annealing schedule and why annealing improves robustness. None of these approaches come with correctness guarantees for the overall sampling procedure.
> > >
> > > We will also add references [23] and [24] when introducing diffusion processes and annealed Langevin dynamics earlier in the paper.
> > >
> > > If there are any additional references or directions you would like to see discussed, we would be more than happy to include them. We are happy to refine the text to make the final version as clear and complete as possible.
> > >
> > > [1] B. Boys et al. 'Tweedie moment projected diffusions for inverse problems', 2024.
> > >
> > > [2] H. Chung et al. 'Diffusion posterior sampling for general noisy inverse problems', 2023.
> > >
> > > [3] X. Meng and Y. Kabashima. 'Diffusion model based posterior sampling for noisy linear inverse problems', 2024.
> > >
> > > [4] L. Rout et al. 'Beyond first-order Tweedie: Solving inverse problems using latent diffusion', 2024.
> > >
> > > [5] J. Song et al. 'Pseudoinverse-guided diffusion models for inverse problems', 2023.
> > >
> > > [6] Y. Wang et al. 'Zero-shot image restoration using denoising diffusion null-space model', 2023.
> > >
> > > [7] G. Cardoso et al. 'Monte Carlo guided diffusion for Bayesian linear inverse problems', 2024.
> > >
> > > [8] Z. Dou and Y. Song. 'Diffusion posterior sampling for linear inverse problem solving: A filtering perspective', 2024.
> > >
> > > [9] Z. Wu et al. 'Principled probabilistic imaging using diffusion models as plug-and-play priors', 2024.
> > >
> > > [10] B. Kawar et al. 'SNIPS: Solving noisy inverse problems stochastically', 2021.
> > >
> > > [11] B. Kawar et al. 'Denoising diffusion restoration models', 2022.
> > >
> > > [12] J. Chen and H. Liu. 'An alternating direction method of multipliers for inverse lithography problem', 2023.
> > >
> > > [13] H. Chung and J. C. Ye. 'Score-based diffusion models for accelerated MRI', 2022.
> > >
> > > [14] H. Chung et al. 'Improving diffusion models for inverse problems using manifold constraints', 2022.
> > >
> > > [15] U. Kamilov et al. 'Plug-and-play methods for integrating physical and learned models in computational imaging: Theory, algorithms, and applications', 2023.
> > >
> > > [16] X. Li et al. 'Decoupled data consistency with diffusion purification for image restoration', 2024.
> > >
> > > [17] B. Song et al. 'Solving inverse problems with latent diffusion models vi yea hard data consistency', 2024.
> > >
> > > [18] Y. Song et al. 'Solving inverse problems in medical imaging with score-based generative models', 2022.
> > >
> > > [19] Y. Zhu et al. 'Denoising diffusion models for plug-and-play image restoration', 2023.
> > >
> > > [20] M. Arvinte et al. 'Deep J-SENSE: Accelerated MRI reconstruction via unrolled alternating optimization', 2021.
> > >
> > > [21] A. Jalal et al. 'Robust compressed sensing MRI with deep generative priors', 2021.
> > >
> > > [22] B. Zhang et al. 'Improving diffusion inverse problem solving with decoupled noise annealing', 2025.
> > >
> > > [23] Y. Song and S. Ermon. 'Generative modeling by estimating gradients of the data distribution', 2019.
> > >
> > > [24] Y. Song et al. 'Score-based generative modeling through stochastic differential equations', 2021.

---

### Official Review · Reviewer_Q4o5 · 2025-06-24

**Clarity:** 3
**Significance:** 3
**Originality:** 3
**Rating:** 4
**Confidence:** 3

**Summary:**

The authors in [1] show that in general posterior sampling using diffusion models is intractable. However, under some conditions on the distribution (i.e., local or global log-concave $p(x)$) and $L^4$ error bounds the authors present a provably correct algorithm for posterior sampling.

[1] Gupta et al. "Diffusion Posterior Sampling is Computationally Intractable" (2024)

**Questions:**

- Is Theorem 1.2 limited to Gaussian noise $x_0 = x + \eta, \eta \sim \mathcal{N}(0, \sigma^2 I)$ or can it be extended to other noise types (e.g. Poisson, mixed Gaussian-Poisson)?
- Why do you have equality for the equation after line 218?
- page 1, line 26: I do not understand the sentence "it is 2-competetive with optimal in any metric" ?
- Why do you express Theorem 1.1 and Theorem 1.2 as existence results (i.e., "there exist an algorithm...") and not directly state that Algorithm 1 satisfies these properties?
- I think the result in Theorem 1.2 is more relevant for practical applications, as local log-concave is weaker than global log-concave. Looking at Section 3.3 (page 8) and Algorithm 1 it seems that you are not using the local log-concave condition? On page 2, line 67-74 you motivate the local condition, but Algorithm 1 does not make use of an initial reconstruction $x_0$.
- I do not really understand the results in Figure 3. Why does the $L_2$ increase if you decrease the step size, is this not counter-intuitive? Also DPS seems to be better than your method in FID for both super-resolution and Gaussian deblurring?

**Ethical Concerns:**

["NO or VERY MINOR ethics concerns only"]

**Final Justification:**

The authors have addressed all of my points. I raised my score to 4. I still think the paper is an important response to [1], showing that posterior sampling is possible under certain conditions on the distribution. However, I have not carefully read all steps of the proof, as most of it is part of the Appendix.

I agree with reviewer afB1 that the paper would benefit from a stronger empirical evaluation.

[1] Gupta et al. "Diffusion Posterior Sampling is Computationally Intractable" (2024)

**Limitations:**

The authors do not address limitations.

**Quality:**

3

**Strengths And Weaknesses:**

### Strengths:

The paper is a good response to [1], proving that one can do posterior sampling with score-based models under additional assumptions to the underlying distribution $p(x)$. The local log-concavity conditions is well motivated (e.g. Corollary 1.3 for compressed sensing). Further, the presentation of the paper is good - including Section 3 "Techniques" to present important details of the proof.

### Weaknesses:

- You always state that you need *merely* or *just* an $L^4$-error bound. Do you mean *merely* in comparison to the MGF bound? Because, in my opinion an $L^4$ error bound is still quite strong, given that the score matching loss is formulated in $L^2$.
- On page 6 you use the estimator for the conditional score as $\hat{s}_y(x) = \hat{s}(x) + A^T(y - Ax)/\eta^2$. It is fairly known, that is approximation is quite rough. Why do not use other approximation, e.g. DPS? Does the proof only work for your approximation?
- You cite MRI as an application in multiple parts of the paper (page 1, line 32 or page 2, line 67 and even the abstract), so I would expect to also see MRI experiments.

Minor things:
- page 1, line 25: In MRI the forward operator should be in $\mathbb{C}^{m \times d}$.
- page 3, line 98: Missing reference for "posterior sampling is know to the near-optimal"
- page 3, line 99: Grammar


[1] Gupta et al. "Diffusion Posterior Sampling is Computationally Intractable" (2024)

[2] Chung et al. "Diffusion Posterior Sampling for General Noisy Inverse Problems" (2024)

---

> ### Author Rebuttal · Authors · 2025-07-30
>
> Thank you for your review. We address your comments and questions below:
>
> > *$L^4$ bound requirement*
>
> Our requirement of $L^4$-accurate scores for posterior sampling is *significantly weaker* than the MGF bound required for log-concave sampling in general; [1] shows that score approximations that are $L^p$-accurate are insufficient for Langevin convergence for *any* $1 \le p < \infty$.
>
> We admit that $L^4$-closeness is still a strong requirement -- we suspect that even $L^2$ accurate scores are sufficient for provable sampling using our algorithm, and hope to address this in future work.
>
> > *On page 6 you use the estimator for the conditional score as $\widehat s_y(x) = \widehat s(x) + A^T (y - Ax)/\eta^2$. It is fairly known, that is approximation is quite rough. Why do not use other approximation, e.g. DPS? Does the proof only work for your approximation?*
>
> We are confused by this comment -- if $s(x)$ is known exactly, $s_y(x) = s(x) + A^T (y - Ax)/\eta^2$ is *precisely* the true conditional score. Thus, our approximation $\widehat s_y(x)$ is the natural one given only approximate scores $\widehat s(x)$.
>
> In contrast, DPS attempts to replace the scores of the *smoothed* conditional distributions with a lossy approximation. We only work with the *unsmoothed* conditional distribution, which is the one that Langevin uses.
>
> > *Is Theorem 1.2 limited to Gaussian noise or can it be extended to other noise types (e.g. Poisson, mixed Gaussian-Poisson)?*
>
> This is a great question -- our analysis certainly gives a natural starting point for analyzing other noise types, but doing so may require modifying the annealing schedule and is beyond the scope of this paper.
>
> > *2-competitive with optimal in any metric*
>
> This means that if the best possible reconstruction algorithm achieves error $\varepsilon$ with probability $1-\delta$ in any metric, posterior sampling has error at most $2 \varepsilon$ with probability $1-2\delta$.
>
> > *Why do you express Theorem 1.1 and Theorem 1.2 as existence results*
>
> Sure, we're happy to restate them more explicitly. The original phrasing follows a common convention in theory papers, where the focus is on the properties and guarantees rather than the specifics of the algorithm itself — to the extent that many such papers don’t even present the algorithm in full.
>
> > *Looking at Section 3.3 (page 8) and Algorithm 1 it seems that you are not using the local log-concave condition*
>
> Algorithm $1$ is for Theorem 1.1, the global log-concave case -- Algorithm $3$ in Appendix $E.2$ handles the locally log-concave setting, and makes use of the initial estimate $x_0$.
>
> > *Figure 3*
>
> In our experiments, we use the step size as a proxy for controlling the time spent running the algorithm. So, as we decrease step size, so that the time spent is decreased, we expect the sample obtained to be farther from the target conditional distribution, and closer to the starting sample $x_0$. Thus, we expect the sample to be farther from the ground truth, as we see.
>
> [1] Convergence in kl and rényi divergence of the unadjusted langevin algorithm using estimated score. Kaylee Yingxi Yang and Andre Wibisono. NeurIPS 2022 Workshop on Score-Based Methods, 2022.

---

> > ### Comment · Reviewer_Q4o5 · 2025-08-01
> > **Response**
> >
> > Thank you for your response.
> >
> > #### Posterior score approximation
> > Thank you for the clarification. Indeed, for the unsmoothed score function this is exactly the posterior score.
> >
> > #### Algorithm 1 and Algorithm 3
> > I think I did not see a direct reference to Algorithm 3 in the main paper.
> >
> > #### Figure 3
> > Why does the time spent decreased if you decrease the step size? Can you please explain how step size and time spent are related in your experiment.

---

> ### Author Response · Authors · 2025-08-03
>
> Thank you for taking the time to look over our previous response and for sending these follow-up questions. We address each point below.
>
> ### **Reference to Algorithm 3**
>
> Thank you for pointing this out. In the revised version, we will state Algorithms 2 and 3 more explicitly. Both are applications of Algorithm 1 to modified distributions that incorporate the auxiliary measurement $x_0$.
>
> The key idea behind applying Algorithm 1 in the local log-concavity setting is simple: once we start from the auxiliary measurement $ x_0 $ in the local log-concave region, we can apply Algorithm 1 directly. With high probability, the trajectory stays within this region, which ensures the algorithm remains effective. This approach underlies each of the results, as summarized below:
>
> | Theorem        | Setting                                                                 | Method                                                                                   | Target Distribution         |
> |----------------|------------------------------------------------------------------------|------------------------------------------------------------------------------------------|-----------------------------|
> | Theorem 1.1    | Global log-concavity                                                   | Directly run Algorithm 1                                                                 | $ p(x \mid y) $           |
> | Theorem 1.2    | Local log-concavity with a Gaussian measurement $ x_0 $              | Run Algorithm 1 using $ p(x \mid x_0) $ as the prior (Algorithm 2)                     | $ p(x \mid x_0, y) $      |
> | Corollary 1.3  | Local log-concavity with an arbitrary noisy measurement $ x_0 $      | Run Algorithm 2 but replace $ x_0 $ with $ x_0' = x_0 + \mathcal{N}(0, \sigma^2 I_d) $ (Algorithm 3) | $ p(x \mid x_0', y) $*    |
>
> \* Standard arguments show that this leads to a competitive compressed sensing algorithm, as detailed in Corollary 1.3.
>
> We will include explicit references and brief descriptions of Algorithm 2 and Algorithm 3 in the revised version to improve clarity.
>
>
> ### **Relationship between step size and time spent**
>
> Thanks also for raising this point. The “time” we refer to is the theoretical time variable in the continuous time Langevin dynamics rather than the computational time of the program. In our experiments, we simulate a Langevin process:
>
> $$
> dX_t = s_i(X_t) dt + \sqrt{2} dB_t
> $$
>
> and discretize it using a fixed number of steps $M$ and step size $h$. This results in a total simulation time of $T = M \cdot h$. Since $M$ is fixed, decreasing $h$ directly reduces $T$, which means the sample has evolved less from its initialization.
> This is what we mean by “less time spent” — it reflects how far the sample has progressed in distribution, not the actual runtime of the program.

---

> > ### Comment · Reviewer_Q4o5 · 2025-08-05
> > **Response**
> >
> > ### Reference to Algorithm 3
> >
> > Thank you for the clarification. I think adding such a Table (like in your response) to the manuscript will be helpful.
> >
> > ### Relationship between step size and time spent
> >
> > Thank you for the clarification. I think the reason for my confusion was that most often score-based diffusion models are run to the same terminal time $T$ (changing the step size, but also changing the number of steps to keep $T$ constant).

---

### Official Review · Reviewer_FxWY · 2025-06-30

**Clarity:** 3
**Significance:** 4
**Originality:** 3
**Rating:** 5
**Confidence:** 4

**Summary:**

## Summary
* This paper shows that there exist polynominal time DPS given $\epsilon$ error conditional score estimation, which is better aligned with the fact that DPS works practically. To the best of my knowledge, this paper is the first one to show there exist polynominal time approximated DPS. Despite many previous work have shown exact DPS can be achived asymptotically [Practical and Asymptotically Exact Conditional Sampling in Diffusion Models] [Diffusion Posterior Sampling for Linear Inverse Problem Solving: A Filtering Perspective], no complexity gauntree is given. Previous theory also shows that exact DPS requires expoential time [Diffusion Posterior Sampling is Computationally Intractable], while no complexity result is shown for approximate DPS.
* The authors shows empirically that their method can improve DPS, given a DPS initialization on FFHQ dataset and linear operators.

**Questions:**

* I am wondering if Algorithm 1 will work without init from DPS, on real world images such as FFHQ.

**Ethical Concerns:**

["NO or VERY MINOR ethics concerns only"]

**Final Justification:**

Thanks for the rebuttal. I still recommend to accept this paper.

I am really surprise that being a theoretical paper is considered a weakness by other reviewers. Personally I think this paper is a very interesting and important step over [Diffusion Posterior Sampling is Computationally Intractable]. And I encourage the other reviewers to take the theoretical contribution of this paper into consideration.

**Limitations:**

Yes.

**Quality:**

3

**Strengths And Weaknesses:**

## Strength
* Despite previous work show that exact DPS is computationally intractable [Diffusion Posterior Sampling is Computationally Intractable], this paper show that there exists polynomial time algorithm for DPS with small score error under some assumptions.
* The proposed theory can be used to construct a very competitive compressed sensing. Their approach can be used as a refiner to a compressed sensing algorithm, such that if there exist any other refiner can do $y$ accurate in expoential time, their approach can do $2y$ accurate in polynomial time.
* This paper seems to be the first one that can achieve DPS with arbitary small error with polynomial time at the same time.

## Weakness
* The log concave assumption of $p(x)$ is really strong and can be really far from true for natural images such as FFHQ. The general diffusion community has got rid of log concave assumption several years ago [Sampling is as easy as learning the score: theory for diffusion models with minimal data assumptions]. It might be better to use a different assumption, i.e. Gaussian Mixture Model [Theoretical insights for diffusion guidance: A case study for gaussian mixture models], which is a lot more aligned to the distribution of natural images.
* Another practical consequence of log concave assumption is that, it is possible that Algorithm 1 works only locally because natural images are log concave only very very locally. In fact, considering the fact that log concavity limits the distribution to single mode, the natural images are log concave only if we limit to the local area of a single image. Therefore in Section 4, I am not even sure if Algorithm 1 can work independently without a good initialization from DPS. That is to say, the line 7 in Algorithm 1 should really be $X_1 \sim \hat{p}(x|y)$, where $\hat{p}$ is some approximation to posterior.
* The DPS works well practically. However, it might work as a maximizing a posterior estimator, not a conditional score estimator. In fact, [Guidance with Spherical Gaussian Constraint for Conditional Diffusion] [Rethinking Diffusion Posterior Sampling: From Conditional Score Estimator to Maximizing a Posterior] show that the score error of DPS is very large. To generate a good sample, the approach proposed is sufficient but might not necessary. It is possible to generate good sample with very large score error, and the theoretical result of DPS might be improved without limiting score error.

---

> ### Author Rebuttal · Authors · 2025-07-30
>
> Thank you for your thorough review and positive evaluation. Below we address each of your comments point by point.
>
> > *The general diffusion community has got rid of log concave assumption several years ago.*
>
> You're right that unconditional diffusion models can work with very flexible distributions, thanks to access to smoothed scores. Posterior sampling, however, is more difficult: you cannot quickly compute the smoothed score of the posterior from the smoothed scores of the prior. As shown in [1], once we lose access to smoothed scores and have to work with unsmoothed ones, no polynomial-time algorithm exists for sampling from a general non-log-concave posterior. That’s why in the literature on unsmoothed-score samplers, log-concavity remains a common assumption.
> In fact, the hardness example in [1] is itself a Gaussian mixture, so replacing the prior by a mixture of Gaussians would not avoid the same barrier.
>
> > *It is possible that Algorithm 1 works only locally.*
>
> We agree, and that’s exactly what we formalize in Theorem 1.2: When the prior isn't globally log-concave, as long as it's log-concave in a local region around the target, our algorithm can still do posterior sampling — once it's localized to that region.
>
> > *The line 7 in Algorithm 1 should really be $X_1 \sim \hat p (x \mid y)$.*
>
> Algorithm 1 is designed for priors that are globally log‑concave, so sampling $X_{1}$ from $p(x)$ is sufficient. For priors that are only locally log‑concave, Algorithm 2 in Appendix E.2 replace the global prior by its localized counterpart -- just as you suggest.
>
> > *The approach proposed is sufficient but might not necessary. It is possible to generate good sample with very large score error, and the theoretical result of DPS might be improved without limiting score error.*
>
> You're absolutely right that DPS sometimes produces high-quality samples even with large score error, especially if it's implicitly maximizing the posterior. But in this work, our goal is sampling from the correct posterior — not just generating plausible outputs. Maximizing the posterior may give sharp samples, but they’re biased, even in simple settings like a mixture of two Gaussians.
>
> > *I am wondering if Algorithm 1 will work without init from DPS, on real world images such as FFHQ.*
>
> The experiment in Section 4 falls under the locally log‑concave setting, where an initializer inside the correct region is required. DPS supplies such an initializer, but a simpler choice like the pseudoinverse $A^{\dagger} y$ is also possible, albeit with weaker performance. For a truly globally log‑concave prior, Algorithm 1 would not need any special initialization.
>
> [1]: Diffusion Posterior Sampling is Computationally Intractable. Shivam Gupta, Ajil Jalal, Aditya Parulekar, Eric Price, and Zhiyang Xun. ICML 2024

---

### Official Review · Reviewer_EJdf · 2025-07-02

**Clarity:** 3
**Significance:** 2
**Originality:** 2
**Rating:** 4
**Confidence:** 3

**Summary:**

This paper looks at how to sample from the posterior distribution  p(x | y) , where y = Ax + \xi  is a noisy observation and the prior  p(x)  is known or well-approximated. Posterior sampling is useful in tasks like image inpainting, deblurring, and MRI reconstruction, but it’s usually hard to do in general.

The authors focus on the case where the prior is log-concave. Normally, Langevin dynamics can be used when the score function (the gradient of the log-density) is exact, but it is very sensitive to errors in score estimation.

This paper shows that by combining diffusion models with an annealed version of Langevin dynamics, it’s possible to do conditional sampling even when the score estimates are not perfect—only needing an  L^4  error bound instead of a much stronger condition. This is a nice theoretical improvement over previous work.

**Questions:**

- Is it possible to test or check in practice whether the L^4  error bound holds for learned score models?

- Can the authors give a simple example where their result applies and shows the improvement clearly?

- Could the method work for more general priors or other types of observations beyond the linear setting?

**Ethical Concerns:**

["NO or VERY MINOR ethics concerns only"]

**Final Justification:**

The authors explain more of my doubts, and according to this, I update my score.

**Limitations:**

The main limitation is that this is only a theory paper—there’s no algorithm or experiments. It also doesn’t connect to a lot of newer work on how Langevin dynamics performs when actually used with step sizes. The setting is limited to log-concave priors and linear problems.

**Quality:**

3

**Strengths And Weaknesses:**

Strengths:

The paper gives a new way to think about posterior sampling using recent tools from diffusion models.

It shows that weaker conditions on score accuracy are enough, which is important for practical settings.

The theory is clean and the goals are clear.

Helps connect ideas between different sampling approaches (Langevin, diffusion, score-based methods).

Weaknesses:

The paper is only theoretical—there’s no actual algorithm or method that someone can implement.

It doesn’t cover more general situations, like non-log-concave priors or nonlinear measurements.

It doesn’t mention or cite recent work on how Langevin methods behave.

While the score error bound is weaker than before, it still might be hard to get in real-world settings.

---

> ### Author Rebuttal · Authors · 2025-07-30
>
> Thank you for your review. We respond to your comments and questions below:
>
> > *The paper is only theoretical—there’s no actual algorithm or method that someone can implement.*
>
> This comment, as well as the related one in the limitations section, perplexes us -- we have implemented the algorithm and shown improvements over the existing DPS method; see the Experiments section for details.
>
> > *It doesn’t cover more general situations, like non-log-concave priors or nonlinear measurements.*
>
> Our analysis *does* cover non-log-concave priors as long as they are *locally* log-concave as we describe on page 2 -- see Theorem 1.2 for the formal statement. For general non-log-concave priors, it is impossible to design a polynomial time posterior sampler, as [1] shows.
>
> We thank you for asking about nonlinear measurements -- in general, unless the posterior density is log-concave, our analysis does not apply; for nonlinear measurements, log-concavity is not guaranteed. If this posterior is log-concave, our analysis gives a natural starting point for this setting, but a complete analysis is beyond the scope of this paper.
>
> > *It doesn’t mention or cite recent work on how Langevin methods behave.*
>
> We have provided a comparison with existing works in the Langevin literature on page 5 -- to reiterate, when the score function is known exactly and the distribution is strongly log-concave, the Unadjusted Langevin Algorithm is known to converge rapidly [2]. If an approximation to the score is known satisfying a strong MGF condition, [3] showed that Langevin converges, but fails to converge if the score approximation is only $L^p$-accurate for any $1 \le p < \infty$. Our analysis shows that for *posterior sampling*, an annealed version of Langevin converges even if the score approximation is only $L^4$-accurate.
>
> > *Is it possible to test or check in practice whether the $L^4$ error bound holds for learned score models?*
>
> This is a great question -- in practice we see that our method *does* improve upon DPS as noted above, which suggests that either the scores are being learned in $L^4$, or that our method continues to work even for $L^2$ accurate estimates.
>
> > *Can the authors give a simple example where their result applies and shows the improvement clearly?*
>
> One simple example is the Poisson distribution -- since it is a log-concave distribution, our algorithm is guaranteed to converge. In contrast, DPS suffers error depending on the Jensen gap, which is non-zero for this distribution.
>
> Note that standard Langevin fails to converge even for the Gaussian distribution, while our annealed version converges -- see Appendix F for a discussion.
>
>
> [1] Diffusion Posterior Sampling is Computationally Intractable. Shivam Gupta, Ajil Jalal, Aditya Parulekar, Eric Price, and Zhiyang Xun. ICML 2024
>
> [2] Theoretical guarantees for approximate sampling from smooth and log-concave densities. Arnak S Dalalyan. Journal of the Royal Statistical Society Series B: Statistical Methodology, 2017.
>
> [3] Convergence in kl and rényi divergence of the unadjusted langevin algorithm using estimated score. Kaylee Yingxi Yang and Andre Wibisono. NeurIPS 2022 Workshop on Score-Based Methods, 2022.

---

> > ### Comment · Reviewer_EJdf · 2025-08-07
> >
> > Thank you for your answers.

---

### Note · Authors · 2025-08-14

We thank the reviewers for their thoughtful comments and are glad that our rebuttal addressed the key concerns. We will revise the paper accordingly.

As all reviewers agree, posterior sampling is an important problem. But what can be shown theoretically? Recent work proved that it is intractable to design a posterior sampler that works as generally as for unconditional sampling. Our paper asks what natural conditions make it possible:
- Even for globally log-concave distributions, the problem is challenging; we give the first algorithm that is provably fast and robust.
- We extend this to the more realistic “locally log-concave” setting, where a sufficiently good warm start ensures the same guarantees.

Our theoretical analysis motivates a new way of annealing for Langevin dynamics. It is practical to implement and has strong potential for extension to broader scenarios. For example, it directly implies a competitive compressed sensing algorithm as a corollary.

Given the known intractability, it is hard to see what qualitatively stronger theorem one could hope to prove without imposing extra structure. Our work bridges the gap between impossibility results and practice, and provides a clean proof framework that others can build on. We believe this is a clear conceptual step with genuine practical value.

---

### Decision · Program_Chairs · 2025-09-17

**Decision:**

Accept (poster)

**Comment:**

This paper addresses the problem of sampling from a posterior defined by way of a diffusion model. Essentially, suppose a sample is drawn from a diffusion model, then put into a linear transformation, and then convolved with Gaussian noise. If that transformed+convolved output is observed, what is the posterior over the original image? This poses a significant computational challenge. This paper is shows that when annealed Langevin dynamics are used, samples from the posterior can be obtained with a provable bound on accuracy. This assumes that the diffusion model is log-concave. In fact, this holds even if the scores in the original model are only estimated with some error, although a moderately strong L^4 bound is needed. The method appears to work reasonably well in practice. All reviewers appreciated the work, although some asked for stronger empirical results.